



# A Multiagent Socio-hydrologic Framework for Integrated Green Infrastructures and Water Resource Management at Various Spatial Scales

Mengxiang Zhang and Ting Fong May Chui

Department of Civil Engineering, The University of Hong Kong, Hong Kong SAR, China

**Correspondence:** T. F. M. Chui (maychui@hku.hk)

**Abstract.** Green infrastructures have been widely used to manage urban stormwater, especially in water-stressed regions. They also pose new challenges to urban and watershed water resources management. This paper focuses on the green infrastructure-induced dynamics of water sharing in a watershed from three spatial scales. A multiagent socio-hydrologic model framework is developed to provide an optimization-simulation method for city-, inter-city-and watershed-scale integrated green infras-
tructures and water resource management (IGWM) that comprehensively considers the watershed circumstances-, the urban water managers-and the watershed manager-urban water managers interactions. We apply the framework to conduct three simulating experiments in the Upper Mississippi River Basin, the US. Four patterns in city-scale IGWM are classified and two dynamics of cost and equity in inter-city- and watershed-scale IGWM are characterized through various sensitivity, scenario, and comparative analyses. The modeling results could advance our understanding of the role of green infrastructures in urban
and watershed water resources management and assist water managers in making associated decisions.

## 1   Introduction

In recent decades, urban water scarcity worldwide caused by urbanization, population growth, and climate change necessitates new approaches to increase water supply (McDonald et al., 2014; Schewe et al., 2014). Green infrastructures (GIs), which are decentralized nature based measures for rainwater and stormwater capture and recharge, have prevailed in many countries,
such as the United States, the United Kingdom, China, and Australia (Dietz, 2007; Coutts et al., 2013; Zhou, 2014; Li et al., 2017). GI systems, as demonstrated by many studies, are effective in increasing water availability and reducing urban flooding, which, to some extent, can supplement the centralized water services provided by grey infrastructure systems (Rozos et al., 2010; Jayasooriya and Ng, 2014). Therefore, more and more cities start to take GIs into account as a component of an urban water system, that is, to generate a hybrid urban water system (HUWS) via combining GIs with the existing grey infrastructures
(Daigger and Crawford, 2007; Sapkota et al., 2014). A HUWS includes two inter-dependent subsystems: centralized (i.e., grey infrastructures) and decentralized (i.e., GIs) systems (Dandy et al., 2019). In a centralized system, different types of water supply, storage, treatment and distribution facilities are constructed and maintained to provide various water supply, sanitation and drainage services to urban water users (Sitzenfrei et al., 2013). And, in decentralized systems, multiple GIs with different hydrologic performances, which are divided into two groups depending on the desired flow regime; retention- and infiltration-



based GIs (Fletcher et al., 2013), are built in an urban area to manage rainwater and stormwater. To be specific, the retention-based GIs, such as green roofs, cisterns, and rain barrels, can retain stormwater, which can be directly used for non-potable water uses, or with filtration and disinfection for potable water uses (McArdle et al., 2011). Furthermore, the infiltration-based GIs, e.g., rain gardens, porous pavements, and wetlands, can restore some aspects of pre-development flow regimes in receiving water through the recharging of subsurface flows and groundwater, which can be indirectly used by groundwater

abstraction (Endreny and Collins, 2009). In general, such a HUWS generally makes urban hydrologic regimes, water supply and demand more complicated (Sapkota et al., 2014; Goonrey et al., 2009), which poses a series of challenges to urban water management and watershed development. For example, such an increased complexity induced by HUWSs may lead to the failures of conventional water management approaches that cope with the issues of grey and green infrastructure systems separately (Poustie et al., 2015). To manage and operate a HUWS scientifically, therefore, it is necessary for an urban water

manager (UWM) to develop new urban water management frameworks considering GIs development and rainfall utilization, that is, integrated GIs and water resources management (IGWM) at a city scale.

For a city-scale IGWM framework, it is in fact a coupling urban land and water management regime, which refers to GIs construction decisions and water supply portfolios choices. Specifically, a UWM needs to supply water extracted from rivers and aquifers via grey infrastructure systems to satisfy various water demands. Meanwhile, it also plans to construct GI systems

to collect, store and use rainwater resources to supplement the urban water supply. The whole process of a city-scale IGWM is interactive because a HUWS is a human-environment system (Sapkota et al., 2014). On the one hand, various external environmental factors, such as hydroclimatic, geographic, and socioeconomic conditions, can affect the decision-making of IGWM. For example, the magnitude and frequency of upstream inflow and precipitation directly determine the available amounts of surface water and stormwater to a city; the socioeconomic development level of a city (e.g., residents' housing

types and water use habits) can influence selections of types, sizes, and locations of GIs. IGWM decisions made by a UWM, on the other hand, can also change external socio-hydrologic circumstances in turn. Hydrologically, developing GIs within a urban area can increase the infiltration rate and rainwater harvesting capacity of the relevant land, which enables a city to retain and detain more water during a rainfall event, thereby changing the urban water cycle. Socially, alternative water sources driven by GIs enrich water users' choices and gradually change their water use habits (Kallis, 2010), which tips the original

water supply and demand balance.

Besides, there are generally multiple urban areas along a river and they share water resources within a watershed, all of which hope to access enough affordable water for maintenance and development. In that watershed, the changes in the urban water cycle induced by a city-scale IGWM may shift the inflow and outflow of an urban area. The effects of such a regional hydrologic shift may expand gradually throughout the watershed because the river network closely connects urban areas. Since

the relevant stakeholders share the common land and water resources within a watershed, the decision-making of IGWM in an urban area might affect that in other areas. Some studies demonstrated that over-development of GIs might decrease the river flow to downstream areas (Glendenning et al., 2012), which may be detrimental to stream health (Fletcher et al., 2007). In comparison, others showed that expansion of GIs might, to some extent, decline the variability of river flow, which is beneficial to downstream water supply (Golden and Hoghooghi, 2018). Under the circumstances, all UWM within a watershed make





their own IGWM decisions rationally, not only depending on natural hydrologic conditions but also anthropogenic activities from upstream urban areas. That is, the behavior of an upstream UWM can affect that of the subsequent downstream UWM because the water is transported along the river. Such interactive behavior among UWMs for IGWM will be bound together to form a unique sequential multi-players interaction at an inter-city scale, which can be characterized as the Markov property (Frydenberg, 1990) - the future hydrologic regime in a urban area depends on the only the current hydrologic regime partly

influenced by the decision-makings of city-scale IGWM in the upstream area and that decisions of the urban area itself. It might lead to unexpected watershed-level performance. So, it is worthwhile to extend the scope of IGWM to a watershed level and explore the interactions among urban areas driven by multiple city-scale IGWM decisions, i.e., IGWM at an inter-city scale.

Understanding IGWM interactions at an inter-city scale is just one side of the picture, and the other side is how watershed water policies influence them. Since the interactive behavior of IGWM among multiple UWMs may significantly alter hydro-

logical regimes and social circumstances in the entire watershed, it might offer a difficulty for a watershed manager (WM) who continues to use traditional water policy approaches that generally ignore such interactions among urban areas. In general, an unordered IGWM at an inter-city scale might lead to an inequitable, costly, and unsustainable sharing of water resources in a watershed (Müller et al., 2017). A WM needs to prescribe various policies and strategies to regulate all UWMs' decision-makings of city-scale IGWM to achieve set targets for watershed environments' health, stabilization, and sustainability. In the

context, a WM and multiple UWMs will construct a bi-level system, which follows a specific hierarchical decision rules for the leader and the multiple followers. That is, the WM at the upper level first set a water policy at the watershed level, and then the multiple UWMs at the lower level make their own city-scale IGWM decisions with full knowledge of the policy set by the WM; the WM also can adjust its own policy prescription based on the rational UWMs' reactions. In economic theory, this hierarchical interactive process between a WM and multiple UWMs is described as a Stackelberg game (Simaan and Cruz,

1973) . Hence, it is also important to introduce a WM into the multi-UWMs system for watershed-scale IGWM and discuss the effect of various water policies on the dynamics of multiple city-scale IGWM decisions, namely, IGWM at a watershed scale.

Currently, there are rich studies related to city-scale IGWM, most of which focused on studying the following three critical aspects: 1) integrating water management framework for HUWS, 2) assessing rainwater potential 3) modeling urban water cycle of a HUWS. The first aspect involves building an integrated water management framework that combines conventional

water supply systems with stormwater/rainwater harvesting schemes (Daigger, 2009). To determine the appropriate stormwater harvesting scheme option under different settings, Goonrey et al. (2009) developed a decision-making framework. To address on-site and catchment urban surface water issues, Ellis (2013) introduced sustainable drainage systems into a GI framework. Dandy et al. (2019) presented an integrated framework to help UWMs select and evaluate stormwater harvesting systems. The second aspect of the studies is mainly focused on the quantitative evaluation of rainwater/stormwater resources supply options

in various urban areas. Kim et al. (2022) investigated the impact of water management strategies, such as rainwater harvesting, on urban water demand in Filton Airfield, UK, using water demand profiles and urban water cycle simulations. Kim et al. (2022) developed a framework to assess a variety of centralized and decentralized water supply options, including rainwater harvesting and groundwater extraction via private wells, for meeting urban water demand in southern India. Souto et al. (2022) studied the effects of a rainwater harvesting system in reducing the demand for drinking water in the city of Goiânia by using two





water balance models. Researches addressing the third aspect has focused on developing models to simulate urban hydrologic regimes within a HUWS, such as the Aquacycle (Mitchell et al., 2001), Urban Cycle (Hardy et al., 2005), City Water Balance (Last, 2011), UrbanBEATS (Bach et al., 2015) and SUWMBA (Moravej et al., 2021). These models, to some extent, allow users to improve their understanding of the impacts of various GIs options at different scales and to assess the comprehensive performance of HUWSs across the entire urban water cycle. Although there is a rich literature addressing issues of city-scale

IGWM, there is very little work that comprehensively considers the selection of centralized and decentralized water supply options, as well as the decision-making associated with GIs construction plans within a HUWS in changing environments. This has become a key issue of concern for UWMs. Therefore, more in-depth studies are needed to develop a decision-making framework that can assist authorities in making effective decisions about IGWM at a city scale.

Besides, several studies associated with inter-city scale IGWM have attempted to investigate the issues of interactions be-

tween water users in a shared water resources system, especially in irrigation systems (Barreteau and Abrami, 2007; Berger et al., 2007). To better understand the effects of water users or managers' behaviors and their interactions in a watershed-scale water resource system, a diverse range of multi-agent systems (MAS) have been constructed and used (Barreteau and Abrami, 2007). Berglund (2015) reviewed an emerging area of research within water resources management that uses agent-based models and MAS to simulate the water resources allocation, and to predict the performance of infrastructure design. Yang

et al. (2009) built a MAS, which includes multiple distributed land and water users within a watershed, to mimic interaction schemes among water users in real-world watershed management problems. Giuliani and Castelletti (2013) explored the effect of different levels of cooperation and information exchange among water users on the upstream-downstream water conflict in a large-scale water resources system through using a well-designed MAS. Some studies have also focused on interactions between water users and the watershed environment, coupling MAS with watershed hydrologic models (He, 2019; Du et al.,

2020). Reeves and Zellner (2010) integrated an agent-based land-use model with MODFLOW to anaylze the complexity inherent in land-use change and its effect on groundwater resources. Montalto et al. (2013) constructed a agent-based framework to analyze the effect of different spatiotemporal distributions of GIs determined by numerous household decision-makers on urban water cycle in South Philadelphia. Du et al. (2020) coupled a rule-based agent-based model with a distributed hydrologic model, i.e., GSFLOW, to simulate spatial and temporal dynamics of human-hydrological interactions. While these studies can

inspire this study, there are some common limitations for modeling IGWM at an inter-city scale. Firstly, the studies of interactions between water users tend to focus on agricultural regions rather than urban regions, and do not address the issues of interaction between urban areas driven by developing GIs and using rainwater sources within a watershed. Secondly, rule-based models have been widely used to simulate the behavior of water users or managers in a water resource system, but are unable to describe the complex decision processes of city-scale IGWM due to over-simplified decision rules.

In the field of research within IGWM at a watershed scale, several studies have focused on the evaluation or design of water policies to manage interactions or conflicts among multiple water users within a watershed via using various MAS (Berger and Ringler, 2002; Akhbari and Grigg, 2013; Lin et al., 2020). For instance, Kock (2008) developed and applied two agent-based models of society and hydrology to test relations between different water policies in a watershed and the level of water conflict in that watershed. Kanta and Zechman (2014) built a MAS framework by integrating a urban water demand and a supply model





and considering water users and managers as agents. A wide set of water policies, such as conservation strategies and interbasin
transfer strategies, set by the water managers and the associated responses from the water users were simulated via using the
framework. Darbandsari et al. (2020) proposed a new conflict resolution model to assess different water management policies
through simulating the interactions of all water users' behaviors. A Stackelberg game theory-based model was used to describe
the leader-follower interactions between water users and managers within a basin. Although these previous studies can provide
guidance for IGWM at a watershed scale in an exploratory way, there is still a gap in building a sound and flexible watershed
policy framework for the design and evaluation of various water strategies to allocate limited water resources to urban areas
that develop GIs and use rainwater resources, as well as for the simulation of the complicated responses of water supply and
GIs development in each urban area to these strategies.

This paper focuses on integrated GIs and water resources management in a watershed that consists of multiple urban areas,
which build green infrastructures and use rainwater, at three spatial scales. For the issue of IGWM at a city scale, we examine
how a UWM can make the best configuration for centralized (i.e., surface water and groundwater) and decentralized (i.e.,
rainwater and stormwater) water resources, as well as the relevant plan for GIs construction, in order to make optimal use of
limited water supplies with minimal cost; For the issue of IGWM at an inter-city scale, we also investigate how the behavior
of city-scale IGWM determined by the upstream UWMs can affect that of the downstream UWMs in a watershed; Finally,
for the issue of IGWM at a watershed scale, we explore how a watershed water policy - a streamflow penalty strategy -
set by WM can influence all UWMs' behaviors of city-scale IGWM and their interactions with each other. Therefore, we
build a multiagent socio-hydrologic framework to solve IGWM-related issues mentioned above. To be specific, it includes
1) an agent-based model of urban water manager (ABM-UWM), which is developed by coupling an economic optimization
model and a hydrologic simulation model, to determine the optimal decision-making of city-scale IGWM for a UWM in
changing watershed settings; 2) a multiagent system for multiple UWMs (MAS-UWM), which is constituted via integrating
the above ABM-UWMs with a streamflow routing models based on the Markov property, to simulate interactions among
them in inter-city-scale IGWM, and assess its impact on the whole watershed; and 3) a bi-level multiagent system (BL-MAS),
which is constructed by combining the above MAS-UWM with an agent-based model of watershed manager (ABM-WM)
on account of the Stackelberg game theory, to mimic the interactions and feedbacks between a WM and multiple UWMs
driven by the implementation of a watershed water policy in watershed-scale IGWM, thereby designing an optimal watershed
strategy. Besides, we also demonstrate our framework to conduct three numerical experiments based on a realistic basin - the
Minneapolis-La Crosse section of the Upper Mississippi River, the US. The results obtained from these experiments allow us
to characterize and classify the decision-making of city-scale IGWM for a UWM under different circumstances, to analyze
the socio-hydrologic dynamic of watershed induced by inter-city-scale IGWM, and to assess the role of a water policy in
watershed-scale IGWM.

The outline of this paper is as follows. Section 2 presents a socio-hydrologic model framework which is established to
simulate decision-makings of city-scale IGWM for UWM and dynamics of inter-city- and watershed-scale IGWM resulting
from interactions 1) among UWMs hydrologically, and 2) between the UWMs and the WM institutionally. In Section 3, we
present three simulation experiments based on a case of the Upper Mississippi River basin to generate understanding and



insight to the IGWM at three spatial scales. In Sections 4 we discuss and analyze the results of the simulation experiments. The features of optimum IGWM for UWM are identified via sensitivity analysis, and the characteristics of UWM-UWM and UWM-WM interactions in inter-city- and watershed-scale IGWM are assessed and summarized by scenarios and comparative analysis, respectively. Section 5 ends this paper with a conclusion, limitations, and proposals for future research.

## 2 Methodology

This section proposes a multiagent socio-hydrologic framework for solving the issues mentioned above of the IGWM at three spatial scales. The framework includes 1) two agent-based models (i.e., urban water and watershed manager agents) to have a more realistic representation of city- and watershed-scale IGWM decision behavior, separately; 2) two multiagent systems to simulate the two types interactions (i.e., UWM-UWM and UWM-WM interactions) in inter-city-and watershed-scale IGWM based on the geographic and social connections. Fig. 1 shows a watershed system for IGWM at three spatial scales. In the

framework, an agent-based model for UWM is used to deal with the issue of IGWM at a city scale (See Fig. 1 A). A multiagent system for UWMs is used for IGWM at an inter-city scale(See Fig. 1 B) , and a bi-level multiagent system is built to simulate the dynamics of IGWM at a watershed scale.

### 2.1 Agent-based model (ABM) for IGWM at a city scale

Fig. 1 (A) illustrates city-scale IGWM activities and resulting changes in urban hydrologic cycles. As shown in Fig. 1 (A), an

UWM needs to consider water supply portfolios and GIs construction simultaneously to meet various urban water demands cost-effectively. In the water supply portfolios selection, four types of water sources, which are classified by water intake locations, can be chosen to withdraw, treat, divert water to an urban area; 1) surface water, 2) groundwater, 3) rainwater, and 4) stormwater (Steffen et al., 2013). Surface and groundwater, which are used for supplying potable and non-potable water, can be obtained from the river near an urban area and the aquifer beneath an urban area via grey infrastructure systems. Rainwater

and stormwater can be collected from roofs and drains separately via specific GI systems. In the GIs construction decision, three types of GIs - infiltration-based GIs, rainwater, and stormwater harvesting systems - can be built to use rainfall resources within an urban area directly or indirectly. The infiltration-based GIs can change infiltration rates of pervious surfaces, which can enhance groundwater recharge. The rainwater and stormwater harvesting systems can collect and store rainfall, which provides non-potable water. To simulate UWM's behavior of city-scale IGWM, an agent-based model for UWM is developed

via coupling the IGWM optimization model (IGWM-OM) and the urban water balance simulation model (UWB-SM). In the model, monthly optimal IGWM decisions are represented by the IGWM-OM, whereas changes in urban hydrologic processes induced by the decisions are simulated in the UWB-SM. The IGWM-OM interacts with the UWB-SM via a coupling strategy. Using this coupling strategy, we can use a simulation-based optimization approach to estimate and predict the UWMs' decision-making of IGWM and the induced urban hydrologic dynamic under different socioeconomic and hydroclimatic conditions.

The UWB-SM, a lumped urban water system model, is developed to describe the dynamic of urban water balance induced by city-scale IGWM decisions. The model includes 1) all urban water flows (natural and anthropogenic), 2) grey urban water





**Figure 1.** Schematic diagram of IGWM at three spatial scales.

systems, and 3) green infrastructures systems to simulate the interactions between the water supply–wastewater discharge network, the rainfall–stormwater runoff network, and the three types of GIs systems within an urban area. In the UWB-SM, the urban water cycle receives input both from rainfall and river inflow, which together pass through the grey and green infrastructures system and output in the form of evapotranspiration and river outflow. And water flows movement between seven storage units within an urban region - roof, other surfaces, rainwater, and stormwater harvesting systems, shallow soil layer, aquifer, and river. The state of these storage units is used to measure the total water balance within an urban area, thereby calculating available amounts of water resources in a period. Hydrologically, various surface-subsurface water interactions and pipe network exfiltration and infiltration are considered via simulating water fluxes transfers between these storage units on the basis of the water mass conservation principle. Besides, the UWB-SM be set as that a UWM agent can supply surface water and groundwater by extracting from the river and the aquifer storage units. It also can access rainwater and stormwater collected by rainwater and stormwater harvesting systems. Details of the UWB-SM are formulated in Appendix A2.





The IGWM-OM is built to simulate optimal IGWM decisions for a UWM agent - it needs to determine cost-effective construction areas of three types of GIs (i.e., infiltration-based GIs, rainwater, and stormwater harvesting systems) to increase rainwater availability according to urban land limitation and also to determine least-cost monthly water supply portfolios (i.e., the proportions of surface water, groundwater, rainwater, and stormwater supply) to satisfy urban, diverse water demand subjecting to associated allowable amounts of water sources. In the IGWM-OM, the objective function - to minimize the annual IGWM cost for the UWM agent - consists of the annual costs of GIs construction, water supply and wastewater drainage. The relevant constraint conditions includes the GIs construction, water supply and demand constraints; GIs construction constraints restrict the available construction areas for infiltration-based GIs, rainwater and stormwater harvesting systems within the urban region; Water supply constraints limit the available monthly water supply amounts for four types water sources - surface water, groundwater, rainwater and stormwater ; Water demand constraints set three types of monthly water requirements the UWM agent need to meet, that is, potable water, total water and urban irrigation demand. The details of the IGWM-OM are illustrated in Appendix A3.

In the ABM for UWM, UWB-SM and IGWM-OM are tightly coupled at the source code level, i.e., the subroutines of the IGWM-OM are embedded into relevant subroutines of UWB-SM. The primary data exchanged and shared between the two models are (1) water supply portfolios and GIs construction plan, and (2) hydrological conditions. Therefore, according to features of the model - some parameters of the IGWM-OM need to be computed by the UWB-SM, a simulation-based adaptive particle swarm optimization (S-APSO) is designed. Compared with particle initialization and evaluation for standard PSO, a coupling procedure proposed above for data exchange between IGWM-OM and UWB-SM is nested to make sure all particles feasible and measurable. Compared with the standard PSO updating mechanism, a Boltzmann selection operator and an evolutionary state-based parameter adaptation scheme are added to avoid premature convergence to improve algorithm performance; a simulation-based check & repair mechanism is introduced to guarantee all particles feasible during the particle updating process. These key features of the proposed S-APSO are explained in Appendix A4.

## 2.2 Multiagent system for IGWM at an inter-city scale

Fig. 1 (B) illustrates the hydrologic connection between the two urban areas. Such hydrologic connection among all UWM agents within the watershed, which means that all UWM agents have to share surface water resources with others, and IGWM activities of upstream agents may affect that of downstream. That is, an optimal decision-makings of city-scale from a UWM agent may not be suitable for the whole watershed economically and hydrologically because of negative externalities, which affect IGWM decision behavior of the other UWM agents being linked with it socially or hydrologically (Glendenning et al., 2012). Therefore, urban areas, and interconnected river networks can constitute a MAS-UWM (See Fig. 1 B). In the MAS-UWM, as shown in Fig. 1 (B), all UWM agents are linked with each other by a stream river network, and each UWM agent makes IGWM decisions - water supply portfolios and GIs construction - independently, only based on its urban hydrologic conditions as well as the upstream inflow, which is partly determined by the outflow of the upstream urban areas influenced by the IGWM decisions of the associated UWM agents. Therefore, the UWM-UWM agent interactions in watershed-scale IGWM can be depicted as a sequence of city-scale IGWM decisions along with river networks, which is a unique sequential decision





process as the upstream UWM agent's IGWM decision affects the decision of the subsequent downstream agent because the water is transported along the river; this interaction process can be regarded as the Markov property (Frydenberg, 1990).

To simulate the up-and downstream hydrologic interaction between UWM agent $i$ and $i+1$ in the associated river reach (See Fig. 1 B), a Muskingum-Cunge (M-C) routing equation is used to simulate changes in streamflow in the river reach connected with two adjacent urban areas (Garbrecht and Brunner, 1991; Weinmann and Laurenson, 1979). That is, taking the UWM agent $i+1$ in month $t$ as an example (See Fig. 1 B), its upstream inflow in a month can be expressed mathematical by outflow of the UWM agent $i$ in the month and the next month. In addition, it should be noted that the Muskingum-Cunge approach is also applicable in the case that there are branches in the main river reach (See UWM agents 1,2 and 3 in Fig. 1 B), which can be solved by dividing the river reach into several sub-reaches based on intersections of the main river and associated branches, and then calculate them in sequence. The details of the Muskingum-Cunge routing equation are illustrated in Appendix B2.

Therefore, the MAS-UWM can be formulated by the integration of the ABM-UWM (Eq. A10) with the Muskingum-Cunge routing model (Eq. B1), depending on its feature of the Markov property. A special type of multi-stage decision system is employed to model the MAS-UWM (Bellman, 1966) and the sequence of decisions-makings for each UWM agent - city-scale IGWM - relies on associated spatial locations along with the river networks, which is in order from upstream to downstream. The hydrologic variable - upstream inflow of each UWM agent - is considered the state variable to describe interactions between UWM agents. It can be written as follows,

$$
\begin{cases}
\text{ABM-UWM } i, & \forall i & (01) \\
\text{M-C routing equation } i \text{ in } t, & \forall i,t & (02) \\
q_{ri}^1(t) = Q_t^1, \text{ and } q_{ri}^i(0) = Q_0^i, & \forall i,t & (03)
\end{cases}
\tag{1}
$$

where the third row of Eq. (1) are initial conditions for the MAS-UWM, and $Q_t^1$ and $Q_0^i$ are the initial amounts of the upstream inflow for UWM agent 1 in month $t$ and UWM agent $i$ in month 0, respectively. The details of the MAS-UWM are illustrated in Appendix B3.

To solve the MAS-UWM, it is available to combine multiple S-APSO algorithms with the Muskingum-Cunge routing equation to simulate the dynamics of the MAS-UWM according to its Markovian property. That is, the optimal solutions for each UWM agent model are solved one by one in a specific order, which follows the sequence of the MAS-UWM via using the associated S-APSO. Notice in particular that the monthly outflow amounts for the optimal solution of each UWM agent model needs to be recorded during the S-APSO search process. They, as an input of the relevant Muskingum-Cunge equation, are used to calculate the monthly upstream inflow amounts in the associated downstream reach - an input data for the adjacent UWM agent model. In this way, the multi-S-APSO framework for simulation of the interactions of the MAS-UWM is developed. The details of the solution approach for MAS-UWM are illustrated in Appendix B4.

## 2.3  Bi-level multiagent system for IGWM at a watershed scale

In the MAS-UWM, each UWM agent minimizes its own IGWM costs without direct regard for the external effects on the other UWM agents' interests because of the Markov property of the system. It may appear to be inequalities of water resources sharing where upstream water users might easily take up more surface water than downstream users (Giuliani and Castelletti,





2013), which might increase IGWM costs of downstream agents. For a watershed manager, of interest is the ability of the water policy to address the inequality and encourage more sustainable water use in watershed-scale IGWM (See Fig. 1 C). Thus, we

develop a ABM-WM and build a bi-level multiagent system via combining the ABM-WM with the MAS-UWM, which can simulate water policy-induced interactions between MW and UWM agents.

To achieve equitable surface water resources allocation in the MAS-UWM, the WM specifies limits for surface and ground-water withdrawal - prescribing the minimum storage levels of the river for each urban areas, and enacts a streamflow penalty strategy to regulate UWM agents' IGWM decision behavior - setting a series of low flow thresholds at each checkpoint; if

the UWM has out-of-threshold surface water withdrawal, then penalties are imposed. In the study, an agent-based model for watershed manager (ABM-WM) is built to describe how a WM set a watershed management policy - a streamflow penalty strategy - to regulate all UWM agents' behaviors of IGWM in a watershed. That is, a WM limits water abstraction decisions of each UWM agent in the period via prescribing a series of low streamflow thresholds in associated hydrological stations based on hydrologic conditions; If streamflow in outlet for an urban area is below its threshold, a penalty fee will be imposed on the

UWM agent. The strategy, in theory, can force UWM agents to recognize one or more of the externalities caused by IGWM, thereby adjusting their IGWM decisions because it can, to some extent, determine the cost of IGWM (Baumol et al., 1988). The WM can share fair water resources among urban areas in a watershed by setting a rational streamflow penalty strategy that affects all UWMs' decisions. The details of the ABM-WM are illustrated in Appendix C2.

Under the policy intervention from a WM, each UWM agent needs to make reasonable IGWM decisions to trade off the

290 previous three types of costs (i.e., GIs construction, water supply, wastewater drainage) and the possible penalty fee set by a WM to minimize their own total IGWM costs under the specified low streamflow thresholds. Therefore, the above ABM-UWM will be extended - its annual IGWM cost function is converted as the sum of GIs construction, water supply, wastewater drainage costs, and penalty fees. The details of the model are shown in Appendix C3.

As mentioned above, a streamflow penalty strategy prescribed by a WM agent might change some UWM agents' decisions

of IGWM - upstream UWM agents might have to adjust their IGWM decisions to increase outflow to avoid over high penalty fees for costs minimization, which is beneficial to the downstream agents. Such changes in UWM agents' behavior can, to some extent, shift the interactions in the MAS-UWM, which might have a potential impact on the watershed environment that can be measured by the assessment index (i.e., water allocation Gini coefficient) set by the MW (See Fig. 1 C). Therefore, a WM agent can assess the effects of the policy on the watershed via checking the given index that reflects feedbacks of the

MAS-UWM and then gradually adjusts it to find the optimal one. This process is a WM-UWM agent interaction in watershed-scale IGWM under a water policy. Fig. 1 (C) illustrates that the WM-UWM agent interaction is no longer determined only by the WM or the UWMs, and both of them try to optimize their objectives (i.e., equity vs. cost objectives for WM and UWMs) under the associated constraints (i.e., steamflow vs. GI construction, water supply and demand constraints) and reactions of the other party. Therefore, they follow a specific decision rule. That is, the WM agent first makes a decision, and then each UWM

agent specifies a decision to optimize their own objectives with full knowledge of the WM's decision; the WM also optimizes its own objective based on the rational UWMs' reactions. In economic theory, this process - the WM-UWM agent interaction - is a Stackelberg game (Von Stackelberg, 2010).





Based on the features of the WM-UWM agent interaction, it can follow a hierarchical decision rule for the leader - the WM agent and the multiple followers - the UWM agents (Dempe, 2002). Besides, for the followers, the UWM agents form a MAS-UWM (Eq. 1) that has a Markov property, which involves a special multi-stage decision-making process (Bellman, 1966). By integrating the ABM-UWM, the ABM-WM and the MAS-UWM (Eq. B1) mentioned above, a BL-MAS for IGWM at a watershed scale can be developed to describe the Stackelberg game between the WM and multiple UWM agents, and unique multi-stage system constructed to reflect the state transitions for the multiple WM-UWM agents, which can be formulated as follows:

$$
\begin{cases}
\text{ABM-WM;} \\
\text{where } W_i, GI_i \text{ solves} \\
\quad \begin{cases}
\text{ABM-UWM } i, & \forall i \\
\text{M-C routing equation } i \text{ in } t, & \forall i, t \\
q_{ri}^1(t) = Q_t^1, \text{ and } q_{ri}^i(0) = Q_0^i. & \forall i, t
\end{cases}
\end{cases}
\tag{2}
$$

where $[-]_*$ represents that the parameter is from simulating calculation of the UWB-SM. $W_i$ and $GI_i$ represent the decision variables of water supply portfolios and GIs construction for UWM agent $i$. The details of the BL-MAS are illustrated in Appendix C4.

To solve the BL-MAS (Eq. C6), the proposed S-APSO is also applied in the BL-MAS, only the fitness function for particles needs to be adjusted. Besides, the above multi-S-APSO framework is also available in simulating the interactions among all UWM agents in the BL-MAS under a given streamflow penalty strategy because of the features of its hydrologic connections - Markovian property. For the ABM-WM, the above S-APSO framework without the simulation-based initialization and the check & repair mechanism is available to look for the optimal solution due to its simple constraint conditions. However, there is a critical factor in the simulation of the BL-MAS that is how to deal with the special decision rule between the WM and the UWN agents - a Stackelberg game, i.e., the WM agents' best response is based on the associated reactions of all UWM agents (Von Stackelberg, 2010). In fact, it is challenging to obtain a Stackelberg solution to the BL-MAS using general solution methods because the bi-level model is an NP-hard problem, even in its simplest linear case (Dempe, 2002). To deal with the specific bi-level model decision rules, the study nests the multi-S-APSO framework for the MAS-UWM mentioned before into the particle performance measurement of the S-APSO for the WM agent, which can simulate the responses of the MAS-UWM to a given streamflow penalty strategy prescribed by the WM agent, thereby assessing the policies' effects accurately. By the nested structure, therefore, a nested S-APSO framework is proposed for searching for the optimal WM-UWN interactions in the BL-MAS under a streamflow penalty strategy. The details of the nested S-APSO is shown in Appendix C5.





## 3   Case study and experimental design

In this section, the proposed multiagent socio-hydrologic framework is utilized in a case study on the Minneapolis-La Crosse section of the Upper Mississippi River, United States, and three numerical experiments are designed to demonstrate its effectiveness and efficiency and to characterize the decision-making of IGWM at a city-and watershed scale.

### 3.1   Overview of the study area

As Fig. 2 shows, the Upper Mississippi River basin ranges in latitude from $47^o$ N to $37^o$ N, and its flows roughly 2,092 $km$, from Lake Itasca (northern Minnesota) to the Ohio River (southern Illinois), which covers seven states of the US, such as Illinois, Iowa, Minnesota and Wisconsin, and has a watershed area of 489,508 $km^2$. The main river and its tributaries have an average annual discharge of 3,576 $m^3/s$, which has three high- (late April, late June, and October) and two low-flow (midsummer and late winter) periods as a result of varied rain and snow conditions (Baldwin and Lall, 1999).

In the watershed, more than 70% of the area is used for agriculture and animal husbandry. Only 5% of the site has been converted to urban areas. However, it has a population of about 24 million, especially in the metropolitan high-density regions ($> 100,000$ $people/km^2$), such as Minneapolis-St. Paul, Minnesota and La Crosse, Wisconsin. It is estimated that over 5.3 million $m^3$ of water are withdrawn from the Upper Mississippi River each day in the 60 counties for municipal and public supplies. Although the basin is water-rich, water sharing among riparian urban areas is still a concern due to the high water demands for the environment and agriculture, the strict water level regulations for navigation, and the high-density urbanization. Groundwater is also a crucial source of water that is widely used by the partial urban areas, and green infrastructures are continually encouraged and expanded to manage urban stormwater by the local communities and authorities (Askew-Merwin, 2020). Therefore, it is appropriate to analyze the watershed-scale IGWM in the Upper Mississippi River basin.

In this study, the Minneapolis-La Crosse section - approximately 236 km long - of the Upper Mississippi River basin, a high-density urban area, is considered as the study area. (See Fig. 2). Notice that the study only focuses on the urban water use and allocation, which perhaps is a small percentage of the basin water resource in the study region. Fig. 2 indicates that there are nine main riparian urban areas along the section; i.e., i = 1, 2,..., 9. Some are metropolises with a large population for these urban areas, such as Minneapolis, St. Paul, and La Crosse. The others are city groups that consist of multiple small cities, such as Red Wing and Bay City. In short, the basic features of urban areas in the study site is shown in Tab. 1.

### 3.2   Experimental design

Given the background of the study area, all urban area - metropolises or city groups - is assumed as a UWM agent that makes city-scale IGWM decisions individually, which can be formulated by Eq. (A10) and the Upper Mississippi River Basin Association (UMRBA) is regarded as the WM agent that regulates these urban areas, and the associated interactions among UWMs and between WM and UWMs are formulated by Eq. (B2) and (C6), respectively. In the case study, we examine the potential properties of IGWM at three spatial scales through observing and comparing the results of the WM and UWM







**Figure 2.** Schematic diagram of the study area. Notice that the base map and metropolis and city group maps in the bottom right of the figure are from Esri (2012) and U.S. Census Bureau (2018), respectively; the US states boundary map and the Upper Mississippi River Basin map in the bottom left of the figure are from U.S. Census Bureau (2018) and U.S. Geological Survey (2021), respectively.

agents' behaviors and their interactions in IGWM simulated by the proposed model under different socio-hydrologic settings. Therefore, three numerical experiments to IGWM at city, inter-city and watershed-scales are designed in the following:

**Experiment 1 of IGWM at a city scale:** The experiment is designed to identify and classify the behavioral characteristics of city-scale IGWM for UWM under different hydroclimatic settings, thereby investigating how watershed hydroclimatic circumstances affect the decision-making of city-scale IGWM. In this experience, a sensitivity analysis to the ABM-UWM



**Table 1.** The basic features of urban areas in the study site.

| No. of urban area | Urban names | Urban types | Population | Urban area $A_u$ $(km^2)$ | Ratio of impervious surface area (%) |
|---|---|---|---|---|---|
| 1 | Minneapolis city; | metropolis | 2,914,866 | 139.86 | 65 |
| 2 | St. Paul, West St. Paul, Mendota Heights, and Lilydale; | metropolis | 350,167 | 172.26 | 63 |
| 3 | South St. Paul, InverGrove Hights, Rosemount, Newport, St. Paul Park, Gottage Grove, and Woodbury; | city group | 230,277 | 367.41 | 60 |
| 4 | Hastings and Prescott; | city group | 71,870 | 33.37 | 61 |
| 5 | Red Wing and Bay City; | city group | 49,167 | 91.75 | 64 |
| 6 | Lake City and Stockholm; | city group | 12,405 | 51.40 | 62 |
| 7 | Wabasha, Alma, and Nelson; | city group | 12,265 | 38.48 | 61 |
| 8 | Winona and Fountain City; | city group | 26,757 | 60.62 | 59 |
| 9 | La Crosse, LaCrescent, French Island, and Onalaska; | metropolis | 80,601 | 96.93 | 66 |

is performed. All urban areas within the studied region are considered as study objects, and the associated ABM-UWMs are run many times under different combinations of upstream inflow and precipitation, which are the model's input parameters, to

calculate associated optimal decision-makings of city-scale IGWM. The combined effects of upstream inflow and precipitation on the city-scale IGWM in the study area is discussed to characterize the city-scale IGWM patterns of UWMs in changing environments. The baselines of these two hydroclimatic parameters are obtained from the USGS, and six situations are set via decreasing and increasing the baselines by 25%, 50%, and 75%, respectively. A k-means clustering method (Likas et al., 2003) is employed to classify all UWMs' decision-making of IGWM - water supply portfolios and GIs construction - under mixed

hydroclimatic conditions. By categorizing the responses of UWMs to different environments, we summarized the similarities of UWMs' decision behavior, which can be used to indicate the features of city-scale IGWM patterns of UWMs.

**Experiment 2 of IGWM at an inter-city scale:** The experiment is set for examining how a UWM agent affects other UWM agents in inter-city scale IGWM and assessing the impacts of the MAS-UWM interactions on the watershed. In this experience, two scenarios are simulated by the MAS-UWM model. The first scenario, as an experiment group, is conducted taking GIs

construction decisions into account, while the second one, as a control group, is configured without GIs development via setting $r_{imax}, r_{rmax}$ and $r_{smax} = 0$ in the UWM agent model. By comparing the results of the two groups, we identify the possible impacts of the IGWM decisions in upstream urban areas on the downstream urban areas and quantify the relative attribution of GI in the impacts. Also, we conduct a sensitivity analysis to the experiment group to analyze the effects of different watershed settings on the UWM-UWM interactions economically. There are various combinations of watershed settings (i.e., watershed

upstream inflow, the precipitation, urban water demands) to be considered. The scenario of the MAS with GIs was set as the baseline in the experiment. Similar to Experiment 1, monthly precipitation and watershed upstream inflow were proportionally changed from 25% to 175% based on the baselines, to simulate the hydroclimatic watershed dynamics; monthly urban water





demands, including the indoor and outdoor demands, were used to represent the socioeconomic changes of the watershed, which were appropriately shifted from 25% to 175% according to the baseline.

**Experiment 3 of IGWM at a watershed scale:** This experiment will be devised to investigate the influence of the streamflow penalty strategy set by a WM agent on IGWM at a watershed scale. A policy simulation to the BL-MAS is conducted. According to the features of the BL-MAS - Stackelberg game between the WM and the UWM agents, the optimal solutions of the BL-MAS is defined as Stackelberg equilibrium points (Dempe, 2002) - equilibrium between the WM and UWMs in the watershed-scale IGWM that none of them have an incentive to alter their decisions. In theory, there are likely to be multiple

equilibrium points in a bi-level system (Dempe, 2002). Therefore, all equilibrium of the BL-MAS need to be identified and analyzed to assess the possible effects of the water policies prescribed by the WM on the UWM agents' decision behavior of the watershed-scale IGWM. In the experiment, the baseline penalty rate is specified as 0.005 $/m^3$. Notice that the penalty rate is artificial due to no penalty strategy to the study area in reality. We have to choose a reasonable rate that might affect all UWMs' decision behavior in the study area through a comprehensive parameter analysis (Parsapour-Moghaddam et al.,

2015; Rosegrant et al., 2000). By comparing with the results of Experiment 2, we evaluate the water policy-induced changes in the interactions in the MAS-UWM and the watershed hydrologic regime. Furthermore, we summarize the effect of the optimal streamflow penalty strategy on watershed-scale IGWM in changing watershed institutional and hydroclimatic conditions through conducting a sensitivity analysis under different parameters mixture of the penalty rate, the watershed upstream inflow, and the precipitation.

### 3.3   Data collection and processing

In the proposed framework to the case study, some parameters of the ABM-UWM and ABM-WM, as shown in Tab. A2, A3 and A7 in Appendix A1 and C1, need to be determined by collecting, processing and estimating actual data from diverse sources.

    For the parameters of UWB-SM, the urban water demand data ($36 \times 9$) were measured based on the associated urban populations and layouts via the Last (2011) method. The hydroclimatic data ($27 \times 9$) were obtained from the USGS (https:

//waterdata.usgs.gov/nwis/rt) and the NOAA (https://www.ncdc.noaa.gov/cdo-web/) databases. The urban area data ($1 \times 9$) were obtained from the U.S. Census Bureau (http://www.census.gov/), and the urban land features data ($4 \times 9$) were calculated through analyzing the remote sensing images (Last, 2011). Also, we used different databases and methods to obtain the urban depth-related data. To be specific, the mean depths of the urban aquifer at the low topographic point ($1 \times 9$) were assumed as ten meters plus the mean depth of the riverbed obtained from the National Elevation Dataset (NED, http://ned.usgs.gov),

because of lack of relevant data about the aquifer. Moreover, they were also used to calculate the associated aquifer mean depth at the high point ($1 \times 9$) by using a linear fitting method to urban hypsometric curves (Sharma et al., 2013). Notice that we believe that the rough assumptions might have little impact on the results of the three experiments because the study area is within a water-rich watershed, and the surface water is the primary source of urban water supply. The mean depths of wells for groundwater withdrawal ($1 \times 9$) were measured via averaging adjacent wells' depths from the USGS database. Besides, the

maximum ratios of the constructed areas of rainwater, stormwater harvesting systems, and infiltration-based GIs to the relevant surface ($3 \times 9$) were set as 50%. Notice that we want to test the maximum potential for urban rainfall utilization in the study





area. However, the actual ratios of GIs areas might be much smaller than these ratios. All of the mean depths of the shallow soil layer ($1 \times 9$) and the wastewater pipe networks ($1 \times 9$) were set as $3\ m$ and $2\ m$ based on similar settings (Frost et al., 2016). The mean effective porosity ($1 \times 9$) was estimated as 10% based on the Prior et al. (1953) report due to lack of related accurate

information.

For the parameters of IGWM-OM, three types of GIs construction cost data ($3 \times 9$) were estimated based on the relevant GIs cost databases in the EPA websites (https://www.epa.gov/) through the Houle et al. (2013) approach. And the associated cost scaling coefficients ($3 \times 9$) were obtained using a linear fitting method to these cost data. Some parameters related to the cost of surface, groundwater supply, and wastewater drainage ($8 \times 9$) and the urban water supply capacities ($2 \times 9$) were

collected, processed, and calculated from various open materials on the websites of the associated water agencies. According to the existing hydrologic regimes and water use framework in the study sites, there are various other water users with different purposes in addition to the chosen urban areas to share water resources, such as agricultural irrigation, commercial navigation, and ecological conservation. Therefore, we set the minimum storage levels for surface and groundwater withdrawals ($24 \times 9$) based on the various materials and databases related to the water level regulation, water use, and streamflow. In addition, the

minimum and maximum historical streamflow ($24 \times 9$) - the parameters of the WM agent - were obtained from the USGS.

### 3.4 Model and algorithm setup

In the framework, the other parameters of the UWB-SM and the Muskingum-Cunge routing model were evaluated by model calibration to hydrologic data (See Tab. A4 and B1 in Appendix A1 and B1). For the calibration parameters of UWB-SM ($13 \times 9$), they were obtained by calibrating UWB-SM against the monthly outflow of an urban area, which is estimated. Notice

that the outflow of an urban area was estimated via using a map correlation method (Archfield and Vogel, 2010) to available monthly streamflow observations from the associated USGS stations in the study system because of the location difference between the urban area's outlet and the associated USGS stations. In this study, time series of observed and simulated outflow at each urban area are analyzed for both calibration (1996-2020) and validation (1971-1995) periods. In the process of the UWB-SM calibration, the GI construction and the relevant rainwater and stormwater supply decisions are not considered,

and monthly surface and groundwater withdrawals are set as actual water use data. Model calibration is performed using the proposed S-APSO framework to solve a specified single objective optimization - maximization of the modified Kling-Gupta Efficiency (He, 2019). Similar to the UWB-SM, the parameters of the Muskingum-Cunge equations were also calibrated based on the estimated monthly outflows.

Furthermore, to guarantee the proposed algorithm's effectiveness, there are also algorithm parameters to be calibrated in

this case. For the S-APSO, only population size and iteration number need to be determined due to the use of the adaptive parameter scheme. They were calibrated based on the results of a trial-and-error algorithm parameter calibration procedure that was carried out to observe the behavior of the algorithm at different parameter settings (Beielstein et al., 2002). Therefore, the calibrated parameters for the S-APSO were selected as follows: the population size and the iteration number are set as 50 and 200, respectively. Three numerical experiments mentioned above were conducted and run on an 8-Core Intel Core i7, 3.8

GHz clock pulse with 40 GB of DDR4 memory, 2,667 MHz.





## 4 Results and discussion

The results of these experiments and the associated analysis are discussed as follows.

### 4.1 Characteristics of city-scale IGWM in changing environments

The classified results of UWMs' decisions of IGWM to different environments are illustrated in Fig. 3 (a). As Fig. 3 (a) shows,
there are four IGWM patterns for UWMs in response to the different combinations of upstream inflow and rainfall inputs. The
light blue dots region is defined as pattern 1, which represents the similar reactions of UWMs in the decision-making of IGWM
to the high upstream inflow and rainfall inputs. The green dots region indicates pattern 2, indicating the consistent behavior of
IGWM for UWMs to the low upstream inflow and high precipitation settings. Similarly, patterns 3 and 4 are set in the dark blue
and yellow dots region, which denotes the analogous decision-makings of IGWM for UWMs in the study site under high (low)
inflow and low (low) rainfall conditions respectively. The result indicates the homogeneous behavior of city-scale IGWM for
all UWMs in the relatively extreme hydroclimatic conditions in the study region. This phenomenon might be that most urban
areas in the study area have similar costs of accessing the four types of water sources, mainly because they are located in the
same states, Minnesota and Wisconsin. Economically, the similar costs of water resources lead to similar responses of UWMs
in IGWM under given hydrologic conditions (Loucks and Van Beek, 2017). In addition, the grey dots region is a transition area
in which UWMs may make different IGWM decisions to minimize water use costs under some hydroclimatic settings. This
might be because of the physical differences among urban areas in the study area, such as impervious ratios, locations, urban
landscape, and water demands, which affect available amounts of different water resources, thereby changing IGWM decisions
under some specific environments.

The characteristics of the four IGWM patterns mentioned above are shown in Fig 3 (b) - (f). The ratios of water supply
portfolios and GIs construction in the four patterns are illustrated in Fig 3 (b) - (d). There are large distinctions in the IGWM
between the four patterns; In pattern 1, centralized and decentralized water account for 52.9% and 47.1% of total water supply,
respectively, which means that stormwater and rainwater are widely utilized. More than 80.0% of centralized water supply is
from surface water. In the aspects of GI construction, over three-fifths of available urban areas are used to develop stormwater
and rainwater harvesting systems, which accounts for 96.2% of total GIs construction areas. These results show the features
of IGWM to the high upstream inflow and rainfall inputs - UWMs prefer to use stormwater and rainwater directly to meet
urban non-potable water demand by stormwater and rainwater harvesting systems for the sake of cost, and to supply surface
water to meet potable water demand. In pattern 2, similar to pattern 1, stormwater and rainwater are also heavily used directly
to satisfy non-potable demand. At the same time, groundwater is the main potable water resource due to the low upstream
inflow input. Accordingly, over 75% of available areas are utilized for GIs development, and 35.9% of which, unlike pattern
1, is for infiltration-based GIs for groundwater recharge. In pattern 3, stormwater and rainwater are hardly used (only 2.9% of
total water supply), and the construction of the associated GIs (only 3.9% of available urban areas) is also limited due to the
scarcity of precipitation. The surface water resource is dominant in urban water supply, accounting for 87.2% of total water
withdrawals because of the high upstream flow inputs. In comparison, stormwater and rainwater are partially used (22.2% of







**Figure 3.** Characteristics of four city-scale IGWM in changing environments.

**Note:** · In Fig. 3 (e), *Ratio of system water output to input* is used to measure the overall balance in an urban area, which is calculated as; the system water output divided by the system water input, where system water output is equal to the sum of monthly evapotranspiration, water consumption and outflow, and system water input is the sum of monthly upstream inflow and rainfall; *Ratio of rainfall harvesting* is set as the ratio of the sum of stormwater and rainwater supply to total rainfall.

· In Fig. 3 (f), *Ratio of water cycle* is set as the ratio of the total amounts of water supply to the total amounts of urban stored water in a year, where the total amounts of water supply is set as the sum of the amounts of surface water, groundwater, stormwater and rainwater supply, and the total amounts of urban stored water is defined as the sum of the monthly amounts of water storage of seven well-defined water storage units - roof, other surfaces, rainwater, stormwater, soil layer, aquifer and river; *Ratio of change in stored water* is calculated as follows; the difference between the amounts of urban stored water in the last and first months divided by that in the first month, where the amounts of urban stored water in the first and last months are equal to the sum of the amounts of water storage of seven well-defined water storage units in the first and last months.





total water supply) directly or indirectly by constructing GIs at a moderate level (41.2% of available urban areas) in pattern 4. This might be because UWMs have to collect and use all kinds of available water resources as much as possible when system water inputs - rainfall and upstream inflow - are low. Hence, the groundwater is supplied more extensively for maintaining water supply steadily. It is worth mentioning that infiltration-based GIs, which account for 16% of total GI construction areas, are, to a great extent, built to enhance aquifer recharge for reducing the costs of groundwater abstraction. Besides, Fig 3 (d) also illustrates the mean IGWM costs per unit of water in the four patterns. Undoubtedly, the costs of the pattern under high system water inputs conditions are lower than those under low water inputs.

Also, we investigate the impacts of the four IGWM patterns on the urban hydrologic regime. Fig. 3 (e) and (d) illustrates the four indexes used to measure urban water cycle defined by Kenway et al. (2011) under these patterns. As Fig. 3 (e) shows, the ratios of system water output to input and rainfall harvesting vary from pattern to pattern. Except pattern 4, the ratio of system water output to input of the rest of the patterns are smaller than one, which means that the relevant IGWM patterns can increase water storage in an urban system, especially the pattern 1 and 2 with relatively large areas of GIs - 17.5% and 12.6% of water inputs are stored, respectively, which, to some extent, decrease the outflow of the urban catchment. The result is consistent with the Glendenning et al. (2012) results. However, in contrast to the other patterns, the ratio of system water output to input of the pattern 4 indicates the decreases in urban water storage. This might be because of the over-exploitation of groundwater under low upstream inflow and precipitation conditions. In this circumstance, to build a small range of infiltration-based GIs (i.e., The ratio of rainfall harvesting = 17.6%) to increase groundwater recharge is limited. Besides, as shown in Fig. 3 (f), the ratios of water cycle increase orderly from the pattern 1 to 4, and the associated ratios of change in stored water have an opposite shift. These results also are consistent with the above results in Fig. 3 (e).

All in all, these results show the interactions and feedbacks between the urban areas and the corresponding hydroclimatic settings in city-scale IGWM. Specifically, the hydroclimatic environment determines the available amounts of the four types of water sources, thereby affecting UWMs' selections of IGWM patterns from an economic perspective. Also, IGWM patterns chosen by UWMs can change urban hydrologic regimes, which can, in turn, shifts the watershed environment. In addition, these results demonstrate that the proposed model can simulate the UWMs' decision-making of IGWM in changing settings.

## 4.2 Characteristics of inter-city-scale IGWM in changing environments

The results of the inter-city-scale IGWM in the two cases are shown in Tab. 2. The ratios of water supply portfolios of all urban areas in the two scenarios are shown in the 2nd-5th and 9th-10th rows of Tab. 2. In the case without GIs - a control group, surface water use fractions gradually decrease from the upstream to the downstream areas. For example, the ratios of surface water in the urban area 1, 5, and 9 are 0.85, 0.79, and 0.73, respectively. There is a trend that the available amounts of surface water gradually decrease along with the river (See the blue line in Fig. 4 a). These results indicate the adverse impacts of upstream water users on the downstream water users in surface water withdrawals. That is, water extraction from a stream in the upstream areas reduces the available amounts of surface water in the downstream regions, which would increase the costs of surface water withdrawal for the downstream urban areas, thereby forcing them to substitute other water resources, such as groundwater. In contrast, in the case with GIs - an experiment group, the fractions of surface water use appear to be





independent of the study area locations due to stormwater and rainwater use via GIs. However, it might aggravate the adverse

impacts - up-and downstream imbalance of available surface water. For example, the ratios of surface water use markedly

decrease in the downstream urban areas, especially in urban areas 6, 8, and 9, which have adopted the IGWM pattern 4 - the

high ratio of groundwater use. This may be because the upstream UWMs prefer to use stormwater and rainwater resources

through developing GIs, such as the IGWM pattern 1 and 2, to minimize the costs of water use (Cooley et al., 2019). However,

as mentioned before, the IGWM pattern 1 and 2 can reduce the outflow of urban subcatchment (See Fig. 3 e), which might

worsen the decrease of available surface water in the downstream region (See the red line in Fig. 4 a). Besides, the 8th and

12th rows of Tab. 2 also shows the Gini coefficients (Eq. C3a) in the two scenarios - 0.0129 and 0.0169, indicating that the GIs

construction to use stormwater and rainwater intensifies the imbalance in water use in the study area.

**Table 2.** Results of the inter-city-scale IGWM in the two scenarios

| 01 | Scenarios | Urban Area $i$ | | $i = 1$ | $i = 2$ | $i = 3$ | $i = 4$ | $i = 5$ | $i = 6$ | $i = 7$ | $i = 8$ | $i = 9$ |
|---|---|---|---|---|---|---|---|---|---|---|---|---|
| 02 | | Ratio of water supply portfolios | Surface water | 0.39 | 0.81 | 0.38 | 0.72 | 0.11 | 0.09 | 0.55 | 0.13 | 0.12 |
| 03 | | | Groundwater | 0.13 | 0.14 | 0.12 | 0.17 | 0.46 | 0.68 | 0.27 | 0.66 | 0.63 |
| 04 | MAS with GIs | | Stormwater | 0.28 | 0.01 | 0.32 | 0.09 | 0.25 | 0.15 | 0.11 | 0.13 | 0.18 |
| 05 | | | Rainwater | 0.19 | 0.04 | 0.19 | 0.05 | 0.19 | 0.08 | 0.08 | 0.08 | 0.07 |
| 06 | | Water use pattern | | P1 | P3 | P1 | P3 | P2 | P4 | P3 | P4 | P4 |
| 07 | | Mean unit water costs | | 2.89 | 2.91 | 2.96 | 3.07 | 3.03 | 3.11 | 3.09 | 3.15 | 3.12 |
| 08 | | Gini coefficient | | | | | | 0.0129 | | | | |
| 09 | | Ratio of water supply portfolios | Surface water | 0.85 | 0.87 | 0.83 | 0.81 | 0.79 | 0.70 | 0.75 | 0.72 | 0.73 |
| 10 | MAS without GIs | | Groundwater | 0.15 | 0.13 | 0.17 | 0.19 | 0.22 | 0.30 | 0.25 | 0.28 | 0.27 |
| 11 | | Mean unit water costs | | 2.98 | 2.96 | 3.03 | 3.10 | 3.09 | 3.14 | 3.11 | 3.18 | 3.14 |
| 12 | | Gini coefficient | | | | | | 0.0169 | | | | |

**Note:** · *Ratio of water supply portfolios* for urban area $i$ is calculated as follows; The amounts of a type of water source supply divided by the total amounts of water supply in the urban area $i$.

· *Mean unit water costs* for urban area $i$ is obtained as follows; The total IGWM costs divided by the total amounts of water supply in the urban area $i$.

The 7th and 11th rows of Tab. 2 illustrate the IGWM cost per unit of water for each urban area in the two scenarios. There

is a trend that the unit water costs constantly increase from the upstream to the downstream region, indicating the adverse

impacts of the up-and downstream imbalance of the surface water on the cost of water use for the downstream urban areas. In

comparison, the costs of water use in the MAS with GIs are smaller than those without no GIs for all urban areas. In contrast,

the differences in these costs between the two scenarios continuously decrease from the urban area 1 to 9 (See Fig. 4 b). The

reasons behind these results might be that there are two main factors - upstream inflow and GIs - affecting the costs of water

use in the MAS-UWM, especially for the downstream UWM agents. In general, the upstream inflow reduction can decrease

the amounts of surface water, thereby increasing the cost of water use for the downstream UWMs. On the contrary, rainwater

and stormwater use via GIs can increase the available amounts of water resources and decrease water supply costs. In the

MAS-UWM, especially for the downstream UWM agents, the cumulative effect of the upstream IGWM decision behavior on

streamflow gradually amplifies along the river. In contrast, the impact of the GIs on rainfall resource use generally remains





stable due to the limitations of climatic, physical, socioeconomic conditions. Therefore, the combination of the two effects might lead to a gradual decrease in the impact of GIs on the cost of water use along with the river.

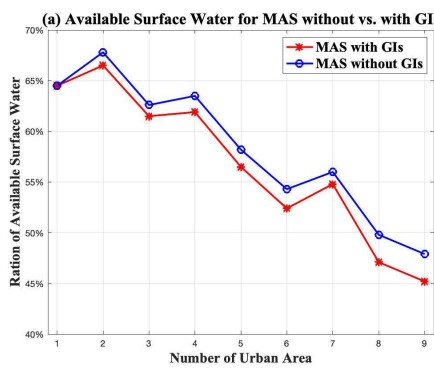
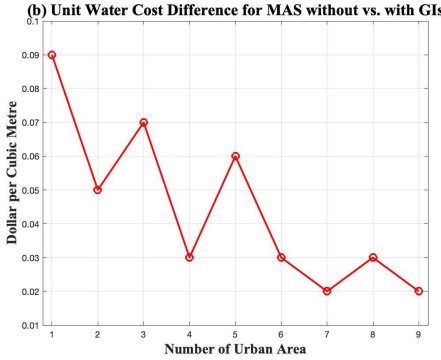

**Figure 4.** Available surface water and unit water cost difference in the two scenarios

**Note:** · In Fig. 4 (a), *Ratio of available surface water* for urban area is set as the average of the ratio of the available monthly storage levels of river for water withdraw to the maximum theoretical that levels in an urban area.

· In Fig. 4 (b), *Unit water cost difference* for urban area calculated as follows; The difference of the mean unit IGWM costs in the urban area in the scenarios of the MAS without and with GIs.

Next, we explored the influences of various social and hydroclimatic settings on the dynamics of the MAS-UWM in a watershed-scale IGWM. A sensitivity analysis of the MAS-UWM model in the study case was run. The mean unit costs and the Gini coefficients in the MAS-UWM under different precipitation, upstream inflow, and urban water demand inputs are illustrated in Fig. 5 (a) and (b).

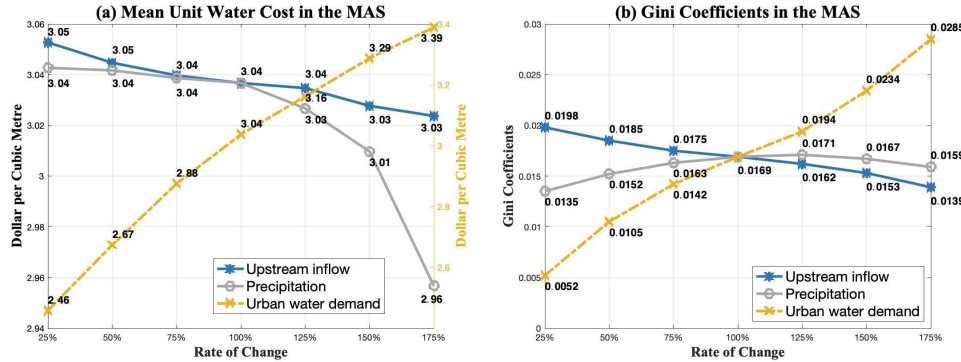

**Figure 5.** Changes in unit water costs and Gini coefficients under different social and hydroclimatic settings.

· **Note:** In Fig. 5 (a), *Mean unit water costs* in the MAS is obtained as; The total IGWM cost divided by the total amounts of water supply in the MAS.

In Fig. 5 (a), the blue and grey lines represent the changes in the mean costs per unit of water use in the MAS-UWM under

different precipitation and upstream inflow inputs. It shows a non-linear inverse relationship between the cost of water use and





the system water inputs - the mean cost of water use in a watershed-scale IGWM decreases with the increases in both watershed upstream inflow and rainfall. Notice that the cost of water use in the MAS-UWM is more sensitive to the precipitation than upstream inflow. Notably, the mean cost of water use decreases from 3.01 $/m^3$ to 2.96 $/m^3$ when the rainfall increases from 150% to 175%. It appears that rainfall inputs, when it goes beyond a certain threshold, might have a profound effect on the

IGWM cost of the MAS-UWM. This might be due to that more and more UWMs can switch IGWM patterns from Pattern 3 or Pattern 4 to Pattern 1 or Pattern 2, with the available rainfall exceeding a given threshold, leading to a significant decline in water use costs. These results are consistent with the results in Fig. 3 (d). In comparison, as the yellow line in Fig. 5 (a) shows, the cost of water use in the MAS-UWM is highly sensitive to the urban water demands. The increases in urban water demands are equivalent to the decreases in available water resources within an urban area, which greatly adds to the costs of water use.

Fig. 5 (b) shows the changes in the equity level of water use in the MAS-UWM under different hydroclimatic and socioeconomic settings. The blue and grey lines in Fig. 5 (b) represent the trends of Gini coefficients in the MAS-UWM under mixed precipitation and upstream inflow conditions - the Gini coefficient curve tends to proportionally decrease as the upstream inflow increases, while there is an inverted U-shaped curve for the Gini coefficient when the precipitation increases; it reached a peak (0.0171) when the ratio of the rainfall to the baseline is equal to 125%. On the side of the upstream inflow, these re-

sults show that the reduction in watershed upstream inflow harms the equity levels of water use in watershed-scale IGWM. The possible reason is that the Markov property of the MAS-UWM has a cumulative effect on the reduction in surface water availability along with the river. It can, to some extent, amplify the surface water conflicts between the up-and downstream urban areas as the watershed upstream inflow decreases. On the side of the precipitation, the impact of rainfall on the equity of water use in the MAS-UWM is more complicated. The reasons for the increase in the Gini coefficient as the precipitation

increase slightly might be that the water use costs for the upstream UWMs decrease remarkably by substituting stormwater and rainwater for surface and groundwater water as the rainfall increases. In contrast, in the downstream regions, the reduction in the relevant costs is limited because the cumulative effect of upstream inflow mentioned above limits the intention of the corresponding UWMs to switch IGWM patterns to use more rainfall resources economically. However, as the precipitation increases significantly, the abundant rainfall encourages the downstream UWMs to use stormwater and rainwater via GIs cost-

effectively, reducing water use costs sharply, thereby making the watershed-scale IGWM more equitable. In comparison, the Gini coefficient in the MAS-UWM is more sensitive to upstream inflow than rainfall. This is because the cumulative effect of upstream inflow might offset the impact of rainfall due to the Markov property of the MAS-UWM.

In short, these results illustrate the impact of different social and hydrologic settings on IGWM at watershed scale; Hydrologically, the change in rainfall is more likely to influence the cost of water use in the study basin due to the relatively low cost

of stormwater and rainwater resources. Yet, the shift in watershed upstream inflow easily affects the equity level of water use in the MAS-UWM because of the Markov property of the MAS-UWM. Socially, the change in the urban water demands has a profound effect on the cost and equity of water use in MAS-UWM because it is equivalent to the combined effects of rainfall and upstream inflow.





### 4.3 Impacts of water policy on watershed-scale IGWM

The results of the policy simulation to the BL-MAS are shown in Tab. 3.

**Table 3.** Results of watershed-scale IGWM in the BL-MAS.

| 01 | Equilibriums | Urban Area $i$ | | $i=1$ | $i=2$ | $i=3$ | $i=4$ | $i=5$ | $i=6$ | $i=7$ | $i=8$ | $i=9$ |
|----|--------------|------------------|------------------|-------|-------|-------|-------|-------|-------|-------|-------|-------|
| 02 | | Ratio of water supply portfolios | Surface water | 0.50 | 0.72 | 0.64 | 0.61 | 0.40 | 0.14 | 0.55 | 0.62 | 0.31 |
| 03 | | | Groundwater | 0.14 | 0.18 | 0.16 | 0.20 | 0.30 | 0.59 | 0.24 | 0.15 | 0.47 |
| 03 | | | Stormwater | 0.19 | 0.03 | 0.09 | 0.13 | 0.14 | 0.16 | 0.12 | 0.14 | 0.16 |
| 05 | Equilibrium 1 in the BL-MAS | | Rainwater | 0.17 | 0.07 | 0.11 | 0.06 | 0.16 | 0.11 | 0.10 | 0.09 | 0.07 |
| 06 | | Water use pattern | | P1 | P3 | P3 | P3 | P1 | P2 | P3 | P3 | P4 |
| 07 | | Mean unit water costs | | 3.05 | 3.06 | 3.06 | 3.07 | 3.06 | 3.07 | 3.06 | 3.07 | 3.07 |
| 08 | | Mean unit penalty fee | | 0.11 | 0.10 | 0.07 | 0.02 | 0.03 | 0.01 | 0.03 | 0.01 | 0.00 |
| 09 | | Gini coefficient | | | | | | 0.0011 | | | | |
| 10 | | Ratio of water supply portfolios | Surface water | 0.36 | 0.30 | 0.39 | 0.30 | 0.19 | 0.23 | 0.14 | 0.15 | 0.08 |
| 11 | | | Groundwater | 0.39 | 0.33 | 0.41 | 0.37 | 0.50 | 0.49 | 0.51 | 0.50 | 0.68 |
| 12 | Equilibrium 2 in the BL-MAS | | Stormwater | 0.15 | 0.21 | 0.10 | 0.20 | 0.16 | 0.14 | 0.22 | 0.18 | 0.14 |
| 13 | | | Rainwater | 0.11 | 0.16 | 0.11 | 0.13 | 0.15 | 0.14 | 0.13 | 0.17 | 0.10 |
| 14 | | Water use pattern | | P4 | P2 | P4 | P2 | P4 | P2 | P4 | P2 | P4 |
| 15 | | Mean unit water costs | | 3.22 | 3.24 | 3.22 | 3.21 | 3.22 | 3.23 | 3.22 | 3.22 | 3.23 |
| 16 | | Mean unit penalty fee | | 0.28 | 0.29 | 0.19 | 0.13 | 0.14 | 0.09 | 0.09 | 0.03 | 0.05 |
| 17 | | Gini coefficient | | | | | | 0.0011 | | | | |

**Note:** · *Mean unit penalty fee* for urban area $i$ is obtained as follows; The penalty fees divided by the total amounts of water supply in the urban area $i$.

As Tab. 3 shows, there are the two Stackelberg equilibrium situations in the BL-MAS, which means the two possible designs of the streamflow penalty strategies for the WM agent can achieve the minimum equity objective in the study region. An illustration of the water supply portfolios and the relevant water use patterns in all urban areas under the two equilibrium situations is in the 2rd-6th and 10th-14th rows of Tab. 3. A spatial homogeneity in the UWMs' responses to the water policy

can be observed based on the associated ratios of water supply portfolios. For example, whether it is Equilibrium 1 or 2, most UWM agents prefer to adopt a similar IGWM pattern, such as Pattern 3 in Equilibrium 1 and the Pattern 2 and 4 in Equilibrium 2, in contrast to the case without the water policy (See the 6th row of Tab. 2). This might be because the streamflow penalty strategy has a similar effect on altering the IGWM cost of all urban areas in the study watershed, forcing some UWMs to change their water supply portfolio selections to a specific IGWM pattern to increase outflows of urban systems for avoiding

high penalty fees. These findings can also be supported by the curves of the fractions of available surface water in each urban area (See in Fig. 6 a). As Fig. 6 (a) illustrates, the available amounts of surface water withdrawal for each urban area in the two equilibriums (i.e., the blue and green lines) are larger than that in the scenario without the water strategy (i.e., the red line).

The costs and equity levels in the BL-MAS are shown in the 7th-9th and 15th-17th rows of Tab. 3. In the aspects of cost, to compare with the no policy case (See the 7th rows of Tab. 2), the mean costs per unit of water use of the BL-MAS in the

two equilibriums (3.06 $/m^3$ and 3.22 $/m^3$) are higher than the cost in the no policy case (3.04 $/m^3$). This is because some urban areas pay penalty fees, such as urban areas 1 and 2, and others switch IGWM patterns, such as urban area 3, which increases the water use cost. These results demonstrate the reactions of the UWMs to the water strategy set by the WM in





IGWM at a watershed scale; the intervention of the water policy to the MAS-UWM can force UWM agents to make a trade-off between the penalty fee and the IGWM cost in a city-scale IGWM. That is, UWM agents have to alter urban water supply

portfolios and GIs construction to reduce water storage and economically increase outflow, especially for the upstream UWM agents. In the aspects of equity, undoubtedly, the Gini coefficients in the two equilibriums (0.0011) are the same, which are lower than that in the no policy case (0.0169, See the 8th rows of Tab. 2). It means that the two policy designs can mitigate the inequity of water sharing in the study region to some extent.

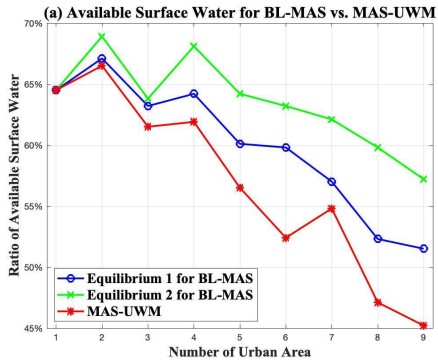
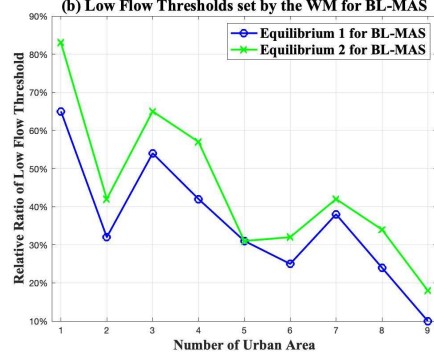

**Figure 6.** Available surface water and low flow threshold in the BL-MAS.

**Note:** · In Fig. 6 (b), *Relative ratio of low flow threshold* for urban area calculated as follows; The average of the ratio of the monthly low flow threshold set by WM agent to the maximum theoretical that threshold in an urban area.

Next, we assessed the two equilibriums - the designs of the streamflow penalty strategy - by comparing them with their

effects on watershed-scale IGWM. Hydrologically, as shown in the 6th and 14th rows in Tab. 2, most UWM agents prefer to select the IGWM pattern 3 - to withdraw more surface water - in equilibrium 1. In contrast, the UWM agents in equilibrium 2 incline to choose the Pattern 4, i.e., to extract more groundwater, which is easy to increase the outflows of urban subcatchment (See Fig. 3 e) to avoid the penalty fees. These results are consistent with the curves of the fraction of available surface water (See the blue and green lines in Fig. 6 a); The available amounts of surface water in the study area in the equilibrium 2 are

higher than that in the equilibrium 1. Economically, the IGWM costs and the penalty fees per unit of water use in equilibrium 2 are greater than those in equilibrium 1. These results can be explained in part by the curves of the relative ratio of low flow thresholds in the two equilibriums (See Fig. 6 b); the low flow thresholds in the equilibrium 2 (i.e., the green line) are higher than those in the equilibrium 1 (i.e., the blue line), which means that the policy in the equilibrium 2 is more stringent than that in the equilibrium 1. As a result, the UWMs in equilibrium 2 have to further change IGWM patterns - over the

withdrawal of groundwater and reduction in rainfall capture - to increase the outflow of the urban subcatchments to avoid the harsh penalty policy. It may cause over-reactions of the UWM agents in IGWM to the watershed water policy, thereby leading to the unnecessary costs of water use as well as the unreasonable water supply portfolios in the watershed. Therefore, the policy design in equilibrium 1 is regarded as a good watershed policy, but equilibrium 2 is not in the study area.





Lastly, we discussed the impacts of different hydroclimatic and institutional circumstances on the dynamics of the BL-MAS
in a watershed-scale IGWM. Similar to Experiment 2, a sensitivity analysis of the BL-MAS model in the study region was
conducted. In the experiment, the above equilibrium 1 of the BL-MAS was assumed as the baseline. As before, the changes
in the hydroclimatic settings were simulated by using different ratios (i.e., from 50% to 150%) of mean total precipitation and
watershed upstream inflow to the baseline. And six penalty rates - 0.002 $/m^3$, 0.004 $/m^3$, 0.005 $/m^3$ (baseline), 0.006
$/m^3$, 0.008 $/m^3$ and 0.010 $/m^3$ - were set to indicate the different institutional circumstances. Notice that we would select
the equilibrium with the lowest mean cost per unit of water to represent the result of the BL-MAS in watershed-scale IGWM
if there are several equilibriums of the BL-MAS under the same conditions.

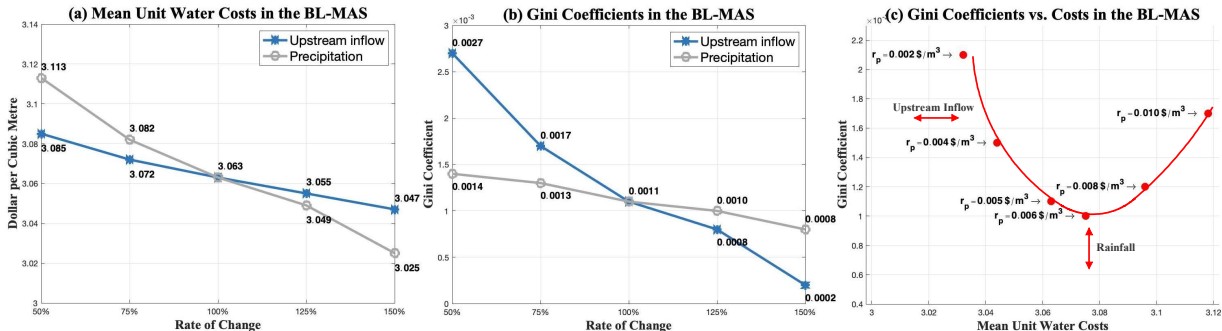

**Figure 7.** Changes in unit water costs and Gini coefficients under different institutional and hydroclimatic settings.

Fig. 7 (a) and (b) illustrate the mean unit water costs and the Gini coefficients of the BL-MAS under different upstream inflow
and precipitation conditions, respectively. As Fig. 7 (a) and (b) show, both the costs of water use and the Gini coefficients of the
BL-MAS decrease as the upstream inflow and the precipitation increase. This might be because the increases in the upstream
inflow and the precipitation not only enable the UWM agents to have more choices to make IGWM decisions to reduce the
costs of water use but also mitigate the water conflicts among urban areas in the study site, encouraging the WM agent to
loosen the water penalty policy. Besides, the cost of water use is more sensitive to the rainfall, while the Gini coefficient, in
contrast, is easily affected by the alteration in upstream inflow, which ties in with the results in Fig. 5 (a) and (b).

The water use costs and the Gini coefficients of the equilibrium situations in the BL-MAS under different penalty rates
settings are illustrated in Fig. 7 (c). As Fig. 7 (c) shows, the costs of water use of the BL-MAS increase as the penalty rate
increases, as expected. To avoid over-high penalty fees, the UWM agents have to alter the IGWM pattern further to increase
the outflows of urban systems, thereby increasing water use costs. However, there is a U-shaped curve for the Gini coefficients
when the penalty rate increases; the Gini coefficient reaches a nadir (0.001) when the penalty rate is $r_p = 0.006$ $/m^3$. The
possible reasons for these results are that the impact of the water strategy is too weak to affect UWMs' decision behavior
of IGWM as the penalty rate is too low, whereas, over-high penalty rate might force UWMs to make unreasonable IGWM
decisions, both of which fail to mitigate water conflicts between urban areas. Therefore, a suitable penalty rate is crucial in





achieving equity in watershed-scale IGWM under a specific socio-hydrologic environment. For example, in the study, the reasonable penalty rate is set as 0.006 $/$m^3$ by comparison with other alternative penalty rates.

## 5    Conclusions, limitations and future research

In this study, we focus on the issues of IGWM at three spatial scales. We developed a multiagent socio-hydrologic framework and the associated algorithm systems for simulating various dynamics of decision-makings of city-, inter-city- and watershed-scale IGWM for UWMs and WM under changing socioeconomic and hydroclimatic circumstances. To be specific, 1) an ABM-UWM, which is made by coupling IGWM-OM and UWB-SM, is developed to optimize water supply portfolios and GIs construction simultaneously - city-scale IGWM - for UWM, and it can be solved by the improved PSO algorithm - S-APSO; 2)

A MAS-UWM, which is made up of the multiple ABM-UWMs and the Muskingum-Cunge routing equations, is constructed to simulate watershed-scale IGWM via the interactions among UWMs in a river network - the Markov property, and it can be calculated via a multi-S-APSO framework. 3) A BL-MAS, which consists of the MAS-UWM and the WM agent model, is built to simulate dynamics of watershed-scale IGWM under a streamflow penalty strategy through the interactions between UWMs and WM - Stackelberg game, which is dealt with via using a nested S-APSO structure.

The combination of the model and the algorithm framework allows us to examine how the hydrological and social settings affect IGWM at different spatial scales and policy effectiveness in a watershed via the UWM- urban environment, -UWM and -WM interactions, meanwhile, to observe how the decisions of IGWM affects the watershed environment. For an illustrative purpose, the multiagent framework is applied to the Minneapolis - La Crosse section of the Upper Mississippi River Basin, the 30 million people watershed in the US. Three simulating experiments to the UWM agents model, the MAS-UWM and the

BL-MAS, are designed to test the framework's functionalities and capabilities for city-and watershed-scale IGWM in the watershed. Through various sensitivity, scenario, and comparison analyses, the results from the UWM agents model demonstrate the nonlinear relationships between city-scale IGWM decisions and hydroclimatic conditions. Even more importantly, they can capture the homogeneity in UWMs' responses of IGWM to the similar hydrologic settings to characterize the city-scale IGWM in changing environment via using a k-means clustering technique; Four types of city-scale IGWM patterns, i.e., 1) the surface

water- and 2) the groundwater-rainwater hybrid, 3) the surface water, and 4) the groundwater dominance, under the different upstream inflow and rainfall inputs are identified. Also, the results from the MAS-UWM show the different impacts of rainfall and upstream inflow on watershed-scale IGWM; the change in rainfall is apt to affect the water use cost for the UWM agents, while the shift in upstream inflow is more likely to affect the equity in water use in the study basin. Finally, the results from the BL-MAS display the various available designs of the streamflow penalty strategy, all of which can achieve the minimum

equity objective for the WM agent. Nevertheless, they can result in different water use costs for the UWM agents. The role of the penalty rate in mitigating water conflict is also revealed in the BL-MAS. Insights from these results are informative for identifying effective watershed water policies, such as setting a reasonable penalty rate and designing a good policy based on least-cost equilibrium.





This paper still has some limitations that can be addressed in future work. For example, in the UWM agent model, we develop
a lumped urban hydrologic model (UWB-SM) to simulate the urban water cycle processes in the context of the combinations
of water supply portfolios and GIs construction, which could lead to simulating results that are not accurate enough. Future
work can couple a distributed urban water balance model, such as SUWMBA (Moravej et al., 2021), with the IGWM-OM to
make water supply portfolio and GIs construction decisions more accurate and detailed for UWM.

*Data availability.*  The data of the case study examined in this study have been obtained from the United States Geological Survey (USGS),
the National Oceanic and Atmospheric Administration (NOAA), the United States Census Bureau, and the United States Environmental
Protection Agency (U.S. EPA) databases. All of the data used to generate the figures in this paper are available publicly at: https://github.
com/suoyuexh/Multiagent_IGWM.git.

## Appendix A:  Details of agent-based model for IGWM at a city scale

This appendix comprehensively elucidates the details of the agent-based model at the city scale. It is meticulously described
from various aspects including the notation, urban water balance simulation model, and the IGWM optimization model, along
with the relevant solution approaches pertaining to this model.

## A1    Notations

To facilitate the model presentation, some of the important notations used hereafter are summarized in Table A1 - Table A7.

**Table A1.** Decision variables of the UWM and WM agent.

| | |
|---|---|
| *Subscripts* | |
| $t$ | Index of month for the IGWM, where $t = 1, 2, ..., 12$; |
| $i$ | Index of the UWM agent and the associated checkpoint, where $i = 1, 2, ..., N$; |
| *Decision variables for the UWM agents* | |
| $w_s^i(t), W_s^i(t)$ | Monthly surface water supply of the UWM agent $i$ in month $t$ $[m^3, mm]$; |
| $w_g^i(t); W_g^i(t)$ | Monthly groundwater supply of the UWM agent $i$ in month $t$ $[m^3, mm]$; |
| $w_{rr}^i(t), W_{rr}^i(t)$ | Monthly rainwater supply of the UWM agent $i$ in month $t$ $[m^3, mm]$; |
| $w_{rs}^i(t), W_{rs}^i(t)$ | Monthly stormwater supply of the UWM agent $i$ in month $t$ $[m^3, mm]$; |
| $IG^i$ | Area constructed infiltration-based GIs of the UWM agent $i$ $[km^2]$; |
| $RG_s^i$ | Area constructed stormwater harvesting systems of the UWM agent $i$ $[km^2]$; |
| $RG_r^i$ | Area constructed rainwater harvesting systems of the UWM agent $i$ $[km^2]$; |
| *Decision variables for the WM agent* | |
| $S_q^i(t)$ | Low flow thresholds in month $t$ at checkpoint $i$ $[m^3]$. |





**Table A2.** Urban water demand and the hydroclimatic input parameters of the UWB-SM.

| *Input parameters for the urban water demand* | |
|---|---|
| $D_{id}(t), D_{in}(t)$ | Monthly indoor potable and non-potable water demand for in month $t$ $[m^3]$; |
| $D_o(t)$ | Monthly outdoor water demand in month $t$ $[m^3]$; |
| *Input parameters for the hydroclimatic data* | |
| $P_g(t)$ | Precipitation on the land cover in month $t$ $[mm]$; |
| $Q_{ri}(t)$ | Upstream river inflow in month $t$ $[mm]$; |
| $E_p, ET_p$ | Mean potential evaporation and evapotranspiration on the impervious and pervious surfaces $[mm]$; |
| $E_{imax}$ | Maximum interception evaporation on the pervious surfaces $[mm]$. |

**Table A3.** Measured parameters of the UWB-SM.

| *Urban area-related parameters* | |
|---|---|
| $A_u, A_p$ | Urban total and the associated pervious surfaces area $[km^2]$; |
| $r_n$ | Ratio of non-effective impervious surfaces area to total impervious surfaces area $[\%]$; |
| $r_r$ | Ratio of roof surfaces area to effective impervious surfaces area $[\%]$; |
| $r_c$ | Ratio of canopy cover area to pervious surfaces area $[\%]$; |
| $r_{rmax}, r_{smax}$ | Maximum ratios of the area constructed rainwater and stormwater harvesting systems to relevant surfaces $[\%]$; |
| $r_{imax}$ | Maximum ratio of the area constructed infiltration-based GIs to relevant surface $[\%]$; |
| *Urban depth-related parameters* | |
| $h_{ul}, h_{uh}$ | Mean depth of urban aquifer at the low and high topographic point $[m]$; |
| $h_s$ | Mean depth of urban shallow soil layer $[m]$; |
| $h_{wp}$ | Mean depth of wastewater pipe network $[m]$; |
| $h_{dp}$ | Mean depth of wells for groundwater withdraw $[m]$; |
| $n$ | Mean effective porosity $[\%]$; |
| *Storage capacity-related parameters* | |
| $S_{rmax}, S_{omax}$ | Maximum storage capacity of impervious roof and other surfaces $[mm]$; |
| $S_{grmax}, S_{gsmax}$ | Maximum storage capacity of rainwater and stormwater harvesting systems $[mm]$; |
| *Water use-related parameters* | |
| $r_{wc}$ | Water consumption rate for indoor water use $[\%]$. |

**A2  Urban water balance simulation model**

Fig. A1 (A) exhibits how the urban water mass balance is modeled in the UWB-SM; arrows illustrate how IGWM-driven water flows movement between seven storage units within an urban region - roof, other surfaces, rainwater, and stormwater harvesting systems, shallow soil layer, aquifer, and river. The state of these storage units is used to measure the total water balance within an urban area, thereby calculating available amounts of water resources in a period. As shown in Fig. A1 (A), the UWB-SM assumes that the urban water cycle receives input both from rainfall and river inflow, which together pass through the grey

and green infrastructures system and output in the form of evapotranspiration and river outflow. The vertical structure of the





**Table A4.** Calibration parameters of the UWB-SM.

| | |
|---|---|
| *Hydrologic parameters for urban surfaces* | |
| $E_{ismax}$ | Maximum evaporation on impervious surfaces when the associated storage level is saturated $[mm]$; |
| $r_{ie}$ | Ratio of average evaporation to rainfall per unit canopy cover [%]; |
| $IF$ | Infiltration factor $[dimensionless]$; |
| $I_{pmax}$ | Maximum infiltration on pervious surfaces when the associated storage level is empty $[mm]$; |
| *Hydrologic parameters for shallow soil layer* | |
| $F_{set}$ | Soil evapotranspiration scaling factor corresponding to the unlimited soil water supply $[dimensionless]$; |
| $k_s$ | Saturated hydrologic conductivity of shallow soil layer $[mm/month]$ |
| *Hydrologic parameters for aquifer and river* | |
| $k_r$ | Retention factor of aquifer [%] |
| $k_{rd}$ | Routing delay factor of river $[dimensionless]$ |
| *Parameters for urban watet system* | |
| $r_{dl}, r_{wl}$ | Leakage rate of supply and wastewater pipe networks [%]; |
| $I_{gi}$ | Mean groundwater infiltration into wastewater pipe networks $[mm]$; |
| $F_{gi}$ | Groundwater infiltration scaling factor when wastewater pipe network is totally submerged by groundwater $[dimensionless]$. |

UWB-SM is illustrated in Fig. A1 (B), which consists of four components; urban surfaces, shallow soil layer, aquifer, and river. Hydrologically, various surface-subsurface water interactions and pipe network exfiltration and infiltration are considered via simulating water fluxes transfers between these storage units on the basis of the water mass conservation principle.

The UWB-SM requires four types of input data; IGWM decision, urban water demand, hydroclimatic, and urban land and water characteristic data. To the specific, IGWM decision data is used to update the amounts of four types of water monthly (i.e., water supply portfolios) and the areas to construct three types of GIs systems determined by a UWM agent. Urban water demand data refers to the monthly indoor and outdoor water demands, which can be estimated based on the associated urban population and economic development levels in a study area. Mean monthly river inflow, precipitation, and potential evapotranspiration within a study region are required as hydroclimatic data. The urban land and water characteristics of the

study area are described by the calibrated and measured parameters. Each measured parameter relates directly to a physical catchment characteristic; an appropriate value can be determined through measurement, observation, or local experience. A list of measured parameters is given in Tab. A3 . The 13 calibration parameters, along with the associated units, symbols, and ranges listed in Tab. A4, are grouped according to their land features; surfaces, soil layer, aquifer, river, and urban water system. These values are adjusted during the calibration to optimize the selected objective function.

There are four water storage units - roof, other surfaces, rainwater, and stormwater harvesting systems - to be set on the urban surfaces (See Fig. A1 C). In Fig. A1 (C), the urban surface consists of the pervious and impervious areas based on their infiltration rates. The impervious surfaces on which the infiltration process is ignored, are divided into effective and





**Table A5.** Urban surface-related variables and parameters of the UWB-SM

| | |
|---|---|
| *Urban area-related parameters* | |
| $A_i$ | Impervious surfaces area $[km^2]$; |
| $A_r, A_o$ | Impervious roof and other surfaces area $[km^2]$; |
| *State variables of storage systems* | |
| $S_r(t), S_o(t)$ | Store levels of impervious roof and other surfaces in month $t$ $[mm]$; |
| $S_{gr}(t), S_{gs}(t)$ | Store levels of rainwater and stormwater harvesting systems in month $t$ $[mm]$; |
| $S_s(t), S_a(t)$ | Store levels of shallow soil layer and aquifer in month $t$ $[mm]$; |
| *Evaporation-related variables of systems* | |
| $E_{ir}, E_{io}$ | Evaporation on the impervious roof and other surfaces $[mm]$; |
| $E_{in}$ | Evaporation on the non-effective impervious surfaces $[mm]$; |
| $E_{pi}$ | Interception evaporation on the pervious surfaces $[mm]$; |
| *Runoff-related variables of systems* | |
| $Q_{rir}, Q_{rio}$ | Runoff on the impervious roof and the other surfaces $[mm]$; |
| $Q_{rrr}$ | Runoff on the rainwater harvesting systems $[mm]$; |
| $Q_{rin}, Q_{rp}$ | Runoff on the non-effective impervious and the pervious surfaces $[mm]$; |
| $Q_{ru}$ | Runoff on the urban surfaces before stormwater harvesting $[mm]$; |
| $Q_r, q_r$ | Runoff on the urban surfaces after stormwater harvesting $[mm, m^3]$; |
| *Infiltration-related variables of systems* | |
| $I_p$ | Infiltration on the pervious surfaces $[mm]$; |
| *Auxiliary variables of systems* | |
| $P_e$ | total outdoor water input due to effective precipitation and outdoor water use $[mm]$; |
| $U_o(t)$ | Outdoor water use amount in month $t$ $[mm]$ |
| $f_{sat}$ | Ratio of saturated area to pervious surfaces $[\%]$ |
| $S_{smax}$ | Maximum storage capacity of shallow soil layer $[mm]$. |

non-effective areas. The effective area is used to represent the portion of impervious surface runoff that directly drains to the stormwater drainage system. It is assumed that runoff on the non-effective part drains onto adjacent pervious areas, and the remaining water is evaporated. Unlike impervious surfaces, the pervious area can infiltrate a part of runoff into the underground soil layer, decreasing the runoff and increasing groundwater recharge. In the effective impervious area, roofs and other surfaces (e.g., roads and paved areas) are classified by the construction conditions of different types of GIs. It is assumed that rainwater harvesting systems are only allowed to be built on the roof area, and infiltration-based GIs (e.g., infiltration trenches and porous pavements) are required to construct on the non-roof impervious area (i.e., Other surfaces), which turns it into the pervious area. The roofs, the rainwater harvesting systems, and other impervious surfaces can generate impervious surface runoff when full. Their storages are depleted by evaporation (i.e., the roofs and other surfaces) or rainwater extraction (i.e., rainwater harvesting systems). Besides, the stormwater harvesting systems are supposed to collect runoff from the whole urban surface (i.e., the pervious and the impervious surfaces) for stormwater supply.

We assume that a UWM agent can supply surface water and groundwater by extracting from the river and the aquifer storage units. It also can access rainwater and stormwater collected by rainwater and stormwater harvesting systems. Indoor





**Table A6.** Underground and river-related variables and parameters of the UWB-SM

*State variables of storage systems*

$S_{ri}(t)$      Store levels of river in month $t$ [$mm$];

*Drainage-related parameters of systems*

$P_s$      Percolation in the shallow soil layer [$mm$];

$Q_b$      Base flow in the aquifer [$mm$];

$S_{amin}$      Minimum storage level of aquifer for generating base flow [$mm$];

$Q_{ro}(t)$      River outflow in month $t$ [$mm$];

*Evapotranspiration-related variables of systems*

$ET_s, ET_a$      Evapotranspiration in the shallow soil layer and aquifer [$mm$];

*Pipe network-related variables*

$L_d, L_w$      Leakage of pipe network for water supply and wastewater drainage [$mm$];

$GI_w$      Groundwater infiltration into wastewater pipe network [$mm$];

$Q_{wd}, q_{wd}$      Wastewater drainage to river [$mm, m^3$];

*Infiltration-related variables of systems*

$I_p$      Infiltration on the pervious surfaces [$mm$];

*Auxiliary variables of systems*

$f_{iw}$      Ratio of submersed wastewater pipelines to total wastewater pipelines [%].

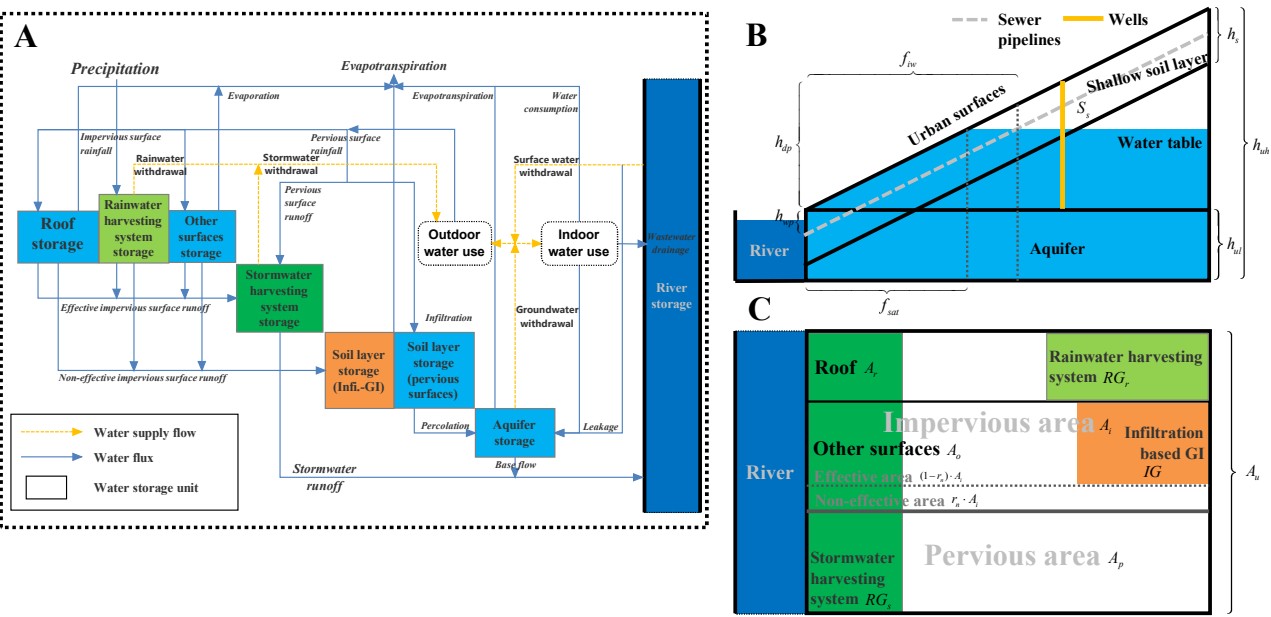

**Figure A1.** The structure of the UWB-SM.

**Note:** In Fig. A1 (B) and (C), relevant symbols are listed in Tab. A2, A3 and A5 in Appendix A1.

and outdoor demands are considered in the model. Following Mitchell et al. (2001) approach, indoor water use is divided





**Table A7.** Variables and parameters of the IGWM-OM

| | |
|---|---|
| *Cost-related parameters of GI systems* | |
| $c_{ig}, c_{rg}, c_{sg}$ | Mean annual construction cost of unit area for infiltration- based GIs and rainwater and stormwater harvesting systems [$/km^2$] ; |
| $e_{ig}, e_{rg}, e_{sg}$ | Cost scaling coefficient for infiltration-based GIs and rainwater and stormwater harvesting systems [$dimensionless$]; |
| *Cost-related parameters of water supply and sewage drainage* | |
| $c_{ri}, e_{ri}$ | Mean cost and associated scaling coefficient of unit surface water supply [$/m^3$, $-$]; |
| $c_a(t)$ | Cost of water withdraw from aquifer in month $t$ [$/m^3$]; |
| $c_{amax}, c_{amin}$ | Maximum and minimum cost of unit groundwater supply [$/m^3$]; |
| $Sc_{amax}, Sc_{amin}$ | Storage level of aquifer for maximum and minimum cost of water withdraw [$mm$]; |
| $c_{rw}, c_{sw}$ | Mean cost of unit rainwater and stormwater supply [$/m^3$]; |
| $c_{wd}, e_{wd}$ | Mean cost and associated scaling coefficient of unit wastewater drainage [$/m^3$, $-$]; |
| *Water availability-related parameters* | |
| $S_{mina}, S_{minr}$ | Minimum storage level of aquifer and river for water withdraw [$mm$]; |
| *Water supply capacity-related parameters* | |
| $WC_a, WC_{ri}$ | Aquifer and River water supply capacity of grey infrastructure systems [$mm$]; |
| *Auxiliary variables of systems* | |
| $f_{sm}$ | Ratio of soil moisture for plant demand to saturated soil moisture [%] |
| $C_{gi}, C_{si}, C_{wi}$ | Total annual costs of GIs development, water supply and wastewater drainage for the UWM agent $i$ [$] ; |
| $TC_i$ | Total cost of IGWM of the UWM agent $i$ [$]. |

**Note:** [$-$] represents dimensionless.

**Table A8.** Parameters of the S-APSO

| | |
|---|---|
| *Subscripts* | |
| $l$ | Index of the particle, where $l = 1, 2, ..., L$; |
| $\tau$ | Index of the iteration, where $\tau = 1, 2, ..., \Gamma$; |
| *Particle-assessment-related parameters* | |
| $Fitness[P_l(\tau)]$ | The fitness value of the $l$th particle at the $\tau$th iteration; |
| *Particle-updating-related parameters* | |
| $P_l(\tau), V_l(\tau)$ | The position and the velocity of the $l$th particle at the $\tau$th iteration; |
| $PB_l$ | Personal best particle of the $l$th particle achieved so far; |
| $GB$ | Global best particle among all the particles; |
| $EP(\tau)$ | Elite particle at the $\tau$th iteration; |
| $\theta(\tau)$ | Inertia weight at the $\tau$th iteration; |
| $c_p(\tau), c_s(\tau)$ | Personal and social acceleration coefficients at the $\tau$th iteration; |
| $r_p, r_s$ | The uniformly distributed random numbers generated within [0, 1]; |
| $\varepsilon(\tau)$ | The ratio for social learning at the $\tau$th iteration; |

into potable and non-potable water demands, and outdoor water use is regarded as non-potable. The potable water demand



is assumed to be only satisfied by surface water and groundwater supply, and all water resource supplies can support the non-potable case.

**Urban Surfaces:** In the UWB-SM, we assume that precipitation is distributed evenly over the entire area and does not take urban spatial features into account due to the limited monthly impact on urban water management decisions. By the assumption, the amounts of rainfall on different urban surfaces can be calculated by ratios of relevant surface areas to the urban area's multiple gross precipitations. Notice that the pervious surface receives input from effective precipitation, outdoor water use, and surface runoff from adjacent impervious areas. The effective precipitation is defined by the relevant gross precipitation

minus the interception and evaporation on the plant cover area within the region. For the infiltration process, the infiltration and saturation excess are modeled to calculate the runoff and the infiltration according to the soil moisture conditions. The impervious and pervious surface runoff flows into the drains, collected by stormwater harvesting systems. The rest discharges into the river. The relevant terms and units of which are defined in Tab. A5. Taking the IGWM decisions of UWM agent $i$ in month $t$ as an example, the detail of the hydrologic process on the urban surface is formulated as follows;

*Hydrologic process on the impervious roof surfaces:* The water mass balance equation for the impervious roof surfaces can be expressed as;

$$S_r(t) - S_r(t-1) = P_g(t) - E_{ir}(t) - Q_{rir}(t), \tag{A1}$$

where the roof area $A_r = r_r \cdot (A_u - A_p)$, $P_g(t)$ represents precipitation on the impervious roof surfaces, and evaporation can be calculated by $E_{ir}(t) = \max[E_{ismax} \cdot \frac{S_r(t-1)}{S_{rmax}}, E_p(t)]$ based on Mitchell et al. (2001) equation. And the storage levels is updated by a reservoir model; $S_r(t) = \min[S_r(t-1) + P_g(t) - E_{ir}(t), S_{rmax}]$.

*Hydrologic process on the other impervious surfaces:* Similar to the impervious roof surfaces, the water mass balance formulation for the other impervious surfaces is shown as;

$$S_o(t) - S_o(t-1) = P_g(t) - E_{io}(t) - Q_{rio}(t), \tag{A2}$$

where the other impervious surfaces area $A_o$ is equal to $(1 - r_r) \cdot (A_u - A_p)$, and the calculation of the evaporation is also dependent of Mitchell et al. (2001) method; $E_{io}(t) = \max[E_{ismax} \cdot \frac{S_o(t-1)}{S_{omax}}, E_p(t)]$. The update of relevant storage levels, $S_o(t)$ is equal to $\min[S_o(t-1) + P_g(t) - E_{io}(t), S_{omax}]$.

*Hydrologic process for the rainwater harvesting systems:* In the rainwater harvesting systems, evaporation is ignored because of the assumption of that rainwater is collected rapidly and storage is sealed. We can draw its mass balance relationship as follows;

$$S_{gr}(t) - S_{gr}(t-1) = P_g(t) - W^i_{rr}(t) - Q_{rrr}(t), \tag{A3}$$

where $W^i_{rr}(t)$ is obtained from the unit conversion from $[m^3]$ to $[mm]$, which is equal to $\frac{w^i_{rr}(t)}{1000 \cdot RG^i_r}$, and the updated storage levels of the system $S_{gr}(t)$ is equivalent to $\min[S_{gr}(t-1) + P_g(t) - W^i_{rr}(t), S_{grmax}]$.

*Hydrologic process on the non-effective area:* Hydrologic process on the non-effective area: For the simplicity of the model, it assumed that rainfall only generates runoff and evaporation on the non-effective area. This study calculates the corresponding runoff based on the following equation - $Q_{rin}(t) = r_n \cdot [\frac{A_r}{A_i} \cdot Q_{rir}(t) + \frac{A_r - RG_r}{A_i} \cdot Q_{rio}(t) + \frac{A_o - IG}{A_i} \cdot Q_{rrr}(t)]$, where the



impervious area $A_i = A_u - A_p - IG$.

$$P_g(t) = Q_{rin}(t) + E_{in}(t). \tag{A4}$$

*Hydrologic process on the pervious surfaces:* The water mass conservation equation for the pervious surfaces is represented as follows;

$$Q_{rin}(t) + P_g(t) + U_o(t) = Q_{rp}(t) + E_{pi}(t) + I_p(t). \tag{A5}$$

The right side of Eqs. A5 donates the inflow on the pervious surface, including runoff from the non-effective area, precipitation, and outdoor water use. The outdoor water use is the total amount of water supply minus indoor water demand - $U_o(t) = (1 - r_{dl}) \cdot [W_s^i(t) + W_g^i(t)] + W_{rr}^i(t) + W_{rs}^i(t) - \frac{d_{ip}(t) + d_{in}(t)}{1000 \cdot A_u^i}$, where $W_s^i(t)$, $W_g^i(t)$ and $W_{rs}^i(t)$ are obtained from the unit conversion from $[m^3]$ to $[mm]$, which are equal to $\frac{w_s^i(t)}{1000 \cdot A_u^i}$, $\frac{w_g^i(t)}{1000 \cdot A_u^i}$ and $\frac{w_{rs}^i(t)}{1000 \cdot RG_s^i}$.

The left hand side of Eqs. A5 represents outflow on the pervious surfaces, consisting of runoff, interception evaporation and infiltration, separately. Interception evaporation is calculated based on urban vegetation canopy area and associated features, following Van Dijk and Bruijnzeel (2001) approach - $E_{pi}(t) = \min[r_c \cdot r_{ie} \cdot P_g(t), E_{imax}]$. The calculation of infiltration is based on two hydrologic processes; saturated and infiltration excess (Viney et al., 2015). we assume that there would be no infiltration on the saturated area within the pervious surface, and its ratio $f_{sat}(t) = \max[\frac{\frac{S_a(t-1)}{1000 \cdot n} - h_{ul}}{h_{uh} - h_{ul}}, 0]$ (See Fig. A1 B). For the other part - unsaturated area, an infiltration rate estimated by using an exponential function of storage levels of the shallow soil layer (Chiew and McMahon, 1999), which infiltration is minimum when the storage level is saturated and continuously increases to a maximum when the storage level is empty. Therefore, the formulation of infiltration is $I_p(t) = [1 - f_{sat}(t)] \cdot \max[I_{pmax} \cdot e^{-IF \cdot \frac{S_s(t-1)}{S_{smax}}}, P_e(t) + Q_{rin}(t)]$, where maximum storage capacity of shallow soil layer $S_{smax} = 1000 \cdot n \cdot h_s$ and total outdoor water input $P_e(t) = P_g(t) + U_o(t) - E_{pi}(t)$.

*Hydrologic process for the stormwater harvesting systems:* Similar to rainwater harvesting systems, stormwater harvesting systems is also hypothesized to have no evaporation. Its mass balance equation can be represented by:

$$S_{gs}(t) - S_{gs}(t-1) = Q_{ru}(t) - W_{rs}^i(t) - Q_r(t), \tag{A6}$$

where runoff that is available for collection is the sum of runoff from the effective impervious and the pervious surfaces - $Q_{ru} = Q_{rp}(t) + (1 - r_n) \cdot [\frac{A_r}{A_i} \cdot Q_{rir}(t) + \frac{A_r - RG_r}{A_i} \cdot Q_{rio}(t) + \frac{A_o - IG}{A_i} \cdot Q_{rrr}(t)]$. The relevant storage levels can be updated by: $S_{gs}(t) = \min[S_{gs}(t-1) + Q_{ru} - W_{rs}^i(t), S_{gsmax}]$.

**Shallow Soil Layer:** As shown in Fig. A1 (B), a non-linear reservoir model with the depth $h_s$ is given to describe relevant hydrologic processes in the shallow soil layer. The shallow soil layer receives water from the urban surfaces by infiltration, and drains water by percolation into the aquifer and evapotranspiration to the atmosphere. Therefore, the water mass balance equation for the shallow soil layer can be written as:

$$S_s(t) - S_s(t-1) = I_p(t) - P_s(t) - ET_s(t) \tag{A7}$$





where percolation is assumed to occur according to the following Frost et al. (2016) equation for shallow soil layer - $P_s(t) =$
$k_s \cdot [\frac{S_s(t-1)}{S_{smax}}]^2$. The evapotranspiration process occurs in the unsaturated portion of the shallow soil layer, and it can be given
by using Frost et al. (2016) method: $ET_s(t) = [1 - f_{sat}(t)] \cdot F_{set} \cdot ET_p(t) \cdot \min[\frac{S_s(t-1)+I_p(t)}{S_{smax}}, 1]$.

**Aquifer:** In the UWB-SM, the only unconfined aquifer is considered as a linear reservoir model to simulate groundwater-related hydrologic dynamics (Mitchell et al., 2001), indicating that there is no deep seepage from the aquifer. The aquifer receives water from percolation, and leakage of water supply and wastewater pipelines, and discharges water in the manners
of base flow, evapotranspiration, groundwater extraction for use and infiltration into wastewater pipelines (Ellis, 2001). Hence, the mass balance formulation of aquifer can be expressed as:

$$S_a(t) - S_a(t-1) = P_s(t) + L_d(t) + L_w(t) - GI_w(t) - W_g^i(t) - ET_a(t) - Q_b(t) \tag{A8}$$

where the leakage of water supply pipe networks is $L_d(t) = r_{dl} \cdot [W_{rr}^i(t) + W_{rs}^i(t)]$, and the calculation of the sewer pipelines infiltration and exfiltration (leakage) is by comparing the depth of wastewater pipelines with groundwater table (Wolf, 2006) - groundwater would infiltrate into wastewater pipelines when the pipe is below the groundwater table, whereas wastewater
would leak into the aquifer from pipelines when the pipe is above the level of the groundwater table. Fig. A1 (C) illustrates that the fraction of the sewer pipelines that is below the groundwater table is defined as $f_{iw}(t) = \max[\frac{\frac{S_a(t-1)}{1000 \cdot n} + h_{wp} - h_{ul}}{h_{uh} - h_{ul}}, 0]$, which is used to calculate the sewer pipelines infiltration $GI_w(t) = f_{iw}(t) \cdot I_{gi} \cdot F_{gi}$ via using Wolf (2006) method, and the associated exfiltration part of pipe networks $L_w(t) = r_{wl} \cdot (1 - r_{wc}) \cdot (1 - f_{iw}(t)) \cdot [\frac{d_{ip}(t) + d_{in}(t)}{1000 \cdot A_u^i}]$ though applying Mitchell et al. (2001) equation. The evapotranspiration in the aquifer occurs in the saturated portion of the shallow soil layer (Fig. A1 C), which can
also be computed according to Frost et al. (2016) method: $ET_a(t) = f_{sat}(t) \cdot F_{set} \cdot ET_p(t)$. The amounts of base flow is assumed to be linearly proportional to the storage level of the aquifer (Fenicia et al., 2006) - $Q_b(t) = \max[k_r \cdot (S_a(t-1) - S_{amin}), 0]$, where $S_{amin} = 1000 \cdot n \cdot h_{ul}$.

**River:** We assume that the "River" component of the UWBv-SM includes all surface water bodies within an urban area. It accepts water from upstream inflow, runoff within the urban surfaces and base flow from the aquifer, and drains water to the
adjacent downstream region. In addition, an urban area can withdraw water from the river, and discharge wastewater into it. So, the water mass conservation equation for a river can written as:

$$S_{ri}(t) - S_{ri}(t-1) = Q_{ri}(t) + Q_r(t) + Q_b(t) + Q_{wd}(t) - W_s^i(t) - Q_{ro}(t) \tag{A9}$$

where total wastewater drainage is calculated as the sum of indoor water drainage and sewer infiltration and sewer leakage have been subtracted: $Q_{wd} = (1 - r_{wc}) \cdot [\frac{d_{ip}(t) + d_{in}(t)}{1000 \cdot A_u^i}] + GI_w(t) - L_w(t)$. The outflow of the river is routed via a notional river store level, which can be controlled by a routing delay factor ($k_{rd}$) according to Frost et al. (2016) equation: $Q_{ro}(t) =$
$(1 - e^{-k_{rd}}) \cdot [S_s(t-1) + Q_{ri}(t) + Q_r(t) + Q_b(t) + Q_{wd}(t) - W_s^i(t)]$.

## A3 IGWM optimization model

Before model construction, the fundamental assumptions are given.

ASSUMPTION 1. *Three types of GIs can only be built in specific urban areas.*





ASSUMPTION 2. *Four types of urban water demand need to be met via four types of water supply.*

ASSUMPTION 3. *All water resources for supply can only be withdrawn or collected in urban areas.*

ASSUMPTION 4. *The combined sewer system is considered in the urban system.*

Some of these assumptions are imposed for the simplicity of the model. Assumption 1 describes urban land features and the space limitations for the development of three types of GIs, which is consistent with corresponding settings of the UWB-SM (See Fig. A1 C). Assumption 2 is coherent with the associated hypothesis of the UWB-SM. Meanwhile, the irrigation
demand of urban green space is also considered here. Assumption 3 indicates that inter-watershed water transfer schemes are not considered in the model for simplicity. Assumption 4 represents that the model would take the sum of stormwater and wastewater into account in the calculation of urban wastewater drainage cost. The relevant decision variables and parameters of the IGWM-OM are listed in Tab. A1 and A7, separately. Based on the above assumptions, using the UWM agent $i$ as an example, the IGWM-OM model is formulated as a non-linear programming as follows:

$$\min_{W,GI} TC_i = C_{gi} + C_{si} + C_{wi} \tag{A10a}$$

$$s.t. \begin{cases} C_{gi} = c_{ig} \cdot (IG^i)^{e_{ig}} + c_{rg} \cdot (RG_r^i)^{e_{rg}} + c_{sg} \cdot (RG_s^i)^{e_{sg}}, & (01) \\ C_{si} = \sum_{t=1}^{12} [c_{ri} \cdot w_s^i(t)^{e_{ri}} + [c_a(t)]_* \cdot w_g^i(t) + c_{rw} \cdot w_{rr}^i(t) + c_{sw} \cdot w_{rs}^i(t)], & (02) \\ C_{wi} = \sum_{t=1}^{12} c_{wd} \cdot [q_r(t) + q_{wd}(t)]^{e_{wd}}, & (03) \\ w_s^i(t) = 1000 \cdot A_u^i \cdot W_s^i(t), \; w_g^i(t) = 1000 \cdot A_u^i \cdot W_g^i(t), & \forall t \quad (04) \\ w_{rr}^i(t) = 1000 \cdot A_u^i \cdot W_{rr}^i(t), \; w_{rs}^i(t) = 1000 \cdot A_u^i \cdot W_{rs}^i(t), & \forall t \quad (05) \\ q_r(t) = 1000 \cdot A_u^i \cdot [Q_r(t)]_*, \; q_{wd}(t) = 1000 \cdot A_u^i \cdot [Q_{wd}(t)]_*, & \forall t \quad (06) \\ 0 \le IG^i \le r_{imax} \cdot A_o, \; 0 \le RG_r^i \le r_{rmax} \cdot A_r, \; 0 \le RG_s^i \le r_{smax} \cdot A_u, & (07) \\ 0 \le W_s^i(t) \le max[0, min([S_{ri}(t-1)]_* + Q_{ri}(t) - S_{minr}, WC_{ri})], & \forall t \quad (08) \\ 0 \le W_g^i(t) \le max[0, min([S_a(t-1)]_* - S_{mina}, WC_a)], & \forall t \quad (09) \\ 0 \le W_{rr}^i(t) \le [S_{gr}(t-1)]_* + \frac{RG_r^i}{A_u} \cdot P_g(t), & \forall t \quad (10) \\ 0 \le W_{rs}^i(t) \le [S_{gs}(t-1)]_* + \frac{RG_s^i}{A_u} \cdot [Q_{ru}(t)]_*, & \forall t \quad (11) \\ (1 - r_{dl}) \cdot [w_s^i(t) + w_g^i(t)] \ge D_{id}(t), & \forall t \quad (12) \\ (1 - r_{dl}) \cdot [w_s^i(t) + w_g^i(t)] + w_{rr}^i(t) + w_{rr}^i(t) \ge D_{id}(t) + D_{in}(t) + D_o(t), & \forall t \quad (13) \\ f_{sm} \cdot (1 - [f_{sat}(t)]_*) \cdot S_{smax} \le [S_s(t)]_*, & \forall t \quad (14) \end{cases} \tag{A10b}$$

where $[-]_*$ represents that the parameter is from simulation results of the UWB-SM. Equation (A10a) represents the objective function of the model, which aims to minimize the annual IGWM cost for the UWM agent. This cost is comprised of GI construction ($C_{gi}$), water supply ($C_{si}$), and wastewater drainage costs ($C_{wi}$). The calculation methods for these three costs are presented in the first to third rows of Equation (A10b). The fourth to sixth rows of Equation A10b indicate the unit conversion process from millimeters ($[mm]$) to cubic meters ($[m^3]$), which is essential for coupling the UWB-SM (using the unit $[mm]$)
with the IGWM optimization model (using the unit $[m^3]$). The seventh row of Equation (A10b) establishes the constraints for GI construction. Meanwhile, the eighth to eleventh rows describe the water supply constraints for surface water, groundwater, rainwater, and stormwater withdrawals, respectively. Lastly, the twelfth to fourteenth rows of the equation outline the water demand constraints concerning potable water demand, total water demand, and irrigation water demand.





## A4 Solution approach

Unlike traditional water management models (Loucks and Van Beek, 2017), it might be hard to solve the UWM agent model (Eq. A10), which consists of the IGWM-OM and UWB-SM, via using general methods, because of the non-linearity of its objective function and complexity of its solution space induced by coupling with the UWB-SM. To solve complex water resources management problems, heuristic search-based techniques, such as particle swarm optimization (PSO) that is a swarm-based intelligence method to search for globally optimal solutions via imitating swarm behavior in birds flocking (Kennedy and 

Eberhart, 1995), are widely applied and developed (Nicklow et al., 2010; Chang et al., 2013). Therefore, in this study, according to features of the UWM agent model, a simulation-based adaptive particle swarm optimization (S-APSO) is designed, and its flowchart diagram is illustrated in Fig. A2 (A). As shown in Fig. A2 (A), compared with particle initialization and evaluation for standard PSO, a coupling procedure proposed above for data exchange between IGWM-OM and UWB-SM is nested to make sure all particles feasible and measurable. Compared with the standard PSO updating mechanism, a Boltzmann selection 

operator and an evolutionary state-based parameter adaptation scheme are added to avoid premature convergence to improve algorithm performance; a simulation-based check & repair mechanism is introduced to guarantee all particles feasible during the particle updating process. These key features of the proposed S-APSO are explained as follows.

**Solution representation and particle performance calculation:** In the S-APSO, a particle representing a feasible solution of the Eq. (A10) can be encoded as an array with 51 dimensions, which indicates 3 GI construction variables and $4 \times 12$ 

monthly water supply variables, respectively. It can be written as follows,

$$
\begin{aligned}
P_l(\tau) \quad &= [p_l^1(\tau), p_l^2(\tau), p_l^3(\tau), p_l^4(\tau), p_l^5(\tau), p_l^6(\tau), p_l^7(\tau), ..., p_l^{51}(\tau)] \\
&\leftrightarrow [IG^i, RG_r^i, RG_s^i, W_s^i(1), W_g^i(1), W_{rr}^i(1), W_{rs}^i(1), ..., W_{rs}^i(12)],
\end{aligned}
\tag{A11}
$$

where the relevant parameters of the S-APSO are listed in Tab. A8 of Appendix A1.

To measure the performance of each particle, the objective function in the UWM agent model (Eq. A10a) is regarded as the fitness function for particles, indicating that the value of the objective function for particles is used to represent their merits in a swarm, which, therefore, can be expressed as

$$
Fitness[P_l(\tau)] = C_{gi} + C_{si} + C_{wi}
\tag{A12a}
$$

$$
\begin{cases}
C_{gi} = c_{ig} \cdot [p_l^1(\tau)]^{e_{ig}} + c_{rg} \cdot [p_l^2(\tau)]^{e_{rg}} + c_{sg} \cdot [p_l^3(\tau)]^{e_{sg}}, \\
C_{si} = \sum_{t=1}^{12} [c_{ri} \cdot p_l^{(4t)}(\tau)^{e_{ri}} + [c_a(t)]_* \cdot p_l^{(4t+1)} + c_{rw} \cdot p_l^{(4t+2)} + c_{sw} \cdot p_l^{(4t+3)}], \\
C_{wi} = \sum_{t=1}^{12} c_{wd} \cdot [q_r(t) + q_{wd}(t)]^{e_{wd}},
\end{cases}
\tag{A12b}
$$

Note that, as seen in Eq. A12, to complete a performance calculation for each particle in S-APSO is required by running the coupling procedure mentioned above.

**Particle Initialization:** The initial storage levels of the river, the aquifer, the rainwater, and the stormwater harvesting systems to determine the available amounts of four water sources and the runoff and the wastewater to calculate the costs of drainage. It means that generating a feasible solution of the IGWM-OM involves conducting the UWB-SM procedure many 

times. Furthermore, as shown in the UWB-SM, the solutions at one stage from the IGWM-OM will also be as input data of the UWB-SM to update urban hydrologic conditions for simulation at next stage. Therefore, the UWB-SM and the IGWM-OM are



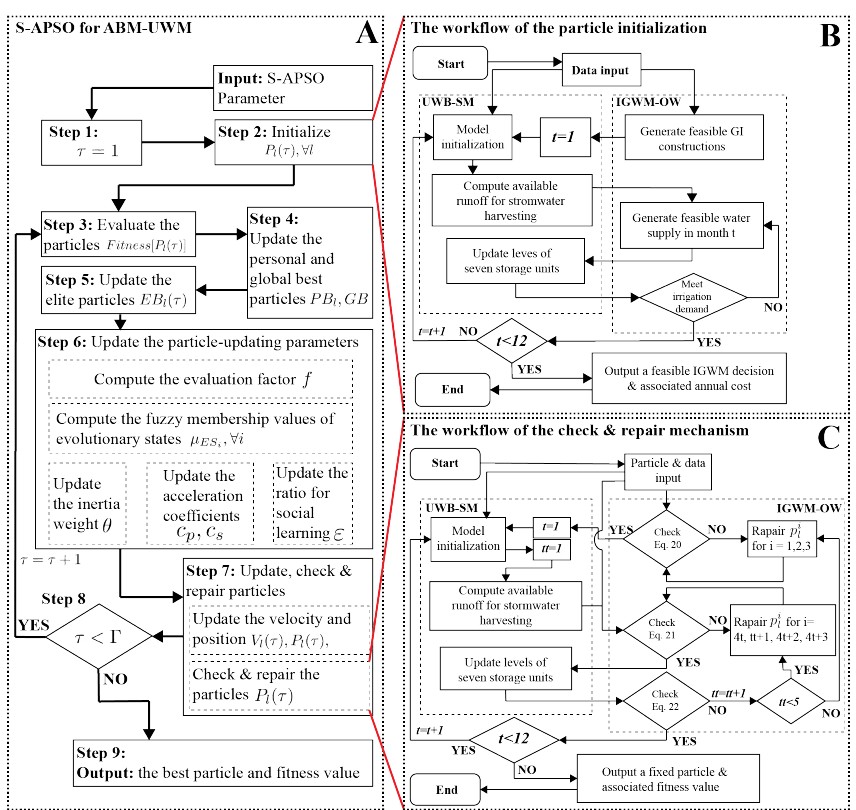

**Figure A2.** Flowchart diagram of the S-APSO

tightly coupled at the source code level, i.e., the routine of the UWB-SM is embedded into the algorithm for the IGWM-OM. The primary data exchanged and shared between the two models are 1) GI construction decision variables (i.e., construction areas for three types of GIs from IGWM-OM), 2) water supply decision variables (i.e., monthly water supply amounts for four

kinds of water sources from IGWM-OM) 3) urban hydrological parameters (e.g., monthly storage levels of storage units from UWB-SM). The workflows of coupling two models for the generation of a feasible city-scale IGWM decision are shown in Fig. A2 (B):

    Fig. A2 (B) illustrates a two-phase data exchange procedure between the UWB-SM and the IGWM-OM for a feasible solution of IGWM-OM. In Phase 1, internal data exchange between two models at a time for UWB-SM initialization. Initial data

(e.g., urban water demand, land and water characteristic data, hydroclimatic data) are inputted into two models, respectively. Available construction areas for infiltration-based GI, rainwater, and stormwater harvesting systems are generated by the GIs construction constraints, all of which, as input data, will update the urban land features of the UWB-SM.

    In Phase 2, internal data exchange between two models at the monthly scale generates a feasible water supply decision variable for a UWM agent month by month. In each month, the UWB-SM runs at least twice and exchanges data with the





IGWM-OM, relying on the internal linkage between their modules, as detailed below: UWB-SM initializes the procedure
to update urban hydrologic conditions based on the storage levels of seven storage units in the previous month and then
computes available runoff for stormwater harvesting this month. Next, runoff along with storage levels of four storage units in
the previous month (i.e., river, aquifer, rainwater, and stormwater harvesting systems) are used by IGWM-OM as part of the
input to generate feasible monthly water supply decision variable for surface water, groundwater, rainwater and stormwater in

this month by the water supply and demand constraints. After that, these decision variables are transferred to the UWB-SM
procedure to simulate relevant urban hydrologic variables in this month (e.g., storage levels of seven storage units), and the
updated urban hydrologic variables (e.g., storage level of soil layer) then is sent back to the IGWM-OM for re-checking the
feasibility of these decision variables using the irrigation demand constraints. These decision variables will be re-generated by
the IGWM-OM if the check fails to pass. At the same time, they will be recorded to calculate relevant IGWM cost in this month

through the IGWM cost objective function, and all updated storage units data will be used to initialize the UWB-SM procedure
for simulating calculation in next month. At the end of this month, the time criteria $t$ update to $t + 1$. Both UWB-SM and
IGWM-OM continue to generate water supply decision variables in the next month until the termination criteria are satisfied.
At the end of the simulation, the coupling procedure can generate a feasible IGWM decision variable for the UWM agent and
associated annual cost.

**Particle-updating mechanism:** There is a shortage of standard PSO when dealing with problems with the complexity of
solution spaces (Liang et al., 2006), such as Eq. (A10). It is easy to occur premature convergence, which hinders the search
for global optima. The main reason behind the premature convergence seems to be that the particle-updating mechanism
makes particles' information exchange frequency, which may result in fast clustering of particles (Riget and Vesterstrøm,
2002). Therefore, to avoid premature convergence, the S-APSO applies two strategies - combination with auxiliary operators

and control of algorithm parameters to modify traditional particle-updating equations. Specifically, for the introduction of
auxiliary operators, a new particle, the so-called elite particle, is selected from all personal best particles so far via using a
Boltzmann selection operator for each iteration. Then the linear combination of the global best and the elite particle is used to
formulate the social learning components of the particle-updating equations. The improved particle-updating mechanism can
be mathematically written as

$$
\begin{aligned}
V_l(\tau + 1) = \quad & \theta(\tau) \cdot V_l(\tau + 1) + c_p(\tau) \cdot r_p \cdot [PB_l - P_l(\tau)] + \\
& c_s(\tau) \cdot r_s \cdot \{\varepsilon(\tau) \cdot [GB - P_l(\tau)] + (1 - \varepsilon(\tau)) \cdot [EP(\tau) - P_l(\tau)]\},
\end{aligned}
\tag{A13a}
$$

$$
P_l(\tau + 1) = V_l(\tau + 1) + P_l(\tau),
\tag{A13b}
$$

where $EP_l(\tau)$ indicates an elite particle at $\tau$th iteration, which is randomly selected from the current personal best particles
pool (i.e., $\{PB_l, l = 1, 2, ..., L\}$) via running the procedure for the Boltzmann selection operator. Instead of the standard social
learning components - only learning from global best particle, there are two advantages in the modified particle-updating
mechanism (Xu et al., 2016). The first is that the intervention of a random elite particle can, to some extent, prevent all
particles from gathering around the global best particle prematurely, enhancing the exploration of the swarm at an early stage

of the search process. The second is that the impact of elite particles on particle updating gradually declines with an increase
in iterations due to the characteristics of the Boltzmann selection operator - the personal best particles close to the global best





one are more likely to be chosen with time, which can guarantee the exploitation of the algorithm in end-stage. Therefore, this approach might be reasonable to balance the global and local search during a PSO process.

On the other hand, for the control of algorithm parameters, an evolutionary state-based adaptive parameter scheme, which is proposed by Zhan et al. (2009), is applied to manage automatically the parameters of the particle-updating equation (Eq. A13a) for each iteration - the inertia weight ($\theta(\tau)$), the personal and social acceleration coefficients ($c_p(\tau)$, $c_s(\tau)$) as well as the ratio for social learning ($\varepsilon(\tau)$). Each parameter is set in the parameter control scheme in terms of a well-defined index that characterizes the current swarm distribution. It is worth mentioning that we use a fuzzy logic system (Jang et al., 1997) to adjust acceleration the coefficients and the ratio for social learning in each generation, and it can increase or decrease them intelligently following four defined evolutionary states - exploration, exploitation, convergence, and jumping out. By automatic control of the algorithm parameters in time, it can improve the search efficiency and convergence speed of the S-APSO.

**Particle check & repair mechanism:** In addition, to avoid a premature convergence of the PSO, it is another crucial factor in successful applications of the PSO to keep all particles feasible during the search process (AP, 2005), especially for this model with the complex solution space induced by integrated with the UWB-SM. Therefore, a check & repair mechanism is developed based on the above coupling strategy, which can examine and fix (if necessary) all updated particles to make sure that they are available. Note that, similar to the particle initialization, there are also many times data exchanges between IGWM-OM and UWB-SM in the process, indicating the reparation in the previous position of a particle might affect that in the subsequent positions, especially in case of fixing the first three positions of a particle that represents GI construction decision variables. It may cause failures of using general check & repair methods.

Therefore, the study uses a multi-round loop structure to check & repair different positions of a particle in terms of the features of the constraints (Eq. A10b), which permits modified particles to be feasible and also to keep the original characters as much as possible. Fig. A2 (C) illustrates the flow chart of the check & repair mechanism, which has a two phases data exchange procedure between the UWB-SM and IGWM-OM for examination and reparation of particles. As reflected in Fig. A2 (C), the first three positions of a particle that indicates GI construction variables in the Eq. A10 are checked and repaired (if necessary) in line with the constraint (7 in A10b) in Phase 1. In Phase 2, the 4 to 51 positions of a particle, which are related the constraints (8 in A10b) - (14 in A10b), are examined and fixed in a multi-round loop structure.

## Appendix B: Details of multiagent system for IGWM at an inter-city scale

This appendix provides a comprehensive and detailed exposition of the multiagent system for IGWM at an inter-city scale. The various components of the system are thoroughly explained, encompassing aspects such as the notation, the application of the Muskingum-Cunge routing equation, and the MAS-UWM framework. Additionally, the appendix delves into the relevant solution approaches associated with this model.

### B1 Notations

To facilitate the model presentation, some of the important notations used hereafter are summarized in Table B1.





**Table B1.** Parameters of the MAS-UWB

| | |
|---|---|
| *Urban area-related parameters* | |
| $A_u^i, A_u^{i+1}$ | Urban total area for UWM agent $i$ and $i+1$ $[m^2]$; |
| *Flow-related parameters* | |
| $Q_{ri}^{i+1}, q_{ri}^{i+1}$ | Upstream river inflow for UWM agent $i+1$ in $[mm]$ and $[m^3]$; |
| $q_{ri}^{i+1}(t), q_{ri}^{i+1}(t-1)$ | Upstream river inflow for UWM agent $i+1$ in month $t$ and $t-1$ $[m^3]$; |
| $Q_{ro}^i, q_{ro}^i$ | River outflow for UWM agent $i$ in $[mm]$ and $[m^3]$; |
| $q_{ro}^i(t), q_{ro}^i(t-1)$ | River outflow for UWM agent $i$ in month $t$ and $t-1$ $[m^3]$; |
| *Model-related parameters* | |
| $f_{r1}^i, f_{r2}^i, f_{r2}^i$ | Coefficient 1, 2 and 3 of the Muskingum-Cunge equation for river reach $i$ $[dimensionless]$. |

## B2   Muskingum-Cunge routing equation

Fig. 1 (B) illustrates the up-and downstream hydrologic interaction between UWM agent $i$ and $i+1$ in the associated river reach. A Muskingum-Cunge routing equation is used to simulate changes in streamflow in the river reach connected with two adjacent urban areas (Garbrecht and Brunner, 1991; Weinmann and Laurenson, 1979). That is, taking the UWM agent $i+1$ in month $t$ as an example (See Fig. 1 B), its upstream inflow in month $t$ can be expressed mathematical by outflow of the UWM agent $i$ in month $t$ and $t+1$, as follows, Fig. 1 (B) illustrates the up-and downstream hydrologic interaction between UWM

agent $i$ and $i+1$ in the associated river reach. A Muskingum-Cunge routing equation is used to simulate changes in streamflow in the river reach connected with two adjacent urban areas (Garbrecht and Brunner, 1991; Weinmann and Laurenson, 1979). That is, taking the UWM agent $i+1$ in month $t$ as an example (See Fig. 1 B), its upstream inflow in month $t$ can be expressed mathematical by outflow of the UWM agent $i$ in month $t$ and $t+1$, as follows,

$$
\begin{cases}
q_{ri}^{i+1}(t) = f_{r1}^i \cdot q_{ro}^i(t) + f_{r2}^i \cdot q_{ro}^i(t-1) + f_{r3}^i \cdot q_{ri}^{i+1}(t-1), & (01) \\
q_{ro}^i(t) = 1000 \cdot A_u^i \cdot [Q_{ro}^i(t)]_*, & (02) \\
q_{ri}^{i+1}(t) = 1000 \cdot A_u^{i+1} \cdot Q_{ri}^{i+1}(t), & (03)
\end{cases}
\tag{B1}
$$

where all parameters of the above notation are listed in Tab. B1 . The first row of Eq. (B1) is the Muskingum-Cunge equation

for calculating the relevant river reach inflow of downstream agents. The second and third rows of Eq. (B1) represent that the units for hydrological parameters were converted from $[mm]$ to $[m^3]$ based on the associated urban total areas. In addition, it should be noted that the Muskingum-Cunge approach is also applicable in the case that there are branches in the main river reach (See UWM agents 1,2 and 3 in Fig. 1 A), which can be solved by dividing the river reach into several sub-reaches based on intersections of the main river and associated branches, and then calculate them in sequence.

## B3   MAS-UWM

The basic assumptions for formulation of the MAS-UWM are shown as follows,

ASSUMPTION 5. *Hydrologically, all UWM agents are interconnected with each other only by a surface water system.*

ASSUMPTION 6. *Socially, all UWM agents are considered to be noncooperative.*





To simplify the model, Assumption 5 demonstrates the hydrologic connection among all UWM agents within the watershed,
which means that all UWM agents have to share surface water resources with others, and IGWM activities of upstream agents
may affect that of downstream. In addition, we do not take the connection between urban areas' groundwater systems into
account since it is assumed that its effect may be negligible in watershed-scale IGWM in the short term, comparing with that
of the surface water system (Brannen et al., 2015). Besides, surface water-groundwater interaction processes induced by city-
scale IGWM have been considered in the UWB-SM, which can, to some extent, reflect the associated effect via the connection
in the surface water system. Assumption 6 exhibits the social relationship between UWM agents in the watershed; that is,
each agent only pays attention to their local objectives and does not share information with the other agents (Giuliani and
Castelletti, 2013). This assumption is reasonable for some watersheds, especially when urban areas within the watershed have
to face competition for urban development in many aspects.

Therefore, the MAS-UWM can be formulated by the integration of the UWM agent model (Eq. A10) with the Muskingum-
Cunge routing model (Eq. B1), depending on its feature of the Markov property. A special type of multi-stage decision system is
employed to model the MAS-UWM (Bellman, 1966) and the sequence of decisions-makings for each UWM agent - city-scale
IGWM - relies on associated spatial locations along with the river networks, which is in order from upstream to downstream.
The hydrologic variable - upstream inflow of each UWM agent - is considered the state variable to describe interactions
between UWM agents. It can be written as follows,

$$
\begin{cases}
\text{Eq. } (A10), & \forall i & (01) \\
\text{Eq. } (B1), & \forall i, t & (02) \\
q_{ri}^1(t) = Q_t^1, \text{ and } q_{ri}^i(0) = Q_0^i, & \forall i, t & (03)
\end{cases}
\tag{B2}
$$

where the third row of Eq. (B2) are initial conditions for the MAS-UWM, and $Q_t^1$ and $Q_0^i$ are the initial amounts of the
upstream inflow for UWM agent 1 in month $t$ and UWM agent $i$ in month 0, respectively.

## B4  Solution approach

It is available to combine multiple S-APSO algorithms with the Muskingum-Cunge routing equation (Eq. B1) to simulate the
dynamics of the MAS-UWM (Eq. B2) according to its Markovian property. That is, the optimal solutions for each UWM
agent model are solved one by one in a specific order, which follows the sequence of the MAS-UWM via using the associated
S-APSO. Notice in particular that the monthly outflow amounts for the optimal solution of each UWM agent model needs
to be recorded during the S-APSO search process. They, as an input of the relevant Muskingum-Cunge equation, are used to
calculate the monthly upstream inflow amounts in the associated downstream reach - an input data for the adjacent UWM agent
model. In this way, the multi-S-APSO framework for simulation of the interactions of the MAS-UWM is developed (See Fig.
B1).





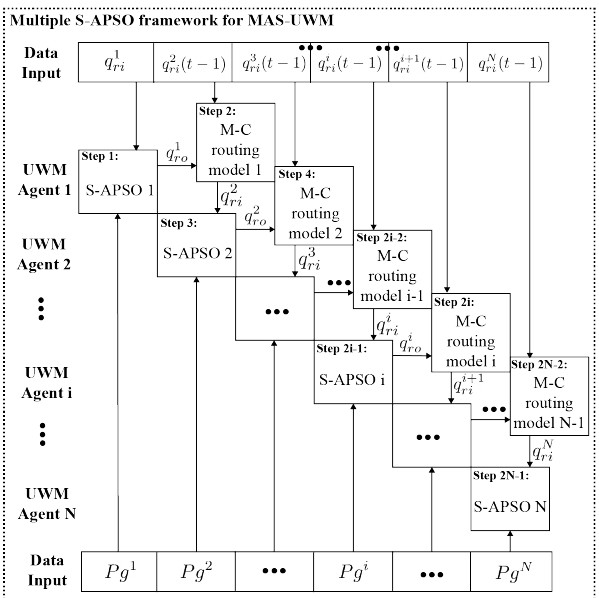

**Figure B1.** Flowchart diagram of the MAS-UWM

## Appendix C: Details of bi-level multiagent system for IGWM at a watershed scale

This appendix presents a thorough and comprehensive examination of the multiagent system for IGWM at a watershed scale. The diverse components of the system are meticulously elucidated, encompassing elements such as the notation, the agent-based model for watershed management, the extended agent-based model for UWM, and the bi-level multiagent system. Furthermore, the appendix explores the pertinent solution approaches associated with this multiagent system.

### C1  Notations

To facilitate the model presentation, some of the important notations used hereafter are summarized in Table C1.

**Table C1.** Parameters of the agent-based model for WM

| | |
|---|---|
| *Objective-related parameters* | |
| $Gini$ | Water allocation Gini coefficient index ; |
| $r_p$ | Penalty rate in the watershed, [$\$/m^3$]; |
| $P_i$ | Total annual penalty fees for the UWM agent $i$ [$\$$]; |
| *Constraints-related parameters* | |
| $SQ_{min}^i(t), SQ_{max}^i(t)$ | Minimum and maximum historical streamflow at checkpoint $i$ in month $t$ [$m^3$]. |





## C2   Agent-based model for watershed manager

The ABM-WM is discussed in detail in the following. The relevant hypotheses for constructing the model are given.

ASSUMPTION 7. *The flow of each urban catchment at the outlet is checked.*

ASSUMPTION 8. *Water withdrawal limit is not considered in the model.*

Assumption 7 simplifies our models as WM set low flow thresholds of each urban area at the outlet. For Assumption 8, computational complexity may be high in optimization of policy portfolio (setting water withdrawal limits and low flow thresholds) for a WM agent model, especially when it is integrated into the MAS-UWM. Also, a policy limiting the water

withdrawal of urban areas, as a mandatory regulation, generally relies on other water users' activities within a watershed, such as agriculture sectors, which is beyond the scope of this study. Therefore, this paper considers water withdrawal limits (i.e., the minimum storage levels of aquifer and river) as UWM agent models' specific parameters (See the 8th and 9th rows of Eq. A10b).

As demonstrated in the above assumptions, a WM agent model is used to describe how a WM set a watershed management

policy - a streamflow penalty strategy - to regulate all UWM agents' decision behavior of IGWM in a watershed. That is, a WM limits water abstraction decisions of each UWM agent in the period via prescribing a series of low streamflow thresholds in associated hydrological stations based on hydrologic conditions; If streamflow in outlet for an urban area is below its threshold, a penalty fee will be imposed on the UWM agent. The strategy, in theory, can force UWM agents to recognize one or more of the externalities caused by IGWM, thereby adjusting their IGWM decisions because it can, to some extent, determine the

cost of IGWM (Baumol et al., 1988). The WM can share fair water resources among urban areas in a watershed by setting a rational streamflow penalty strategy that affects all UWMs' decisions. Therefore, And details of the objective and constraints of the agent-based model for WM are illustrated as follows.

**Equity objective:** The Gini coefficient is widely used to assess resource allocation inequality (Gini, 1921; Nishi et al., 2015). Therefore, the WM agent model uses a water allocation Gini coefficient index proposed by Hu et al. (2016) and Xu

et al. (2019) - the equitable sharing of the used water quantity for each unit of cost - to measure equity of water sharing in watershed-scale IGWM. Based on the definition of the index, to minimize the Gini coefficient means maximal fairness of water resources distributions in a watershed; accordingly, the minimization of the equity objective for the WM can be expressed mathematically as follows,

$$\min_{S_q^i} Gini = \frac{1}{2 \cdot N \cdot \sum_{i=1}^{N} \frac{\sum_{t=1}^{12} [w_s^i(t) + w_g^i(t) + w_{rr}^i(t) + w_{rs}^i(t)]}{TC_i}} \cdot \sum_{i=1}^{N} \sum_{j=1}^{N} \left| \frac{\sum_{t=1}^{12} [w_s^i(t) + w_g^i(t) + w_{rr}^i(t) + w_{rs}^i(t)]}{TC_i} - \frac{\sum_{t=1}^{12} [w_s^j(t) + w_g^j(t) + w_{rr}^j(t) + w_{rs}^j(t)]}{TC_j} \right|. \tag{C1}$$

**Streamflow constraints:** The WM specifies the low streamflow thresholds at each checkpoint, which should adapt to actual

hydrologic conditions. Therefore, the low streamflow thresholds at each checkpoint cannot exceed the associated maximal historical streamflow and cannot be below the minimal one in each month:

$$SQ_{min}^i(t) \le S_q^i \le SQ_{max}^i(t). \quad \forall t, i \tag{C2}$$





In short, the WM agent model is represented as follows:

$$
\min_{S_q^i} Gini = \frac{1}{2 \cdot N \cdot \sum\limits_{i=1}^{N} \frac{\sum\limits_{t=1}^{12} [w_s^i(t) + w_g^i(t) + w_{rr}^i(t) + w_{rs}^i(t)]}{TC_i}} \cdot \sum_{i=1}^{N} \sum_{j=1}^{N} | \frac{\sum\limits_{t=1}^{12} [w_s^i(t) + w_g^i(t) + w_{rr}^i(t) + w_{rs}^i(t)]}{TC_i} - \frac{\sum\limits_{t=1}^{12} [w_s^j(t) + w_g^j(t) + w_{rr}^j(t) + w_{rs}^j(t)]}{TC_j} | \tag{C3a}
$$

$$
s.t. \left\{ SQ_{min}^i(t) \leq S_q^i(t) \leq SQ_{max}^i(t), \quad \forall t, i \right. \tag{C3b}
$$

## C3 Extended agent-based model for UWM

Under the policy intervention from a WM, each UWM agent needs to make reasonable IGWM decisions to trade off the
previous three types of costs (i.e., GIs construction, water supply, wastewater drainage) and the possible penalty fee set by
a WM to minimize their own total IGWM costs under the specified low streamflow thresholds. Therefore, the above UWM
agent model will be extended - its annual IGWM cost function is converted as the sum of GIs construction, water supply,
wastewater drainage costs, and penalty fees. Taking the UWM agent $i$ as an example, the extended UWM agent model for
city-scale IGWM under the streamflow penalty strategy is shown as follows,

$$
\min_{W, GI} TC_i = C_{gi} + C_{si} + C_{wi} + P_i \tag{C4a}
$$

$$
s.t. \left\{ \begin{array}{ll} P_i = \sum_{t=1}^{12} r_p \cdot [\frac{[q_{ro}^i(t) - S_q^i(t)] + |q_{ro}^i(t) - S_q^i(t)|}{2}]; & (01) \\ \text{Eq. } (A10b); & (02) \\ q_{ro}^i(t) = 1000 \cdot A_u^i \cdot [Q_{ro}^i(t)]_*. \quad \forall t & (03) \end{array} \right. \tag{C4b}
$$

where the first row of Eq. (C4b) indicates the annual penalty fee for the UWM agent $i$, and its calculation in detail shown as
follows. The third row of Eq. (C4b) represents that the units for hydrological parameters were converted from $[mm]$ to $[m^3]$
based on the associated urban total areas.

**Penalty fees:** In the extended UWM agent model, a penalty fee must be imposed on UWM agent when the outflow in its
urban catchment is below the specified low streamflow threshold at the corresponding checkpoint. It is assumed that the WM
prescribes a constant penalty rate for the watershed and that a penalty fee is only imposed on out-of-threshold streamflow.
Hence, for UWM agent $i$ in month $t$, if the outflow at checkpoint $i$ is not below the low streamflow threshold (i.e., $q_{ro}^i(t) \geq
S_q^i(t)$), the penalty fee is 0; however, if it is below the quota (i.e., $q_{ro}^i(t) < S_q^i(t)$), a fee equal to $r_p \cdot [S_q^i(t) - q_{ro}^i(t)]$ is imposed.
By integrating the above cases, the annual penalty fee for UWM agent $i$ can be written as:

$$
P_i = \sum_{t=1}^{12} r_p \cdot [\frac{[q_{ro}^i(t) - S_q^i(t)] + |q_{ro}^i(t) - S_q^i(t)|}{2}]. \tag{C5}
$$

## C4 Bi-level multiagent system

As mentioned above, a streamflow penalty strategy prescribed by a WM agent might change some UWM agents' decisions of
IGWM - upstream UWM agents might have to adjust their IGWM decisions to increase outflow to avoid over high penalty fees
for costs minimization, which is beneficial to the downstream agents. Such changes in UWM agents' behavior can, to some
extent, shift the interactions in the MAS-UWM, which might have a potential impact on the watershed environment that can
be measured by the assessment index (i.e., water allocation Gini coefficient) set by the MW (See Fig. 1 (C)). Therefore, a WM





agent can assess the effects of the policy on the watershed via checking the given index that reflects feedbacks of the MAS-UWM and then gradually adjusts it to find the optimal one. This process is a WM-UWM agent interaction in watershed-scale IGWM under a water policy.

Fig. 1 (C) illustrates that the WM-UWM agent interaction is no longer determined only by the WM or the UWMs, and both of them try to optimize their objectives (i.e., equity vs. cost objectives for WM and UWMs) under the associated constraints
(i.e., steamflow vs. GI construction, water supply and demand constraints) and reactions of the other party. Therefore, they follow a specific decision rule. That is, the WM agent first makes a decision, and then each UWM agent specifies a decision to optimize their own objectives with full knowledge of the WM's decision; the WM also optimizes its own objective based on the rational UWMs' reactions. In economic theory, this process - the WM-UWM agent interaction - is a Stackelberg game (Von Stackelberg, 2010).

Therefore, the WM-UWM agent interaction can follow a hierarchical decision rule for the leader - the WM agent and the multiple followers - the UWM agents (Dempe, 2002). Besides, for the followers, the UWM agents form a MAS-UWM (Eq. B1) that has a Markov property, which involves a special multi-stage decision-making process (Bellman, 1966). By integrating the WM (Eq. C3), the UWM (Eq. C4) agent model and the MAS-UWM (Eq. B1) mentioned above, a BL-MAS for WM-UWM agent interaction can be developed to describe the Stackelberg game between the WM and multiple UWM agents, and unique
multi-stage system constructed to reflect the state transitions for the multiple WM-UWM agents, which can be formulated as follows:

$$
\text{Eq. } (C3a)
$$

$$
s.t. \begin{cases} \text{Eq. } (C3b); \\ \text{where } W_i, GI_i \text{ solves} \\ \begin{cases} \text{Eq. } (C4); & \forall i \\ \text{Eq. } (B1); & \forall i,t \\ q_{ri}^1(t) = Q_t^1, \text{ and } q_{ri}^i(0) = Q_0^i. & \forall i,t \end{cases} \end{cases} \tag{C6}
$$

where $[-]_*$ represents that the parameter is from simulating calculation of the UWB-SM.

**C5  Solution approach**

In the BL-MAS (Eq. C6), the UWM agent models are converted to Eq. (C4) due to the introduction of a streamflow penalty
strategy. Compared with the above UWM agent model (A10), only the objective function of the transformed model (C4a) is changed - adding a penalty fee. Therefore, to apply the proposed S-APSO to solve the UWM agent models in the BL-MAS, only the fitness function for particles needs to be adjusted. Besides, the above multi-S-APSO framework is also available in simulating the interactions among all UWM agents in the BL-MAS under a given streamflow penalty strategy because of the features of its hydrologic connections - Markovian property.

For the WM agent model, the above S-APSO framework without the simulation-based initialization and the check & repair mechanism is available to look for the optimal solution due to its simple constraint conditions (C3b). However, there is a





critical factor in the simulation of the BL-MAS that is how to deal with the special decision rule between the WM and the UWN agents - a Stackelberg game, i.e., the WM agents' best response is based on the associated reactions of all UWM agents (Von Stackelberg, 2010). In fact, it is challenging to obtain a Stackelberg solution to the BL-MAS using general solution methods because the bi-level model is an NP-hard problem, even in its simplest linear case (Dempe, 2002). To deal with the specific bi-level model decision rules, the study nests the multi-S-APSO framework for the MAS-UWM mentioned before into the particle performance measurement of the S-APSO for the WM agent, which can simulate the responses of the MAS-UWM to a given streamflow penalty strategy prescribed by the WM agent, thereby assessing the policies' effects accurately. By the nested structure, therefore, a nested S-APSO framework is proposed for searching for the optimal WM-UWN interactions in the BL-MAS under a streamflow penalty strategy. The flowchart diagram for the nested S-APSO is illustrated in Fig. C1 (C).

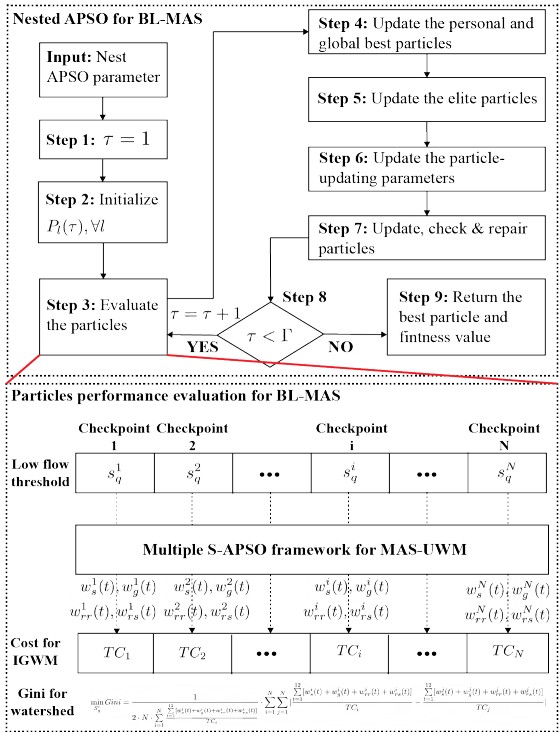

**Figure C1.** Flowchart diagram of the BL-MAS

*Author contributions.* MXZ designed the study, acquired the data, wrote the code, conducted the numerical experiments, analyzed the results, and prepared and revised the paper. TFMC contributed to the design of study framework, supervised the study, validated the results, and revised the paper.



*Competing interests.* The authors declare that they have no conflict of interest.

*Acknowledgements.* We thank the anonymous reviewers and editors for providing valuable suggestions and comments for improving the quality of this paper.



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
