# Peer review of "A Multiagent Socio-hydrologic Framework for Integrated Green Infrastructures and Water Resource Management at Various Spatial Scales"

_Hydrology and Earth System Sciences, 2024_

## Author Comment (AC2)

This manuscript discusses an interesting question in urban water management where green infrastructure is integrated into multiple cities' water supply operations. The authors approach this problem by developing an agent-based modeling (ABM) framework to discuss cross-scale interactions among city and watershed water managers (i.e., agents) and explore water equity and policy implication through imposing a penalty for overdraft.

I appreciate the authors ambition to take on the challenge of solving the complex urban water problem and efforts in developing an integrated modeling tool. The introduction effectively highlights the importance of integrated water management at watershed scale and the need for integrated modeling approaches. I believe the scope of this study will be of great interest to the HESS community. That being said, the current manuscript suffers several major flaws that make it difficult to follow and obscure its contributions and intellectual merits. I am fully committed to helping elevate the quality of this manuscript. If any comments arise from my lack of knowledge on specific points, please accept my apologies in advance. Below are the summaries of my comments and suggestions followed by the specific comments.

My first comment is about the writing. The current manuscript is difficult to follow due to the excessive technical terms and ambiguous language. For example, IGWM (short for integrated green infrastructure and water resources management) was applied in describing models, agents, and agents' decisions, which gave me a headache. Other examples include WM (water manager), UWM (urban water manager), and HUWS (hybrid urban water system). Some technical terms are not well-defined. For example, ABM (agent-based model) and MAS (multi-agent system) are often used interchangeably in the literature, but it was presented in this manuscript as two distinct modeling approaches applied for building two models (i.e., city-scale and inter-city). Similarly, rainwater and stormwater are the same thing to me, and yet they are listed as two water sources (lines 183-184). In the results section, the authors discuss the usage of the four water sources (surface water, groundwater, stormwater, and rainwater), so the model must have simulated the water sources. However, I could not find their definitions nor how the water supply portfolio is simulated except for surface water. I will recommend a more rigorous quality control and assurance to improve the flow and readability.

**Response:**

Thank you for your thorough review and valuable feedback on our manuscript. We understand your concerns regarding the clarity and readability of the text, particularly related to the use of technical terms and acronyms. We apologize for any confusion caused by the excessive use of acronyms and the lack of clear definitions for some key concepts.

To address the issues you raised, we propose the following revisions:

1. **Reduction of Acronyms**: We will reduce the number of acronyms by eliminating those that are infrequently used. Additionally, we will include a comprehensive list of acronyms at the beginning of the manuscript to assist readers in understanding the terms used throughout the text.

2. **Clarification of Key Concepts**: We will add clear definitions for key concepts that were omitted in the current version of the manuscript. For instance, we will provide explicit definitions for rainwater and stormwater. As referenced in studies by Khan et al. (2023) and Fielding et al. (2015), stormwater is the water that drains off land areas from rainfall, including water from rooftops, ground surfaces, and other areas. In contrast, rainwater refers specifically to the rain that falls on roofs and can be collected into storage tanks before contacting the ground, resulting in higher quality due to fewer contaminants. The distinction between these two sources is crucial as it impacts their respective urban water cycle and costs of water use (MSSC, 2008), which is why both are considered in our model framework.

3. **Detailed Explanation of Simulation Processes**: We will enhance the methodology and results sections with additional details on the simulation processes, particularly how the water supply portfolio is simulated within our proposed model framework. Although these details are included in the appendix, we will ensure that key points are clearly presented in the main sections of the manuscript to improve readability and comprehension.

**References**

Khan, A., Park, Y., Park, J., Sim, I. and Kim, R., 2023. Analysis of Stormwater and Rainwater Harvesting Potential Based on a Daily Water Balance Model: A Case Study of Korea. Water, 16(1), p.96.

Fielding, K.S., Gardner, J., Leviston, Z. and Price, J., 2015. Comparing public perceptions of alternative water sources for potable use: The case of rainwater, stormwater, desalinated water, and recycled water. Water Resources Management, 29, pp.4501-4518.

Minnesota Stormwater Steering Committee (MSSC)., 2008. State of Minnesota Stormwater Manual: Version 2. Minnesota Pollution Control Agency, St. Paul, MN. https://stormwater.pca.state.mn.us/index.php/Overview_for_stormwater_and_rainwater_harvest_and_use/reuse

My second major comment is about the framing of the model. After reading the method section a few times, the modeling components become clear to me. The modeling framework includes three models coupling together. However, it is essentially one agent-based model with two agent types (city agents, UWM, and a watershed agent, WM) and a hydrologic model (including UWB-SM and M-C) representing the spatial connections among the city agents and the watershed environment. I can understand the authors' intention in examining the interactions among agents across multiple sales; however, framing the models separately at different scales has had the opposite effect for me, leaving me confused and obscuring my understanding of the study. The

suggestion here may be somehow subjective, but I am hoping that the manuscript could benefit from my perspective.

**Response:**

Thank you for your insightful comments on the framing of our model. We appreciate your efforts in thoroughly reviewing the methodology section and providing valuable feedback.

The primary reason for framing our models separately at different scales is to explore the role of Green Infrastructures (GIs) in water resources management from the city scale to the watershed scale. Our intention was to simulate the socio-hydrologic interactions driven by the introduction of GIs within multiple agent systems. We believe that the impacts of introducing GIs for rainwater use in urban water resources systems are not only local (city-scale) but also overall (watershed-scale) due to the social and hydrologic connections between urban areas. Therefore, as described in the methodology section, we constructed our model from the city scale to the inter-city scale to the watershed scale.

However, based on your feedback, we understand that the current presentation of our model framework may be confusing to some readers. To address this, we propose the following revisions:

1. **Introduction of the Model Framework**: We will add a paragraph at the beginning of the methodology section to briefly introduce the entire framework of our models, the corresponding components, and their relationships. This will provide readers with a clear overview of the model structure and its components, facilitating a better understanding of the overall framework.

2. **Improvement of Model Details**: We will enhance the subsequent parts of the methodology section to ensure coherence, cohesiveness, and consistency. Specifically: a. We will highlight and explain the relationships between models at different scales, making it easier for readers to understand the underlying logic of framing the models separately at different scales. b. We will delete redundant and overlapping content between the parts detailing models at different scales, making the manuscript more readable and straightforward.

Generally, I found it difficult to follow the results and discussion, partly attributed to not fully understanding the modeling components. Since I did not go through all the details in the Appendices, I was not sure whether I could agree or disagree with the findings and discussion. I will suggest presenting the key components of the models in the main text. For example, in lines 297 – 298, the authors mentioned an assessment index (Gini coefficient) set by the WM agent but did not go any further to explain how it was incorporated into WM's decision-making nor describe what the Gini coefficient means and how it is calculated. I feel the results and discussion can be condensed to focus on key findings as a long discussion would lose its audience. Another suggestion is to

provide more details of the urban water balance model (UWB-SM) in the main text as the model of the physical environment since it is where the water partition is determined. Contrarily, the texts related to the routing model (M-C method) and solution approach (S-APSO) can be moved to the Appendix. Is the S-APSO approach the original creation of this work? If so, I think the solution approach as well as the UWM model could be a separate paper.

**Response:**

Thank you for your constructive comments and suggestions on our manuscript. We apologize for any confusion caused by the presentation of our results and discussion sections. We appreciate your detailed feedback and propose the following revisions to address your concerns:

**Key Model Components in the Main Text**: We acknowledge that the detailed explanations and associated calculation equations for the assessment index (Gini coefficient) and the urban water balance model (UWB-SM) are currently located in the appendix (see lines 1018-1024 and lines 695-815, respectively). To improve clarity, we will reorganize and rewrite the methodology and corresponding appendix sections. Specifically:

1. We will present the some technical details for key components of the models, including the urban water balance model, in the main text to provide a clearer understanding of the modeling framework.

2. We will move the minor components of the model, such as the routing model (M-C method) and the solution approach (S-APSO), to the appendix. Although the S-APSO approach is an original creation of this work, designed specifically to solve the proposed model framework, we will consider your suggestion to further develop this approach and potentially publish it as a separate paper in the future.

**Clarification of the Gini Coefficient**: We will provide a detailed explanation of the Gini coefficient, including its meaning, how it is calculated, and how it is incorporated into the WM agent's decision-making process, within the main text. This will help readers understand the relevance of this index to our study.

**Results and Discussion**: In response to your comments, we will rewrite and rearrange the Results and Discussion section. We will condense the discussion to focus on key findings and remove any redundant or unimportant parts to make the discussion more focused and easier to understand. We believe these changes will enhance the discussion and provide clearer and more actionable insights for the readers.

Overall, this manuscript has the potential to be a high-quality paper (by the modeling framework itself) if the authors can improve the clarity in the methodology, experiment designs, and discussion and highlight its contributions.

**Specific Comments**

- Lines 27-34: The introduction highlights the need for multi-scale green infrastructure frameworks in urban water management. The introduction needs to provide a detailed positioning within recent literature and how the current research contributes to the body of knowledge. Integrating findings from recent studies on similar frameworks could help contextualize their suggested approach within the broader field and clarify its unique contributions. Suggestion: Expand the literature review to include recent ABM applications in socio-hydrology and water resources.

**Response:**

Thank you for your valuable suggestion regarding the introduction section of our manuscript. We agree with your opinion that the introduction should provide a detailed positioning within recent literature and clearly demonstrate how the current research contributes to the body of knowledge. In the current version of the introduction, we have referenced some previous studies on multi-scale green infrastructure frameworks in urban water management, particularly IGWM at the city scale (lines 82-103), inter-city scale (lines 104-124), and watershed scale (lines 125-138). We also analyzed their contributions and identified gaps. However, we acknowledge that the literature review may be insufficient and its current positioning may not effectively highlight the research problem, motivation, and gaps addressed by our study. To address your comments, we will undertake the following revisions:

1. **Expand the Literature Review**: We will expand the literature review to include recent applications of agent-based models (ABM) in socio-hydrology and water resources management. This will help contextualize our suggested approach within the broader field and clarify its unique contributions.
2. **Rearrange the Literature Review**: We will re-arrange the location of the literature review within the introduction to ensure a logical flow. This will help readers better understand the research problem, motivation, and gaps that our study aims to address.

- Line 15–20: Add brief mention of the specific experimental scenarios (e.g., "streamflow penalty" and GI adoption) to give readers a clearer picture of the paper's approach and key findings at the outset. This will make the abstract more informative for readers skimming the content.

**Response:**

Thank you for your insightful suggestion regarding the introduction section of our manuscript. To address your comment, we will revise the beginning of the introduction section to include brief descriptions of these specific experimental scenarios. This will help

set the stage for our research and make the introduction more informative and engaging for readers.

- Line 35–40: The statement on the importance of GIs could be made more impactful by adding specific challenges (e.g., "urban flooding, groundwater depletion, and inter-city water conflicts") that this framework aims to address. This would help sharpen the focus on the practical problems the model intends to resolve.

**Response:**

Thank you for your valuable suggestion regarding the statement on the importance of green infrastructures (GIs) in our manuscript. To address your comment, we will revise the relevant section to include specific challenges such as inter-city water conflicts and groundwater depletion. This will demonstrate the importance of GIs more effectively and highlight the practical significance of our framework.

- Ensure that acronyms such as "GI" (for Green Infrastructure) and "UWM" (for Urban Water Manager) are consistently defined and used throughout the text. For instance, Line 42 introduces GI without explicitly defining it, which may confuse readers unfamiliar with the abbreviation.

**Response:**

Thank you for your helpful suggestion regarding the use of acronyms in our manuscript. To address your comment, we will undertake the following actions:

1. **Review and Modify Acronyms**: We will carefully review the manuscript to ensure that all acronyms, such as "GI" for Green Infrastructure and "UWM" for Urban Water Manager, are explicitly defined when first introduced and consistently used throughout the text.

2. **Reduce the Number of Acronyms**: We will reduce the number of acronyms by eliminating those that are used infrequently, thereby simplifying the text and reducing potential confusion.

3. **Add a Comprehensive List of Acronyms**: We will include a comprehensive list of acronyms in the manuscript to help readers easily understand the terms used.

- Specific terms, such as "hydrologic regime" and "multiagent system," are used inconsistently. A brief definition of these terms early in the manuscript (in the Introduction or Methods) would improve consistency.

**Response:**

Thank you for your valuable suggestion regarding the use of specific terms in our manuscript. To address your comment, we will take the following actions:

1. **Review and Ensure Consistency**: We will review the manuscript to ensure that all specific terms are used consistently throughout the text.

2. **Provide Definitions**: We will include brief definitions of these terms in the Introduction and Methods sections to enhance understanding and consistency for the readers.

- The methodology section presents a layered framework with urban and watershed scales involving socio-economic and hydrologic variables. However, the description of how these scales is integrated within a complex system would benefit from additional detail and clarity.

**Response:**

Thank you for your insightful suggestion regarding the methodology section of our manuscript. To address your comment, we will take the following actions:

1. **Add an Overview Paragraph**: We will add a paragraph at the beginning of the methodology section to briefly introduce the entire framework of our models, including the corresponding components and their relationships. This paragraph will focus on the interactions and relationships between models at different scales and how these local-scale models are integrated within a complex system.

2. **Enhance Detail and Clarity**: We will improve the subsequent parts of the methodology section to make the descriptions of models at different scales more coherent, cohesive, and consistent. Specifically, we will highlight and explain the relationships between models at different scales, making it easier for readers to understand the underlying logic of framing the models separately at different scales.

- Lines 170-182: The agent-based modeling (ABM) setup could be explained more systematically. Clarifying the assumptions behind each agent's decision-making process, especially for UWMs and watershed managers, would make the model's structure more understandable. Additionally, line 175 references the "Markov property," but a brief explanation or contextualization within the model would benefit readers unfamiliar with this concept.

**Response:**

Thank you for your valuable suggestion regarding the agent-based modeling (ABM) setup in our manuscript. To address your comment, we will take the following actions:

1. **Systematic Explanation of ABM Setup:** We will add a paragraph to briefly introduce the entire framework of our models and the corresponding components at the beginning of the methodology section. This will include a more systematic explanation of the assumptions behind the decision-making processes of different agents, including UWMs and watershed managers. Although these assumptions are detailed in the Appendix (see lines 817-827; lines 961-973; lines 999-1009), we will extract key assumptions and place them in appropriate positions within the methodology section to make the model's structure more understandable.

2. **Clarification of the Markov Property:** While a brief explanation of the Markov property is provided in the introduction section (see lines 63-65), we will further improve this explanation. We will also re-arrange this explanation within the methodology section to ensure that readers unfamiliar with this concept can easily understand its relevance and application within our model.

- Line 250 briefly mentions historical hydrologic data without indicating the data sources, calibration metrics, or validation techniques. Include a clear explanation of the calibration and validation processes. A summary table with parameter ranges, calibration techniques, and validation outcomes would strengthen the model's reliability and replicability.

**Response:**

Thank you for your insightful suggestion regarding the hydrologic data used for calibration and validation of the hydrologic model components in our framework. To address your comment, we will take the following actions:

1. **Clarify Data Sources**: We have mentioned the data sources in Section 3.3 - Data Collection and Processing (lines 409-410). We will ensure that this information is clearly stated and easily accessible to the reader.

2. **Detail Calibration and Validation Techniques**: We have discussed the calibration metrics and validation techniques in Section 3.4 - Model and Algorithm Setup (lines 437-447). We will further elaborate on these processes to provide a clearer understanding.

3. **Include a Summary Table**: We agree that a summary table with parameter ranges, calibration and validation techniques and processes would greatly enhance the clarity and comprehensibility of our model. Although some details are provided in the Section 3.4 and Appendix, we will add a comprehensive summary table in the methodology section. This table will include the parameter ranges, data sources, and calibration and validation techniques for the hydrologic models, making it easier for readers to understand the model details.

- Line 280–285: The hydrologic and socio-economic data sources description is somewhat broad. Including a brief list of specific datasets used, such as U.S. Geological Survey data or climate records, and their date ranges would clarify the model's foundation. The manuscript presents three spatial scales (city, inter-city, and watershed) for experimental analysis, focusing on GI policies. However, these scenarios are presented with minimal contextual detail.

**Response:**

Thank you for your valuable suggestion regarding the description of hydrologic and socio-economic data sources, as well as the contextual detail of our experimental design. To address your comment, we will take the following actions:

1. **Enhance Data Sources Description**: We will rewrite Section 3.3 - Data Collection and Processing to include more detailed information about the data sources. This will involve listing specific datasets used, such as U.S. Geological Survey data and climate records. Additionally, we will provide basic information about the selected gauge and weather stations, as well as details on data ranges and data processing methods.

2. **Improve Experimental Design Description**: We will enhance Section 3.2 - Experimental Design by adding more details about the experiments. This will include the purpose of the experiments, the methods used, the settings of key experiment parameters, and the evaluation metrics for the experimental results. We will ensure that the scenarios involving the three spatial scales (city, inter-city, and watershed) and their focus on GI policies are presented with sufficient contextual detail.

- Lines 315-327: Discussing the experimental conditions would help explain why specific scenarios were chosen, such as the "streamflow penalty" policy in line 319. A description of how this penalty reflects real-world practices would better convey the practical relevance of this scenario.

**Response:**

Thank you for your insightful suggestion regarding the discussion of experimental conditions in our manuscript. To address your comment, we will take the following actions:

1. **Add Detailed Discussion of Experimental Conditions**: We will expand Section 3.2 - Experimental Design to include a more thorough discussion of the experimental conditions. This will involve explaining the rationale behind selecting specific scenarios. For example, we will describe how this penalty reflects real-world practices and provide key experimental parameter settings, such as the base penalty rate.

2. **Motivation and Mechanism Analysis**: We will also include an analysis and description of the motivation and underlying mechanisms for choosing these specific scenarios. This will help convey the practical relevance and importance of the

scenarios in the context of integrated green infrastructures and water resource management.

- Lines 390-420: This section would be more accessible if the results for each spatial scale (city, inter-city, watershed) were divided into distinct subsections rather than being presented together. This would help readers understand the unique impacts observed at each scale.

**Response:**

Thank you for your valuable suggestion regarding the presentation of results for each spatial scale in our manuscript. To address your comment, we will take the following actions:

1. **Division into Distinct Subsections**: We will ensure that Section 4 - Results and Discussion is clearly divided into three distinct subsections: Subsection 4.1 for the city-scale model, Subsection 4.2 for the inter-city scale model, and Subsection 4.3 for the watershed-scale model. Each subsection will focus on presenting and discussing the results specific to that spatial scale.
2. **Prioritize Unique Impacts**: In each subsection, we will prioritize showing and analyzing the results of the model at the corresponding scale. This will help readers clearly understand the unique impacts observed at each spatial scale.
3. **Discuss Relationships Between Scales**: After presenting the results for each scale, we will discuss the relationships between models at smaller and larger scales. This discussion will include results from different scales to illustrate how they interrelate, but we will ensure this is done in a way that does not obscure the unique impacts at each scale.

- Lines 460-475: While the discussion briefly mentions the potential impacts of GI policies, it could provide more concrete suggestions for policymakers, especially regarding implementing penalty-based policies. For instance, specifying how such policies could be enforced across jurisdictions or considering potential limitations would strengthen the section.

**Response:**

Thank you for your insightful suggestion regarding the discussion of potential impacts of GIs policies. To address your comment, we will take the following actions:

1. **Enhance Policy Suggestions**: We will add one or two paragraphs in Subsection 4.3 - Impacts of Water Policy on Watershed-Scale IGWM. These paragraphs will provide concrete suggestions for policymakers based on the results and analysis of our watershed-scale model.

2. **Discuss Enforcement and Limitations**: We will include a discussion on how penalty-based policies could be enforced across different jurisdictions. Additionally, we will address potential limitations of such policies, considering practical aspects and challenges in implementation.

- Lines 490-500: This discussion could explore the model's adaptability to other similar regions or hydroclimatic conditions and cross-case comparisons. The authors could broaden the study's relevance by highlighting how it might apply to other areas. Expand the discussion on the policy implications of GIs, considering practical challenges and enforcement strategies. Including recommendations for policymakers, such as adaptive management guidelines or climate-resilient infrastructure planning, would enhance the study's applicability.

**Response:**

Thank you for your constructive suggestion regarding the exploration of our model's adaptability to other regions and hydroclimatic conditions, as well as the expansion of policy implications. To address your comment, we will take the following actions:

1. **Expand Discussion on Model Adaptability**: We will add a relevant discussion about the adaptability of the model at different scales to other regions in Subsections 4.1, 4.2, and 4.3 of Section 4 - Results and Discussion. This discussion will include: a) The possibility and operability of applying the proposed model framework to other regions with similar hydroclimatic conditions. c) Further analysis of the potential effects of GIs on watersheds similar to our study area. b) The potential for extending our model framework to simulate GI-driven socio-hydrology dynamics in other watersheds under different water policies, such as water trading schemes (Eheart and Lyon, 1983).
2. **Broaden Policy Implications**: We will expand the discussion on the policy implications of GIs, considering practical challenges and enforcement strategies. Specifically, we will include: a) Recommendations for policymakers, such as adaptive management guidelines and climate-resilient infrastructure planning. b) An analysis of practical challenges and strategies for enforcing GI policies across different jurisdictions.

**References**

Eheart, J.W. and Lyon, R.M., 1983. Alternative structures for water rights markets. Water Resources Research, 19(4), pp.887-894.

- Lines 520-530: The conclusion summarizes the key findings well but could further emphasize the study's contributions and the potential for broader application. Highlight how the study advances the field of socio-hydrologic

modeling, specifically regarding multi-agent frameworks for GI integration. A concluding sentence on how this framework could guide future studies in water management would leave a stronger impression.

**Response:**

Thank you for your valuable suggestion regarding the conclusion section of our manuscript. To address your comment, we will take the following actions:

1. **Enhance the Conclusion**: We will improve the conclusion section to further emphasize our study's contributions, specifically highlighting how the proposed multi-agent socio-hydrologic framework advances the field of socio-hydrologic modeling and the integration of GIs.
2. **Broader Application**: We will discuss the potential for broader application of our model framework to other watersheds, focusing on urban and watershed water management scenarios.

**Technical corrections**

- Minor grammatical issues and ambiguous phrases appear throughout the text. For example, line 175, "up-and downstream imbalances," could be clarified as "upstream-downstream imbalances." A thorough proofreading would enhance readability.

**Response:**

Thank you for your careful review and for pointing out the minor grammatical issues and ambiguous phrases. We appreciate your attention to detail. We will make the following revisions to address your comments:

Line 175: Change "up-and downstream imbalances" to "upstream-downstream imbalances."

In addition to this correction, we will thoroughly proofread the entire manuscript to identify and correct any similar minor errors and ambiguous phrases.

- Figure 1 provides a schematic of the model, but its components and interconnections need to be labeled clearly. A legend or detailed figure description indicating each component's function within the model would enhance interpretability. Improvement Suggestion: Include a step-by-step description or flowchart illustrating the interactions between socio-economic factors, hydrologic processes, and policy influences. This would help in clarifying the multi-agent interactions and coupling between scales.

**Response:**

Thank you for your valuable suggestion regarding Figure 1. To address your comment, we will take the following actions: We will add detailed descriptions for each component in Figure 1, clearly labeling all elements and their interconnections. This will make the figure easier to understand.

Besides, we will include a step-by-step flowchart illustrating the interactions between socio-economic factors, hydrologic processes, and policy influences.

- Figures 2 and 5: These figures would benefit from concise captions that specify what variables or trends they are intended to show. For instance, state whether they display policy impacts, flow distributions, or demand-supply imbalances explicitly.

**Response:**

Thank you for your insightful suggestion regarding the captions of Figures 2 and 5. To address your comment, we will take the following actions: We will expand the captions of Figures 2 and 5, as well as other figures associated with the results demonstration of our models, to explicitly specify what variables or trends they are intended to show. This will include clearly stating whether the figures display policy impacts, flow distributions, demand-supply imbalances, or other relevant information.

- Figure 3: While Figure 3 illustrates IGWM patterns for UWMs, more context on the visualized policy implications and the decision-making dynamics among UWMs would make the figures more impactful. Additionally, increasing the color contrast between scenarios in this figure would improve readability. Improvement Suggestion: Provide a rationale for each experimental scenario, focusing on its real-world applications. Consider adding a flowchart or table summarizing the experimental setups and their objectives to help readers follow the study's design.

**Response:**

Thank you for your valuable suggestion regarding Figure 3. To address your comment, we will take the following actions:

1. **Context and Dynamics**: We will add more context on the visualized policy implications and the decision-making dynamics among UWMs to Figure 3. This will help readers better understand the IGWM patterns and their significance.

2. **Color Contrast**: We will increase the color contrast between scenarios in the figure to improve readability and distinguishability.

Additionally, we will include a table summarizing the experimental setups and their objectives in this section. This table will help readers follow the study's design and understand the rationale behind the experimental scenario.

- Figure 4: The information presented in Figure 4 lacks sufficient labeling to identify different variables. Clear labels or a more detailed caption would clarify how the results vary by scale and policy. Improvement Suggestion: Reorganize the results section by scale and add clear interpretations of findings about policy scenarios. Additional labels and color coding in figures, especially Figures 3 and 4, would improve clarity. Why are the lines zigzagging?

**Response:**

Thank you for your insightful comments regarding Figure 4. We appreciate your suggestions to enhance the clarity and interpretability of our figures and results section. To address your comments, we will take the following actions:

1. **Labels and Captions**: We will add clear labels to identify different variables in Figure 4 and other figures associated with the results demonstration of our models. Additionally, we will provide more detailed captions to clarify how the results vary by scale and policy.
2. **Reorganization**: We will reorganize the results section by scale, providing clear interpretations of findings related to different policy scenarios. This reorganization will help readers follow the study's design and understand the implications of our findings.
3. **Trend Analysis**: We will further analyze and explain the trends and features observed in the figures, such as the zigzagging line in Figure 4b. This analysis will provide additional context and clarity for the presented results.
4. **Color Coding**: We will incorporate additional labels and color coding in Figures 3 and 4 to improve clarity and distinguishability of different scenarios and variables.

- Figures 3 and 4: Improve color contrast and include labels for specific variables to make visualizations easier to interpret. Including a note explaining the data or variables displayed in each figure would enhance accessibility.

**Response:**

Thank you for your valuable suggestions regarding Figures 3 and 4. To address your comment, we will take the following actions:

1. **Add Labels**: We will add clear labels to Figures 3 and 4 to identify different variables.

2. **Increase Color Contrast**: We will increase the color contrast between scenarios in Figures 3 and 4.

3. **Detailed Captions**: We will include detailed captions for Figures 3 and 4, providing explanations of the data or variables displayed.

Besides, We will also apply these improvements to other figures associated with the results demonstration of our models to ensure consistency and clarity throughout the paper.

- Line 367: Typo 'experiment'

**Response:**

Thank you for your careful review and for pointing out the typo on line 367. We appreciate your attention to detail. We will correct the typo by changing "experience" to "experiment".

Additionally, we will thoroughly review the entire manuscript to identify and correct any similar minor errors.

- Line 381: What are $r_{imax}$, $r_{rmax}$, and $r_{smax}$?

**Response:**

Thank you for your question regarding the parameters $r_{imax}$, $r_{rmax}$, and $r_{smax}$ mentioned on line 381. To clarify:

- $r_{imax}$ represents the maximum ratio of the area constructed with infiltration-based GIs to the relevant surface area.

- $r_{rmax}$ represents the maximum ratio of the area constructed with rainwater harvesting systems to the relevant surface area.

- $r_{smax}$ represents the maximum ratio of the area constructed with stormwater harvesting systems to the relevant surface area.

All of these parameters are part of the urban water balance simulation model (UWB-SM). By setting these parameters to zero, we can simulate scenarios without GI development using the UWM agent model.

While a detail explanation of these parameters is provided in the Appendix, we will rewrite this section in the manuscript to make it clearer and easier for readers to understand.

- Line 430: References?

**Response:**

Thank you for pointing out the omission of references on line 430. We apologize for this oversight. We will add the relevant references to this part of the section to ensure proper citation and to support the statements made.

---

## Author Response (AR1)

Dear Editor and Reviewers,

First of all, we deeply appreciate for all your valuable and insightful suggestions to the paper titled "*A Multiagent Socio-hydrologic Framework for Integrated Green Infrastructures and Water Resource Management at Various Spatial Scales*". We have checked the paper carefully and made revisions as reviewers suggested and also spent some time in checking the English writing and grammar. In addition, we have further carefully edited the manuscript to eliminate the shortcomings. All the revisions will be showed in the list as follows.

If you have any question about this paper, please don't hesitate to let me know. Your acknowledgement will be highly appreciated.

Sincerely yours, Mengxiang Zhang & Ting Fong May Chui

Department of Civil Engineering, The University of Hong Kong, Hong Kong SAR, China

Corresponding Author: Dr. Ting Fong May Chui Tel: +852-22194687; Fax: +852-25595337. E-mail: maychui@hku.hk

**Response to Reviewers #1**

First of all, thank you very much for your arduous and excellent work on our paper. Your comments for the paper would take you a lot of energy, which is undoubtedly of great significance to improve the quality of the paper. We have tried to revise the paper thoroughly following your instruction, and according to your other helpful suggestions, we have made revisions as following:

The overall paper is an important contribution to the science of decision-making with GIs across spatial scales. There are a few conceptual questions I had that I was not able to follow in the paper, but it appears the work is there. A bit further clarification would be helpful, and in that case, I would accept the manuscript for publication.

Please see below a few comments:

• The grammar and writing style is excellent. I do not have any major comments on the technical writing. However, while I understand the rationale for doing so, the many use of acronyms reads to me as confusing. Several of these acronyms mean essentially the same thing at the decision-making scale. Perhaps consider condensing the number of acronyms if feasible? Not necessary, it just was hard for me to follow. I see the graphic in Figure 1, which does help explain this concept a bit across the 3 spatial scales, but it is still hard to follow when reading the introduction. Perhaps re-state the acronyms in Fig. 1 caption and use all in the graphic? (e.g., UWB, SM are missing).

**Response:**

Thank you for your positive feedback on the grammar and writing style of our manuscript. We appreciate your insightful comments regarding the use of acronyms, which we understand can be confusing when overused.

To address your concerns, we will take the following actions:

- 1. We have added a comprehensive list of acronyms to the manuscript to help readers understand the terms used (See Lines 742-746).
- 2. We have reduced the number of acronyms by eliminating those that are infrequently used in the previous manuscript, such as HUWS, MAS, MAS-UWM, BL-MAS, ABM, ABM-UWM, ABM-WM, S-APSO and IGWM-OM.
- 3. We have added a detailed explanation of Figure 1 and restated all remaining acronyms within the figure's caption to enhance clarity and comprehension. Please refer to Page 4, Figure 1 for the updated content.

These revisions can be incorporated to enhance the readability of the introduction and provide a clearer understanding of the concepts across the three spatial scales.

- Very minor corrections noted here:
  - Line 116: "a agent-based framework" should be "an agent-based framework"
  - o Line 159: "dynamic(s) of (the) watershed"?
  - Figure 1: Should the text near WM Agent state "Bi-level multiagent system"?
  - Line 578: Period instead of ;?

**Response:**

Thank you for your careful review and for pointing out these minor corrections. We appreciate your attention to detail. We have made the following revisions:

- Line 131: Change "a agent-based framework" to "an agent-based framework".
- Line 175: Change "dynamic of watershed" to "dynamics of the watershed".
- Figure 1: Update the text near WM Agent to "Bi-level multiagent system".
- Line 612: Replace the semicolon with a period.

In addition to these corrections, we also have thoroughly reviewed the manuscript to avoid similar minor errors.

• The socio-hydrological application at various spatial scales applies to GIs in any urban community, not just alongside rivers. Perhaps consider re-phrasing the references to how GIs interact with hydrology near river networks.

**Response:**

Thank you for your insightful comment. You are correct that the socio-hydrological application at various spatial scales applies to green infrastructures (GIs) in any urban community, not just those alongside rivers. The distribution of green and blue water throughout a watershed can change with the development of GIs, which collect and use rainwater both directly and indirectly. For example, GIs can increase groundwater recharge (Zhang and Chui, 2019) and evapotranspiration (Ebrahimian et al., 2019), while also potentially altering urban water use patterns and subsequently urban and watershed hydrology (Pennino et al., 2016; Chen et al., 2019).

We have add explanations and relevant references to the introduction section to clarify how GIs interact with hydrology beyond river networks. (See Line 60 - 63).

The primary reasons for our initial focus on river connections are twofold: 1) River connections are a significant factor in the effect of GIs on urban and watershed hydrology concerning water resource allocation, and 2) Including other hydrological connections driven

by GIs into the multi-spatial scale system introduces complexity that can be challenging to address comprehensively. We have included additional explanations regarding these considerations in the model assumptions section of the methodology (See Lines 303 - 304) and relevant appendix sections (See Lines 1003 -1012).

Moreover, the multiagent socio-hydrologic framework we proposed is designed to be flexible. It is feasible to incorporate additional hydrological connections driven by GIs within the framework to provide a more comprehensive description of the interaction between GIs and hydrology in the watershed. We have added corresponding explanations in the conclusion and future research sections (See Lines 718 - 724).

**References**

Zhang, K. and Chui, T.F.M., 2019. A review on implementing infiltration-based green infrastructure in shallow groundwater environments: Challenges, approaches, and progress. *Journal of Hydrology*, *579*, p.124089.
Ebrahimian, A., Wadzuk, B. and Traver, R., 2019. Evapotranspiration in green stormwater infrastructure systems. *Science of the total environment*, *688*, pp.797-810.
Pennino, M.J., McDonald, R.I. and Jaffe, P.R., 2016. Watershed-scale impacts of stormwater green infrastructure on hydrology, nutrient fluxes, and combined sewer overflows in the mid-Atlantic region. *Science of the Total Environment*, *565*, pp.1044-1053.
Chen, J., Liu, Y., Gitau, M.W., Engel, B.A., Flanagan, D.C. and Harbor, J.M., 2019. Evaluation of the effectiveness of green infrastructure on hydrology and water quality in a combined sewer overflow community. *Science of the Total Environment*, *665*, pp.69-79.

 Most of the mentioning of the "socio" part of GIs being part of a sociohydrological system in the introduction refer to housing types or anthropogenic activities in upstream portions of the drainage basin. While this is true, the socio component of GI systems extends far beyond these considerations and might be worth a mention. For example, how GIs become community recreational meeting spots, have been shown to reduce human health issues, improve mental well-being, provide urban sources of food, improve heat island effects, reduce noise, etc. Or, are you mostly referring to the "socio" component being the complex decision-making required? This was a bit confusing to me.

**Response:**

Thank you for your valuable comment. We appreciate your suggestion to expand on the "socio" component of green infrastructure (GI) systems beyond housing types and anthropogenic activities in upstream portions of the drainage basin.

In our decision-making framework, we primarily focus on the dynamics of socioeconomic relationships related to water use and conflict driven by the introduction of GIs in urban and watershed water resource allocation. This is an important social issue associated with balancing cost with the equity of water use rights among various water use agents (urban cities) within a watershed (Baumol, 1988; Fisher, 1981).

Specifically, on a city scale, the development of GIs can reduce the cost of accessing water resources. GIs can enrich urban water users' choices and gradually change their water use habits. In general, water demand might increase as the cost of water use decreases (water supply-demand cycle effect – Kallis, 2010).

However, on a watershed scale, due to geographic location effects (i.e., upstream and downstream conflicts) and GI introduction, although the overall cost of water use for each city decreases compared to the scenario without GIs, the degree of cost reduction varies between cities. Upstream cities experience a higher reduction in water use costs compared to downstream cities, leading to inequity in water resources distributions among urban areas and potentially worsening water conflicts among urban areas.

Therefore, while the introduction of GIs like a types of new technology, does reduce the cost of water resources use at a local level, it can also exacerbate inequity in water use among different regions within the watershed without appropriate policy interventions (Kristal and Cohen, 2017). To address this conflict between cost and equity of water use driven by GIs, we propose the inclusion of a watershed manager agent in the IGWM framework. This agent would guide urban water managers' decisions by implementing a water management policy, such as a streamflow penalty strategy. This solution aims to ensure that each city can access water resources equitably and at a relatively low cost by developing GIs to use rainwater directly and indirectly, thereby promoting stable development within the watershed.

We acknowledge that the "socio" component in the socio-hydrologic framework we proposed primarily addresses complex decision-making related to water use and conflict driven by GIs. We have included more explanation about the issues in the introduction (See Lines 19 - 30), results, discussion sections (See Lines 575 - 578; 687 - 689) to provide a more comprehensive view of the "socio" component.

**References**

Baumol, W.J., 1988. The theory of environmental policy. Cambridge University Press.
Fisher, A.C., 1981. Resource and environmental economics. Cambridge University Press.
Kallis, G., 2010. Coevolution in water resource development: The vicious cycle of water supply and demand in Athens, Greece. Ecological economics, 69(4), pp.796-809.
Kristal, T. and Cohen, Y., 2017. The causes of rising wage inequality: the race between institutions and technology. Socio-Economic Review, 15(1), pp.187-212.

• Conceptually, while I agree that the planning paradigm described at the city-level in Figure 1 is ideally how city-scale GIs should be constructed, in practice, I do not think this is happening. Instead, due to the long timeframe associated with stormwater funding and construction, GIs tend to be developed sporadically on a project-by-project basis, not in real-time, tightly coupled with the water use and demand, as depicted here. Although I understand that in order to model this as an ABM, you had to make such a coupling decision, perhaps consider a time delay in the model, or mention this limitation in reflecting real-world decisionmaking patterns. This also applies to the hydrologic connections in Fig 1B, between each urban area linked by real-time riverine flows between them. The city-scale decision-making of GI does not align temporally with the hydrologic flows of connecting river systems, even though the time steps appear to be monthly.

**Response:**

Thank you for your thoughtful comment. We agree that GIs tend to be developed sporadically on a project-by-project basis rather than in real-time, tightly coupled with water use and demand. To address this, we have considered different decision variables with different time periods in our agent-based model for urban water management at the city scale.

Specifically, for GI construction decisions, we have set the time step to annually, while for water supply portfolio selections, it is set to monthly (See Equation A10, page 36). The total construction area of the three types of GIs within an urban area over a year approximates the total construction area built by project-by-project efforts within the same period. In our model, the decision variables associated with GI construction represent the total construction area of the three types of GIs within an urban area from the previous year.

The model mechanism operates as follows: the decision variables associated with GI construction are first generated and input into the urban water balance model as parameters, which can change urban land features. Then, the decision variables associated with water supply are generated based on the specific urban water balance model. This approach addresses the issue of different time periods for GI construction and water supply.

Similarly, for simulating hydrologic connections between urban areas, we consider the average monthly flow dynamics between urban areas. This fits the time step for water supply portfolio selections by urban water managers.

In summary, we acknowledge that urban water supply, discharge, and river flow are realtime processes, whereas GI construction periods can vary from several months to several years. While our model assumes an annual time step for GI construction and a monthly time step for water supply and river flow, we believe this approach reasonably approximates real decision-making scenarios for urban and watershed water managers. This tradeoff balances model complexity and computational feasibility.

We have added more explanation to the corresponding part of the methodology section to help readers understand the model approximation assumptions (See Lines 282 - 297). Additionally, we have discussed the limitations of this model assumption in the conclusion section (See Lines 725 - 729).

 Another conceptual question I have - the overall model framework depends on GIs being used at-large for water storage and demand. Perhaps this is common in some parts of the world, but in the US, where the case study is conducted, most GIs are used for runoff abatement, which is then linked back into the greywater infrastructure system and sent offsite to reduce flooding issues. Some rainwater harvesting systems are used for on-site capture and use for irrigation, but these are very small-scale in nature (like someone's personal lawn) and not designed systematically to be a major contributor to widespread irrigation needs. At least I am not aware of this being common practice in the US.

**Response:**

Thank you for your insightful comment. We agree that the widespread use of green infrastructures (GIs) for rainwater utilization in the US is not yet common practice. However, there are several reasons we chose a watershed in the US for our case study.

First, while GIs are more commonly used for runoff abatement and linked to greywater infrastructure in the US, we believe there is a strong foundation and potential for their broader application. Some regions in the US face serious water scarcity issues (Schmidt et al., 2023), and various government levels are already encouraging the development of GIs for purposes such as sustainable stormwater management (Roy et al., 2008), reducing urban flood risk (Bhandari et al., 2018), and improving runoff quality (Guo et al., 2014). For example, rainwater harvesting is both legal and encouraged in the state of Texas. Texas gevernemnt requires new state facilities to add rainwater harvesting systems in their designs, according to the Texas Water Development Board (see link 1). These initiatives provide a solid basis for the potential expansion of GIs to include rainwater harvesting and integration with grey infrastructure for comprehensive urban water cycle management. Current, rainwater collection is actively encouraged in New Zealand (See link 2) and throughout Australia (See link 3), the UK (See link 4) and Canada (See link 5).

Second, there are existing water conflict issues within US watersheds (Philpot et al., 2016), and few studies have examined the potential impact of GIs on these conflicts. Exploring the potential of GIs for rainwater utilization in US watersheds is therefore a valuable area of study.

Additionally, using the proposed model framework requires access to various types of data, such as hydrological, meteorological, urban land data, and GI-related data (e.g., construction and maintenance costs). US watersheds often have extensive and easily accessible data, which supports the feasibility and accuracy of our model. This was a practical consideration in selecting a US watershed for our case study.

While we acknowledge that the current widespread use of GIs for rainwater harvesting in the US is limited, we believe that investigating this potential is worthwhile. We have added more explanation to the corresponding part of the case study section (See Lines 365 - 369). to clarify our rationale and discuss the limitations of our study in the conclusion section (See Lines 732 - 736).

**References**

Khan, Z., Alim, M.A., Rahman, M.M. and Rahman, A., 2021. A continental scale evaluation of rainwater harvesting in Australia. Resources, Conservation and Recycling, 167, p.105378. Ennenbach, M.W., Concha Larrauri, P. and Lall, U., 2018. County-scale rainwater harvesting feasibility in the United States: Climate, collection area, density, and reuse considerations. JAWRA Journal of the American Water Resources Association, 54(1), pp.255-274. Schmidt, J.C., Yackulic, C.B. and Kuhn, E., 2023. The Colorado River water crisis: Its origin and the future. Wiley Interdisciplinary Reviews: Water, 10(6), p.e1672. Roy, A.H., Wenger, S.J., Fletcher, T.D., Walsh, C.J., Ladson, A.R., Shuster, W.D., Thurston, H.W. and Brown, R.R., 2008. Impediments and solutions to sustainable, watershed-scale urban stormwater management: lessons from Australia and the United States. Environmental management, 42, pp.344-359. Bhandari, S., Jobe, A., Thakur, B., Kalra, A. and Ahmad, S., 2018, May. Flood damage reduction in urban areas with use of low impact development designs. In World Environmental and Water Resources Congress 2018 (pp. 52-61). Reston, VA: American Society of Civil Engineers. Guo, J.C., Urbonas, B. and MacKenzie, K., 2014. Water quality capture volume for storm water BMP and LID designs. Journal of Hydrologic engineering, 19(4), pp.682-686. Philpot, S., Hipel, K. and Johnson, P., 2016. Strategic analysis of a water rights conflict in the south western United States. Journal of Environmental Management, 180, pp.247-256.

**Links**

- https://brazos.org/about-us/education/water- school/articleid/358#:~:text=Rainwater%20harvesting%20is%20both%20legal,h arvesting%20systems%20for%20personal%20use.
- 2. https://www.building.govt.nz/getting-started/smarter-homes-guides/water-and-waste/collecting-and-using-rainwater
- 3. https://rainwaterharvesting.org.au/
- 4. https://www.rainwaterharvesting.co.uk/
- 5. https://www.harvestingrainwater.ca/
  - It seems to me that the concept being simulated is actually systematic decision-making for detention-pond and reservoir storage at both the city watershed scales, and how inter-city feedbacks can impact the overall cycle. Which is not necessarily "Green Infrastructure" as I understand it to be used in the literature and community.

**Response:**

Thank you for your insightful comment. We understand your concern regarding the distinction between green infrastructure (GI) and grey infrastructure, such as detention ponds and reservoir storage, and how they impact urban water cycles and inter-city feedbacks.

We agree that large-scale centralized grey infrastructure, such as reservoir storage, indeed plays a significant role in urban water cycles and interactions between urban areas. However, the dominance of grey infrastructure in socio-hydrologic dynamics also presents challenges in urban and watershed water resource management. These challenges include unsustainable development (Munoz, 2016) and the supply-demand cycle (Di Baldassarre et al., 2018). In watersheds with severe water scarcity issues, building large-scale centralized grey infrastructure can lead to over-extraction of water to support a population beyond the region's carrying capacity, potentially harming the watershed environment (Di Baldassarre et al., 2018). Additionally, in economically disadvantaged regions, the high cost of constructing and maintaining centralized grey infrastructure may be prohibitive.

These challenges have led to increased interest and advocacy from governments, the academic community, and practitioners for the development of decentralized green infrastructures (Sitzenfrei et al., 2020; Daigger and Crawford, 2007). GIs are generally considered more sustainable and affordable. Numerous studies have demonstrated the effectiveness of GIs in urban water systems, and several countries have begun to implement GI practices extensively. Examples include Low Impact Development (LID) in the US (Zahmatkesh et al., 2014), Water Sensitive Cities in Australia (Howe and Mitchell, 2011), Sustainable Urban Drainage Systems (SuDS) in the UK (Andoh and Iwugo, 2002), and Sponge Cities in China (Guan et al., 2021).

The increasing scale of GI development and its role in urban water systems have motivated us to explore the potential of GIs for direct and indirect rainwater reuse in urban water systems and their implications for water conflicts within watersheds. Additionally, we are interested in understanding how to make IGWM decisions, including GI construction and water supply portfolio selections, when GIs are widely used for rainwater reuse.

In our paper, we adopt a broad definition of GIs, encompassing three types: rainwater harvesting systems, stormwater harvesting systems, and infiltration-based GIs. This includes GI practices that convert impervious areas into pervious areas to increase groundwater recharge and indirectly use rainwater, such as urban green spaces and constructed wetlands. For example, a detention pond can be considered a GI in our model. If the water in the detention pond is used for infiltration, it is regarded as an infiltration-based GI. If the rainwater collected by the detention pond is reused, it is considered a stormwater harvesting system. By using this generalized definition of GIs, our model can include a wide range of GI practices within urban areas.

Although the impact of a single GI practice on the urban water cycle may be small, the cumulative effect of all GI practices within urban areas on the urban water cycle and hydrological interactions between urban areas cannot be ignored, especially with the continuous development of GIs.

We have added more explanation to the corresponding sections of our paper to clarify our rationale and discuss the implications of using GIs in urban and watershed water resource management (See Lines 24 - 26; Lines 254 - 256).

**References**

Di Baldassarre, G., Wanders, N., AghaKouchak, A., Kuil, L., Rangecroft, S., Veldkamp, T.I., Garcia, M., van Oel, P.R., Breinl, K. and Van Loon, A.F., 2018. Water shortages worsened by reservoir effects. Nature Sustainability, 1(11), pp.617-622. Sitzenfrei, R., Kleidorfer, M., Bach, P.M. and Bacchin, T.K., 2020. Green infrastructures for urban water system: Balance between cities and nature. Water, 12(5), p.1456. Daigger, G.T. and Crawford, G.V., 2007. Enhancing water system security and sustainability by incorporating centralized and decentralized water reclamation and reuse into urban water management systems. Journal of Environmental Engineering and Management, 17(1), p.1. Munoz, N.J., 2016. What Is The Economic Feasibility Of Implementing Grey Water Infrastructure At The Citywide Level?. Master's Projects and Capstones. 353. https://repository.usfca.edu/capstone/353 Howe, C. and Mitchell, C. eds., 2011. Water sensitive cities. IWa Publishing. Zahmatkesh, Z., Karamouz, M., Burian, S.J., Tavakol-Davani, H. and Goharian, E., 2014. LID implementation to mitigate climate change impacts on urban runoff. In World Environmental and Water Resources Congress 2014 (pp. 952-965). Andoh, R.Y. and Iwugo, K.O., 2002. Sustainable urban drainage systems: a UK perspective. In Global Solutions for Urban Drainage (pp. 1-16). Guan, X., Wang, J. and Xiao, F., 2021. Sponge city strategy and application of pavement materials in sponge city. Journal of Cleaner Production, 303, p.127022.

• Perhaps this is me not understanding the Markov property, but it seems counterintuitive to apply this property, essentially stating the decision-making process is stochastic and its future evolution is completely independent of its history, as the opposite paradigm is key to a system operating "socio-hydrologically".

**Response:**

Thank you for your insightful comment. We appreciate the opportunity to clarify our use of the Markov property in the context of our model.

The basic concept of the Markov property refers to the memoryless characteristic of a stochastic process, meaning that the future state of the process depends only on the present state and not on the sequence of events that preceded it (Frydenberg, 1990).

In our framework, we approximate the interaction between urban areas within an inter-cityscale IGWM framework as having Markov property. Specifically, consider a scenario where multiple urban areas are situated along a river, sharing water resources from the same river. The decision-making processes for water withdrawal and discharge in upstream urban areas affect those in downstream urban areas due to the flow of water along the river. To simulate these interactions, we begin with the decision-making process of the first urban area upstream. This initial decision influences the subsequent decision-making processes of the adjacent downstream urban areas, following the river's flow. Our multiagent framework arranges the decision-making processes in a sequence based on the urban areas' locations along the river. The hydrologic state of the first upstream urban area after making own IGWM decisions is the subsequent state. The hydrologic state in an urban area depends on the only the hydrologic state of the adjacent upstream area after making own IGWM decisions and the decisions makings of the urban area itself. The state transition process continues sequentially along the river.

Thus, the interaction process among urban areas in our model resembles the Markov property: the future state (i.e., the hydrologic state of a downstream urban area) depends only on the present state (i.e., the hydrologic state of the adjacent upstream area) and not on earlier states (i.e., the hydrologic state of other upstream areas).

This is why we consider the interaction between urban water manager agents in watershedscale IGWM as exhibiting Markov property. We have added more explanation to the corresponding parts of the introduction (See Lines 74 - 79) and methodology sections (See Lines 306 - 313) to help readers understand this basic assumption for the multiagent system.

**References**

Frydenberg, M., 1990. The chain graph Markov property. Scandinavian journal of statistics, pp.333-353.

**Case Study:**

This is an extremely large watershed area for designing with GI. Was this selected at random, or is there a decision-making entity that manages this trans-state watershed systematically? I looked up the UMRBA association, and they don't seem to actively manage water supply and use in this basin.

**Response:**

Thank you for your insightful comment. We appreciate the opportunity to provide clarification regarding the selection of our study area and the role of decision-making entities.

Our bi-level multiagent framework is designed to capture the interactions among multiple stakeholders, including urban water managers and a watershed manager, at different authority levels within the watershed-scale IGWM.

For urban water managers, their responsibility is to make decisions regarding city-scale IGWM within their respective urban areas. This includes decisions on GI construction and water supply portfolios.

For the watershed manager, we refer to an entity such as the Upper Mississippi River Basin Association (UMRBA). While the UMRBA does not actively manage water supply and use within urban areas, it plays a crucial role in managing water resources across the Upper Mississippi River Basin (see weblink 1). Our model framework involves the watershed manager enacting a streamflow penalty strategy to ensure equitable surface water allocation. This strategy involves setting low flow thresholds at various checkpoints along the river. Urban areas that withdraw surface water beyond these thresholds incur penalties. This approach is inspired by the water level management priorities established by the UMRBA, which aim to balance multiple objectives such as disaster preparedness, economic growth, and ecological health (see weblink 2).

We acknowledge that the Upper Mississippi River Basin Association's (UMRBA) actual water level management policy integrates multiple purposes, particularly balancing ecosystem benefits and navigation operations. It is important to note that participants at the lower authority levels in this policy are not urban water managers, which differs from the model we proposed. Our model focuses solely on the goal of equitable water resource allocation to explore the conflict between cost and equity of water use driven by GIs in the Upper Mississippi River Basin (Guo, 2023). However, our proposed model's multiple-player bi-level structure is somewhat similar to the actual water level management structure. This similarity suggests that, despite differing management goals, the interactions between participants might be comparable, potentially offering valuable insights to inform UMRBA's actual water level management policy. To the above issue, we have added a discussion on relevant parts of the case study (See Lines 374 - 377).

**References**

Guo, Q., 2023. Strategies for a resilient, sustainable, and equitable Mississippi River basin. River, 2(3), pp.336-349.

**Links**

- 6. https://www.encyclopediadubuque.org/index.php/UPPER\_MISSISSIPPI\_RIVER\_BASIN\_ASSO CIATION
- 7. https://umrba.org/document/umrba-2022-water-level-management-priority-actions
  - I looked at the 1 citation mentioned for using GI in this geographical region 0 (Askey-Merwin, 2020), and this publication addresses mitigating flooding, not storing rainwater via GIs and re-use in widespread irrigation projects. Moreover, the Mississippi River is one of the highest-flow rivers in the US and has a complex network of laws and regulations regarding water extraction for municipal or industrial use. I am not aware of water quantity being an issue here for irrigation purposes, so I am confused why there is a study suggesting GI is being actively proposed, managed, and constructed at the watershed scale and the city scales in real-time to ensure water availability here. If it is purely for a conceptual purpose of explaining complexities of GI planning in general, that is fine, but in that instance, I wouldn't necessarily limit the model to connecting urban communities along a main river stem, as this limits the application substantially. However, the model is already built under these assumptions, so I do not recommend re-designing. I am just pointing out

it conceptually is difficult for me to see the application outside of this theoretical explorative study.

**Response:**

Thank you for your detailed comment and for highlighting important considerations regarding the application of our model framework to the Upper Mississippi River Basin.

We acknowledge that the Upper Mississippi River Basin is one of the water-rich regions in the US. However, this basin hosts a diverse range of stakeholders and water users. Over 70% of the area is dedicated to agriculture and animal husbandry, while only 5% is urbanized, yet it supports a population of approximately 24 million, particularly in high-density metropolitan areas. Additionally, the basin includes the national Wildlife and Fish Refuge and 12,000 miles of commercially navigable channels, which require maintenance of environmental flows for ecological health and navigation purposes (see weblink 1).

Consequently, water competition among different users with varying objectives becomes significant, especially in the context of climate change. Climate change can drastically alter the probability distribution of streamflow, increasing the frequency and magnitude of both high and low streamflow extremes (Asadieh and Krakauer, 2017). Additionally, some high-flow river basins, such as the Yellow River basin in China, face the risk of a gradual decrease in stream flows due to climate change (Wang et al., 2017). Under these conditions, the water quota for urban use might be limited, making water conflicts among urban areas a pertinent issue, even in this high-flow river basin that might face nonstationary drought risk (Dierauer and Zhu, 2020).

Currently, the primary purpose of developing GIs in the basin is to manage stormwater, mitigate flooding (Askey-Merwin, 2020), filter pollutants, and enhance the quality of life. Our framework also incorporates the development of infiltration-based GIs to manage stormwater and increase groundwater recharge, which complements the integration of GIs with grey infrastructure in urban water systems.

We acknowledge that there is no existing government policy specifically advocating for rainwater harvesting and direct reuse via GIs in the basin. However, recent studies have begun to call for developing GIs to lessen stormwater runoff, alleviating stress on traditional grey systems. (Guo, 2023). The study from Ennenbach et al., (2018) have demonstrated the potential of rainwater harvesting in the basin, attributed to its humid climate and abundant annual precipitation. Nevertheless, the research also indicates that seasonal variations in water demand necessitate a more accurate assessment of rainwater harvesting potential. This assessment should consider factors such as local rooftop area and population density. Therefore, we believe it is both feasible and valuable to apply our proposed model framework to the Upper Mississippi River Basin to explore the potential and outcomes of widespread GI development for direct and indirect rainwater use at different scales.

We also agree that applying our model framework to a basin facing severe water scarcity, such as the Colorado River Lower Basin (Schmidt et al., 2023), or to regions with established policies for GI development for rainwater use, such as the Albemarle-Pamlico river basins

(Ghimire and Johnston, 2013), may be more suitable and meaningful. We have added a discussion on case study selection in the relevant sections of the case study (See Lines 365 - 374) and limitations and future research sections of the conclusion (See Lines 732 - 736).

**References**

Ennenbach, M.W., Concha Larrauri, P. and Lall, U., 2018. County-scale rainwater harvesting feasibility in the United States: Climate, collection area, density, and reuse considerations. JAWRA Journal of the American Water Resources Association, 54(1), pp.255-274. Askew-Merwin, C., 2020. Natural Infrastructure's Role in Mitigating Flooding Along the Mississippi River, Northeast-Midwest Institute Report, 16 pp. Guo, Q., 2023. Strategies for a resilient, sustainable, and equitable Mississippi River basin. River, 2(3), pp.336-349. Ghimire, S.R. and Johnston, J.M., 2013. Impacts of domestic and agricultural rainwater harvesting systems on watershed hydrology: A case study in the Albemarle-Pamlico river basins (USA). Ecohydrology & Hydrobiology, 13(2), pp.159-171. Schmidt, J.C., Yackulic, C.B. and Kuhn, E., 2023. The Colorado River water crisis: Its origin and the future. Wiley Interdisciplinary Reviews: Water, 10(6), p.e1672. Asadieh, B. and Krakauer, N.Y., 2017. Global change in streamflow extremes under climate change over the 21st century. Hydrology and Earth System Sciences, 21(11), pp.5863-5874. Wang, G., Zhang, J., Jin, J., Weinberg, J., Bao, Z., Liu, C., Liu, Y., Yan, X., Song, X. and Zhai, R., 2017. Impacts of climate change on water resources in the Yellow River basin and identification of global adaptation strategies. Mitigation and adaptation strategies for global change, 22, pp.67-83. Dierauer, J.R. and Zhu, C., 2020. Drought in the twenty-first century in a water-rich region: modeling study of the Wabash River Watershed, USA. Water, 12(1), p.181.

**Links**

- 1. https://www.fws.gov/refuge/upper-mississippi-river
  - Where are you getting water demand data for irrigation, and how does this change over time? What factors drive this in the model? I see that you simulated "urban" water demand via population and urban layouts, but as mentioned, this is a tiny percentage of the overall water resources in the basin, and is likely to be impacted significantly by urban-scale GI units. I see some mentioning of irrigation demand in Eq A10, but it is unclear to me what these equations mean or how the underlying data were gathered. Is the irrigation a basis of cropland type?

**Response:**

Thank you for your insightful comment and for giving us the opportunity to clarify our approach to water demand data and its drivers in the model.

In our model, we focus exclusively on water demand within urban areas, and agricultural water demand is not included. We consider three types of urban water demands: indoor potable, indoor non-potable, and outdoor non-potable water demand. These categories are

inspired by the work of Last (2011) and are measured based on the associated urban populations and layouts using the method described in Last (2011).

Regarding irrigation demand within urban areas, it primarily refers to the water needed for maintaining vegetation in both large-scale and small-scale infiltration-based GIs. Large-scale GIs include street trees, urban green spaces, parks, and gardens (Fam et al., 2008; Caetano et al., 2014) etc., while small-scale GIs encompass green roofs, rain gardens, and bioretention systems (Mechelen et al., 2015) etc. These plants require irrigation to sustain their basic ecological functions.

As outlined in the 14 row of Eq. A10 (Page 36), the estimation of urban irrigation demand involves the ratio of soil moisture for plant demand to saturated soil moisture  $(f_{sm})$ . This ratio is used to determine the minimum storage level of the shallow soil layer required to meet the basic water demands of plants. The concept is inspired by the study conducted by Mitchell et al. (2001), which establishes a trigger-to-irrigate threshold based on soil moisture levels. According to this approach, irrigation is initiated when the soil moisture level in the shallow soil layer falls below a user-defined 'trigger-to-irrigate' threshold to compensate for the deficit. In our case study, the relevant parameter  $(f_{sm})$  is set as 0.31, in accordance with the recommendations provided in Mitchell et al. (2001).

In our agent-based model for city-scale IGWM, the estimation of the water required in the soil layer at time t is inspired by the Australian Water Resources Assessment-Landscape (AWRA-L) Model (Frost et al., 2016). As shown in Fig. 1, the ratio of the saturated area to the pervious surface ( $f_{sat}(t)$ ) at time t can be estimated based on the height of the groundwater table. In the saturated area, the soil is fully saturated, indicating no irrigation demand.

Consequently, the corresponding ratio of the unsaturated area with irrigation demand to the pervious surface is equal to  $(1 - [f_{sat}(t)]_*)$ . The total water deficit in the soil layer at time t can be estimated by multiplying this ratio by the maximum storage capacity of the shallow soil layer,  $S_{smax}$ , which is set according to the recommendations provided in the referenced work (Frost et al., 2016). Therefore, the minimum storage level of the shallow soil layer required to meet the basic water demands of plants can be calculated as  $f_{sm} \cdot (1 - [f_{sat}(t)]_*) \cdot S_{smax}$ . Thus, the constraint of urban irrigation demand in month t can be formulated by ensuring that the storage level of the shallow soil layer in month t does not fall below this specific minimum storage level required for plant water requirements, namely,

$$f_{sm} \cdot (1 - [f_{sat}(t)]_*) \cdot S_{smax} \le [S_s(t)]_*$$

Fig. 1. The vertical structure of the UWB-SM.

We haved added a detailed explanation of the agent-based model for UWM to the associated Appendix section to ensure readers can understand its technical details (See Lines 213 - 214; Lines 774 - 778). Additionally, we have provided more detailed explanations and references to demonstrate how the underlying data for our model were gathered in the case study (See Lines 475 - 476).

**References**

Last, E.W., 2011. City water balance: a new scoping tool for integrated urban water management options (Doctoral dissertation, University of Birmingham).

Caetano, F., Pitarma, R. and Reis, P., 2014, June. Intelligent management of urban garden irrigation. In 2014 9th Iberian Conference on Information Systems and Technologies (CISTI) (pp. 1-6). IEEE.

Fam, D., Mosley, E., Lopes, A., Mathieson, L., Morison, J. and Connellan, G., 2008. Irrigation of urban green spaces: A review of the environmental, social and economic benefits. CRC for Irrigation Futures Technical Report, 4(08).

Van Mechelen, C., Dutoit, T. and Hermy, M., 2015. Adapting green roof irrigation practices for a sustainable future: A review. Sustainable Cities and Society, 19, pp.74-90.

Frost, A.J., Ramchurn, A. and Smith, A., 2016. The bureau's operational AWRA landscape (AWRA-L) Model. Bureau of Meteorology technical report.

Mitchell, V.G., Mein, R.G. and McMahon, T.A., 2001. Modelling the urban water cycle. Environmental modelling & software, 16(7), pp.615-629.

• I am not qualified to review the set-up of the ABM model, particle optimization schemes, or economic theory choices. Please ensure one of the other reviewers has this expertise and can comment on the methodology.

**Response:**

Thank you for your suggestion, and we greatly appreciate your academic integrity and professionalism in reviewing our manuscript. The majority of the technical details of our model, the associated solution approach, and the motivation for our economic theory choices are provided in the relevant sections of the introduction, methodology, and appendix. We also hope that other reviewers with expertise in these specific areas can comment on the model and the associated solution approach we propose. Their feedback can help strengthen our study and improve the quality of our paper.

• I do not see where the channel geometry is used for the Muskingum-Cunge routing method.

**Response:**

Thank you for your question regarding the use of channel geometry in the Muskingum-Cunge routing method.

In fact, Muskingum-Cunge routing method does not require the use of channel geometry information. This method is a data-driven river channel model that uses a storage relation to link inflow and outflow in a channel reach (Garbrecht and Brunner, 1991). The outflow of the river reach at time (t), (Q(t)), can be expressed as:

 $Q(t) = C_1 \cdot I(t) + C_2 \cdot I(t-1) + C_3 \cdot O(t-1)$ ,

where (Q(t-1)) is the outflow of the river reach at time (t-1), and (I(t)) and (I(t-1)) are the inflow rates at times (t) and (t-1), respectively. The Muskingum-Cunge routing method is flexible enough to simulate flow changes in river channels at any time step. The model parameters  $(C_1)$ ,  $(C_2)$ , and  $(C_3)$  can be calibrated and verified using inflow and outflow time series data for specific time steps. Given the model parameters  $(C_1)$ ,  $(C_2)$ , and  $(C_3)$ , and the initial inflow time series data, we can simulate the outflow time series in the river reach using the model.

We hope this clarifies our approach and the rationale behind using the Muskingum-Cunge routing method in our study (See Lines 898 - 913).

**References**

Garbrecht, J. and Brunner, G., 1991. Hydrologic channel-flow routing for compound sections. Journal of Hydraulic Engineering, 117(5), pp.629-642.

• I am not following how USGS stations for the Mississippi river could be used to calibrate urban water use.

**Response:**

Thank you for your insightful comment regarding the use of USGS stations for calibrating urban water use.

In our model framework, we need to calibrate two types of models: the Urban Water Balance Simulation Model (UWB-SM) and the Muskingum-Cunge routing model. The calibration process is somewhat intricate due to the spatial differences between the inlet and outlet of urban areas and the locations of the adjacent USGS stations.

Firstly, we identify the inlet and outlet locations of an urban area based on its GIS map. We then estimate the monthly inflow and outflow time series data for the urban area using a map correlation method (Archfield and Vogel, 2010) with available monthly streamflow observations from the associated USGS stations in the study system.

These estimated monthly inflow and outflow time series data for each urban area are used to calibrate both models. The calibration process for the Muskingum-Cunge routing model is straightforward and involves using the outflow data from the upstream urban area and the inflow data of the adjacent downstream urban area.

The calibration process for the UWB-SM is more complex. For this calibration, the estimated monthly inflow time series data and the associated monthly rainfall time series data collected from NOAA databases are used as inputs for the UWB-SM. In this model, we assume that rainwater and stormwater supply are not considered, and urban water demands—both indoor and outdoor—are met solely through surface water and groundwater supply. The ratio of surface water to groundwater supply is set as 1:1. The monthly amounts of surface water and groundwater withdrawals are determined based on this setting.

Subsequently, the simulated monthly outflow time series data for the urban area, computed using the UWB-SM, are compared with the estimated monthly outflow data to calibrate and validate the UWB-SM.

We have added more explanation about the model calibration and validation in the relevant part of the case study section (See Lines 480 - 493).

**References**

Archfield, S.A. and Vogel, R.M., 2010. Map correlation method: Selection of a reference streamgage to estimate daily streamflow at ungaged catchments. Water resources research, 46(10).

• The discussion of results is very convoluted. Perhaps consider a concluding paragraph with bullet-points of the main take-aways that can be widely applied. For example, key insights about water costs and conflicts among adjacent communities, the importance of communication of watershed-scale managers and city-scale planners, the impact of assuming agent rationality / Markov property / Stackelberg in understanding the overall socio-hydro dynamics, what this means on overall water policy as GIs become more popular, interaction between urban and irrigation water use and demand, what this study informed us about social equity in water decisions, etc.

**Response:**

Thank you for your valuable feedback. We appreciate your suggestion to improve the clarity and readability of the discussion section.

In response to your comment, we have added rewrite and re-arrange the discussion part in the Results and Discussion section. We will remove any redundant and unimportant parts to make the discussion more focused and easier to understand. Additionally, we have included a concluding paragraph with bullet points highlighting the main takeaways that can be widely applied (See Lines 685 - 695). We believe these changes will enhance the discussion and provide clearer and more actionable insights for the readers.

• The Conclusions section is too convoluted for me. I recommend removing a lot of the technical jargon and focusing on the bigger picture here.

**Response:**

Thank you for your suggestion regarding the Conclusions section. We appreciate your feedback and understand the need for clarity and focus on the bigger picture.

In response to your comment, we have re- written the Conclusions section to remove redundancy and unnecessary technical details (See Lines 696 - 710). We ensure that the revised section highlights the broader implications and key insights of our study (See Lines 711 - 724), making it more accessible and easier to understand.

**Response to Reviewer #2**

Dear reviewer, you have accurately pointed out the key problems of this paper. Thank you so much for your objective and constructive comments to this study, which are of great significance to improve the quality of the paper.

This manuscript discusses an interesting question in urban water management where green infrastructure is integrated into multiple cities' water supply operations. The authors approach this problem by developing an agent-based modeling (ABM) framework to discuss cross-scale interactions among city and watershed water managers (i.e., agents) and explore water equity and policy implication through imposing a penalty for overdraft.

I appreciate the authors ambition to take on the challenge of solving the complex urban water problem and efforts in developing an integrated modeling tool. The introduction effectively highlights the importance of integrated water management at watershed scale and the need for integrated modeling approaches. I believe the scope of this study will be of great interest to the HESS community. That being said, the current manuscript suffers several major flaws that make it difficult to follow and obscure its contributions and intellectual merits. I am fully committed to helping elevate the quality of this manuscript. If any comments arise from my lack of knowledge on specific points, please accept my apologies in advance. Below are the summaries of my comments and suggestions followed by the specific comments.

My first comment is about the writing. The current manuscript is difficult to follow due to the excessive technical terms and ambiguous language. For example, IGWM (short for integrated green infrastructure and water resources management) was applied in describing models, agents, and agents' decisions, which gave me a headache. Other examples include WM (water manager), UWM (urban water manager), and HUWS (hybrid urban water system). Some technical terms are not well-defined. For example, ABM (agent-based model) and MAS (multi-agent system) are often used interchangeably in the literature, but it was presented in this manuscript as two distinct modeling approaches applied for building two models (i.e., city-scale and inter-city). Similarly, rainwater and stormwater are the same thing to me, and yet they are listed as two water sources (lines 183-184). In the results section, the authors discuss the usage of the four water sources (surface water, groundwater, stormwater, and rainwater), so the model must have simulated the water sources. However, I could not find their definitions nor how the water supply portfolio is simulated except for surface water. I will recommend a more rigorous quality control and assurance to improve the flow and readability.

**Response:**

Thank you for your thorough review and valuable feedback on our manuscript. We understand your concerns regarding the clarity and readability of the text, particularly related to the use of technical terms and acronyms. We apologize for any confusion caused by the excessive use of acronyms and the lack of clear definitions for some key concepts.

To address the issues you raised, we propose the following revisions:

- Reduction of Acronyms: We have reduced the number of acronyms by eliminating those that are infrequently used, such as HUWS, MAS, MAS-UWM, BL-MAS, ABM, ABM-UWM, ABM-WM, S-APSO and IGWM-OM. Additionally, we have included a comprehensive list of acronyms to assist readers in understanding the terms used throughout the text (See Lines 741-746).
- 2. Clarification of Key Concepts: We have added clear definitions for key concepts that were omitted in the current version of the manuscript, such as, agent-based model (See Lines 122 123) and multi-agent system (See Lines 123 -125). Specifically, we have provided explicit definitions for rainwater and stormwater (See Lines 203 205). As referenced in studies by Khan et al. (2023) and Fielding et al. (2015), stormwater is the water that drains off land areas from rainfall, including water from rooftops, ground surfaces, and other areas. In contrast, rainwater refers specifically to the rain that falls on roofs and can be collected into storage tanks before contacting the ground, resulting in higher quality due to fewer contaminants. The distinction between these two sources is crucial as it impacts their respective urban water cycle and costs of water use (MSSC, 2008), which is why both are considered in our model framework.
- 3. **Detailed Explanation of Simulation Processes**: We have enhanced the methodology sections with additional details on the simulation processes, particularly how the water supply portfolio is simulated within our proposed model framework (See Lines 282 297).

**References**

- Khan, A., Park, Y., Park, J., Sim, I. and Kim, R., 2023. Analysis of Stormwater and Rainwater Harvesting Potential Based on a Daily Water Balance Model: A Case Study of Korea. Water, 16(1), p.96.
- Fielding, K.S., Gardner, J., Leviston, Z. and Price, J., 2015. Comparing public perceptions of alternative water sources for potable use: The case of rainwater, stormwater, desalinated water, and recycled water. Water Resources Management, 29, pp.4501-4518.
- Minnesota Stormwater Steering Committee (MSSC)., 2008. State of Minnesota Stormwater Manual: Version 2. Minnesota Pollution Control Agency, St. Paul, MN. https://stormwater.pca.state.mn.us/index.php/Overview\_for\_stormwater\_and\_rainwater\_harvest\_a nd\_use/reuse

My second major comment is about the framing of the model. After reading the method section a few times, the modeling components become clear to me. The modeling

framework includes three models coupling together. However, it is essentially one agent-based model with two agent types (city agents, UWM, and a watershed agent, WM) and a hydrologic model (including UWB-SM and M-C) representing the spatial connections among the city agents and the watershed environment. I can understand the authors' intention in examining the interactions among agents across multiple sales; however, framing the models separately at different scales has had the opposite effect for me, leaving me confused and obscuring my understanding of the study. The suggestion here may be somehow subjective, but I am hoping that the manuscript could benefit from my perspective.

**Response:**

Thank you for your insightful comments on the framing of our model. We appreciate your efforts in thoroughly reviewing the methodology section and providing valuable feedback.

The primary reason for framing our models separately at different scales is to explore the role of green infrastructures (GIs) in water resources management from the city scale to the watershed scale. Our intention was to simulate the socio-hydrologic interactions driven by the introduction of GIs within multiple agent systems. We believe that the impacts of introducing GIs for rainwater use in urban water resources systems are not only local (city-scale) but also overall (watershed-scale) due to the social and hydrologic connections between urban areas. Therefore, as described in the methodology section, we constructed our model from the city scale to the inter-city scale to the watershed scale.

However, based on your feedback, we understand that the current presentation of our model framework may be confusing to some readers. To address this, we have proposed the following revisions:

- Introduction of the Model Framework: We have added a new section at the beginning of the methodology section to briefly introduce the entire framework of our models, the corresponding components, and their relationships (See Lines 187 - 272). This can provide readers with a clear overview of the model structure and its components, facilitating a better understanding of the overall framework.
- 2. Improvement of Model Details: We have enhanced the subsequent parts of the methodology section to ensure coherence, cohesiveness, and consistency. Specifically: a. We have highlighted and explain the relationships between models at different scales, making it easier for readers to understand the underlying logic of framing the models separately at different scales (See Lines 273 349). b. We have deleted redundant and overlapping content between the parts detailing models at different scales, making the manuscript more readable and straightforward.

Generally, I found it difficult to follow the results and discussion, partly attributed to not fully understanding the modeling components. Since I did not go through all the details

in the Appendices, I was not sure whether I could agree or disagree with the findings and discussion. I will suggest presenting the key components of the models in the main text. For example, in lines 297 – 298, the authors mentioned an assessment index (Gini coefficient) set by the WM agent but did not go any further to explain how it was incorporated into WM's decision-making nor describe what the Gini coefficient means and how it is calculated. I feel the results and discussion can be condensed to focus on key findings as a long discussion would lose its audience. Another suggestion is to provide more details of the urban water balance model (UWB-SM) in the main text as the model of the physical environment since it is where the water partition is determined. Contrarily, the texts related to the routing model (M-C method) and solution approach (S-APSO) can be moved to the Appendix. Is the S-APSO approach the original creation of this work? If so, I think the solution approach as well as the UWM model could be a separate paper.

**Response:**

Thank you for your constructive comments and suggestions on our manuscript. We apologize for any confusion caused by the presentation of our results and discussion sections. We appreciate your detailed feedback and propose the following revisions to address your concerns:

**Key Model Components in the Main Text**: We acknowledge that the detailed explanations and associated calculation equations for the assessment index (Gini coefficient) and the urban water balance model (UWB-SM) are located in the appendix in the previous version of the manuscript. To improve clarity, we have reorganized and re-written the methodology and corresponding appendix sections. Specifically:

- 1. We have presented the some technical details for key components of the models, including the urban water balance model in the main text to provide a clearer understanding of the modeling framework (See Lines 230 267).
- We will move the minor components of the model, such as the routing model (M-C method) and the solution approach (S-APSO), to the appendix (See Lines 898 913; 920 996). Although the S-APSO approach is an original creation of this work, designed specifically to solve the proposed model framework, we maybe consider your suggestion to further develop this approach and potentially publish it as a separate paper in the future.

**Clarification of the Gini Coefficient**: We have provided a detailed explanation of the Gini coefficient, including its meaning, how it is calculated, and how it is incorporated into the WM agent's decision-making process, within the main text (See Lines 224 - 228). We also added a figure to introduce the basic principle for calculating the Gini coefficient in the main text (See Figure. 2 page 9). This can help readers understand the relevance of this index to our study.

**Figure 2.** (a) Basic principle for calculating water allocation Gini coefficient, and (b) The objective function for agent-based model for WM. **Note:** In Fig. 2 (B), relevant symbols are listed in Tab. B1 and B2 in Appendix B1.

**Results and Discussion**: In response to your comments, we have re-written and rearranged the Results and Discussion section. We have condensed the discussion to focus on key findings and remove any redundant or unimportant parts to make the discussion more focused and easier to understand (See Lines 496 - 693). We believe these changes can enhance the discussion and provide clearer and more actionable insights for the readers.

Overall, this manuscript has the potential to be a high-quality paper (by the modeling framework itself) if the authors can improve the clarity in the methodology, experiment designs, and discussion and highlight its contributions.

**Specific Comments**

 Lines 27-34: The introduction highlights the need for multi-scale green infrastructure frameworks in urban water management. The introduction needs to provide a detailed positioning within recent literature and how the current research contributes to the body of knowledge. Integrating findings from recent studies on similar frameworks could help contextualize their suggested approach within the broader field and clarify its unique contributions. Suggestion: Expand the literature review to include recent ABM applications in socio-hydrology and water resources.

**Response:**

Thank you for your valuable suggestion regarding the introduction section of our manuscript. We agree with your opinion that the introduction should provide a detailed positioning within recent literature and clearly demonstrate how the current research contributes to the body of knowledge. In the current version of the introduction, we have referenced some previous studies on multi-scale green infrastructure frameworks in urban water management, particularly IGWM at the city scale (See Lines 96 -118), inter-city scale (See Lines 119 - 124), and watershed scale (See Lines 125-143). We also analyzed their contributions and identified gaps. However, we acknowledge that the literature review may be insufficient and its current positioning may not effectively highlight the research problem, motivation, and gaps addressed by our study. To address your comments, we have undertaken the following revisions:

**Expand the Literature Review**: We have expanded the literature review to include recent applications of agent-based models (ABM) in socio-hydrology and water resources management (See Lines 133 - 138). This can help contextualize our suggested approach within the broader field and clarify its unique contributions.

• Line 15–20: Add brief mention of the specific experimental scenarios (e.g., "streamflow penalty" and GI adoption) to give readers a clearer picture of the paper's approach and key findings at the outset. This will make the abstract more informative for readers skimming the content.

**Response:**

Thank you for your insightful suggestion regarding the introduction section of our manuscript. To address your comment, we have revised the beginning of the introduction section to include brief descriptions of these specific experimental scenarios (See Lines 19 - 38). This can help set the stage for our research and make the introduction more informative and engaging for readers.

 Line 35–40: The statement on the importance of GIs could be made more impactful by adding specific challenges (e.g., "urban flooding, groundwater depletion, and inter-city water conflicts") that this framework aims to address. This would help sharpen the focus on the practical problems the model intends to resolve.

**Response:**

Thank you for your valuable suggestion regarding the statement on the importance of green infrastructures (GIs) in our manuscript. To address your comment, we have revised the relevant section to include specific challenges for IGWM at city, inter-city and watershed scales (See Lines 50 - 56; 74 - 81; and 91 - 95). This can demonstrate the importance of GIs more effectively and highlight the practical significance of our framework.

• Ensure that acronyms such as "GI" (for Green Infrastructure) and "UWM" (for Urban Water Manager) are consistently defined and used throughout the text. For instance, Line 42 introduces GI without explicitly defining it, which may confuse readers unfamiliar with the abbreviation.

**Response:**

Thank you for your helpful suggestion regarding the use of acronyms in our manuscript. To address your comment, we will undertake the following actions:

- 1. **Review and Modify Acronyms**: We have carefully reviewed the manuscript to ensure that all acronyms, such as "GIs" for Green Infrastructures and "UWM" for Urban Water Manager, are explicitly defined (GIs see Lines 13-15 and UWM and WM see lines 32-34) when first introduced and consistently used throughout the text.
- 2. **Reduce the Number of Acronyms**: We have reduced the number of acronyms by eliminating those that are used infrequently, such as HUWS, MAS, MAS-UWM, BL-MAS, ABM, ABM-UWM, ABM-WM and S-APSO, thereby simplifying the text and reducing potential confusion.
- 3. Add a Comprehensive List of Acronyms: We have included a comprehensive list of acronyms in the manuscript to help readers easily understand the terms used (See Lines 741-746).
- Specific terms, such as "hydrologic regime" and "multiagent system," are used inconsistently. A brief definition of these terms early in the manuscript (in the Introduction or Methods) would improve consistency.

**Response:**

Thank you for your valuable suggestion regarding the use of specific terms in our manuscript. To address your comment, we have taken the following actions:

- 1. **Review and Ensure Consistency**: We have reviewed the manuscript to ensure that all specific terms are used consistently throughout the text. For example, we have avoided to using "hydrologic regime" terms.
- 2. **Provide Definitions**: We have included brief definitions of these terms in the Introduction and Methods sections, such as GIs (See Lines 13-15), UWM (See Lines 33-34), WM (See Lines 34-35), agent-based model (See Lines 122-123), multiagent system (See Lines 123-124), infiltration-based GIs, stormwater and rainwater harvesting systems (See Lines 202-204), to enhance understanding and consistency for the readers

• The methodology section presents a layered framework with urban and watershed scales involving socio-economic and hydrologic variables. However, the description of how these scales is integrated within a complex system would benefit from additional detail and clarity.

**Response:**

Thank you for your insightful suggestion regarding the methodology section of our manuscript. To address your comment, we have taken the following actions:

- Add an Overview Paragraph: We have added a paragraph at the beginning of the methodology section to briefly introduce the entire framework of our models, including the corresponding components and their relationships (See Lines 187 272). This paragraph can focus on the interactions and relationships between models at different scales and how these local-scale models are integrated within a complex system.
- 2. Enhance Detail and Clarity: We have improved the subsequent parts of the methodology section to make the descriptions of models at different scales more coherent, cohesive, and consistent (See Lines 273 349). Specifically, we have highlighted and explained the relationships between models at different scales, making it easier for readers to understand the underlying logic of framing the models separately at different scales.
- Lines 170-182: The agent-based modeling (ABM) setup could be explained more systematically. Clarifying the assumptions behind each agent's decision-making process, especially for UWMs and watershed managers, would make the model's structure more understandable. Additionally, line 175 references the "Markov property," but a brief explanation or contextualization within the model would benefit readers unfamiliar with this concept.

**Response:**

Thank you for your valuable suggestion regarding the agent-based modeling (ABM) setup in our manuscript. To address your comment, we have taken the following actions:

 Systematic Explanation of ABM Setup: We have added a new section to briefly introduce the entire framework of our models and the corresponding components at the beginning of the methodology section (See Lines 187 - 272). This includes a more systematic explanation of the assumptions behind the decision-making processes of different agents, including UWMs (See Lines 200 - 214) and watershed managers (See Lines 215 - 228). Although these assumptions are detailed in the Appendix (See Lines 750-760; Lines 779-787), we have extracted key assumptions and place them in appropriate positions within the methodology section to make the model's structure more understandable (See Lines 201 – 208).

- 2. Clarification of the Markov Property: While a brief explanation of the Markov property is provided in the introduction section, we have further improved this explanation (See Lines 75-79). We have also re-arranged this explanation within the methodology section to ensure that readers unfamiliar with this concept can easily understand its relevance and application within our model (See Lines 304 313).
- Line 250 briefly mentions historical hydrologic data without indicating the data sources, calibration metrics, or validation techniques. Include a clear explanation of the calibration and validation processes. A summary table with parameter ranges, calibration techniques, and validation outcomes would strengthen the model's reliability and replicability.

**Response:**

Thank you for your insightful suggestion regarding the hydrologic data used for calibration and validation of the hydrologic model components in our framework. To address your comment, we have taken the following actions:

- 1. **Clarify Data Sources**: We have mentioned the data sources in Section 3.3 Data Collection and Processing (See Lines 445 464). We ensure that this information is clearly stated and easily accessible to the reader.
- 2. **Detail Calibration and Validation Techniques**: We have discussed the calibration metrics and validation techniques in Section 3.4 Model Setup (See Lines 482 493).
- 3. **Include a Summary Table**: We agree that a summary table with parameter ranges, calibration and validation techniques and processes would greatly enhance the clarity and comprehensibility of our model. Although some details are provided in the Section 3.4 and Appendix, we have added a comprehensive summary table in the model setup section (See Page 19, Tab. 2. Details and representative results of calibration and validation for two hydrologic models).

 Table 2. Details and representative results of calibration and validation for two hydrologic models

 Note: \* represents the representative results for the two hydrologic models for urban area 1. KGE, NSE, R and B represents four types of performance metrics - the Kling-Gupta

 Efficiency (Kling et al., 2012), the Nash–Sutcliffe coefficient of efficiency (Nash and Sutcliffe, 1970), the correlation coefficient between simulated and observed streamflow and the percent bias (Gupta et al., 1999), respectively.

| Model                 | UWB-SM                                             | Muskingum-Cunge routing model        |
|-----------------------|----------------------------------------------------|--------------------------------------|
| Calibration parameter | See Tab. C3                                        | See Tab. C6                          |
| Dimensions            | $12 \times 9$                                      | $3 \times 9$                         |
| Calibration data      | Estimated monthly outflow;                         | Estimated monthly inflow and outflow |
| Data periods          | Calibration (1996-2020) and Validation (1971-1995) |                                      |
| Data sources          | USGS Current Water Data for the Nation database    |                                      |
| Calibration objective | Maximization of the KGE (Kling et al., 2012)       |                                      |
| Calibration algorithm | S-APSO framework (See Appendix D2)                 |                                      |
| Validation results*   | KGE=0.66, NSE=0.46, R=0.8, B=4.1%;                 | KGE=0.78, NSE=0.51, R=0.82, B=3.8%   |

• Line 280–285: The hydrologic and socio-economic data sources description is somewhat broad. Including a brief list of specific datasets used, such as U.S. Geological Survey data or climate records, and their date ranges would clarify

the model's foundation. The manuscript presents three spatial scales (city, intercity, and watershed) for experimental analysis, focusing on GI policies. However, these scenarios are presented with minimal contextual detail.

**Response:**

Thank you for your valuable suggestion regarding the description of hydrologic and socioeconomic data sources, as well as the contextual detail of our experimental design. To address your comment, we have taken the following actions:

- 1. Enhance Data Sources Description: We have revised Section 3.3 Data Collection and Processing to include more detailed information about the data sources. This section now lists data sources for all model parameters, and the specific datasets used also are shown, such as USGS Current Water Data for the Nation and the Global Historical Climatology Network daily (GHCNd) databases. Additionally, we have provided an overview of the data processing methods employed (see Lines 443-475).
- 2. Improve Experimental Design Description: We have enhanced Section 3.2 -Experimental Design by adding more details about the experiments. This includes the purpose of the experiments, the methods used, and the settings of key experiment parameters (See Lines 394 - 441). We ensure that the scenarios involving the three spatial scales (city, inter-city, and watershed) and their focus on GI policies are presented with sufficient contextual detail.
- Lines 315-327: Discussing the experimental conditions would help explain why specific scenarios were chosen, such as the "streamflow penalty" policy in line 319. A description of how this penalty reflects real-world practices would better convey the practical relevance of this scenario.

**Response:**

Thank you for your insightful suggestion regarding the discussion of experimental conditions in our manuscript. To address your comment, we have taken the following actions:

- Add Discussion of Experimental Conditions: We have expanded Section 3.2 Experimental Design to include a more thorough discussion of the experimental
  conditions (See Lines 394 441). This involves explaining the rationale behind
  selecting specific scenarios. For example, we provided key experimental parameter
  settings, such as the base penalty rate (See Lines 429 433).
- Motivation and Mechanism Analysis: We have included an analysis and description
  of the motivation and underlying mechanisms for considering a streamflow penalty
  strategy as WMs' watershed policy, which is inspired from the actual water
  withdrawal regulation in some regions, such as South Carolina (Nix and Rouhi, 2022)
   Over-extraction of surface water can incur penalty (See Lines 215 221).

**References**

- Nix HB, Rouhi Rad M. Water Withdrawal Regulation in South Carolina. Clemson (SC): Clemson Cooperative Extension, Land-Grant Press by Clemson Extension; 2022 Apr. LGP 1143. https://lgpress.clemson.edu/publication/water-withdrawal-regulation-in-south-carolina/.
- Lines 390-420: This section would be more accessible if the results for each spatial scale (city, inter-city, watershed) were divided into distinct subsections rather than being presented together. This would help readers understand the unique impacts observed at each scale.

**Response:**

Thank you for your valuable suggestion. We have divided Section 3.2 - Experimental Design into three distinct subsections for greater clarity (see Lines 394, 406, and 422).

• Lines 460-475: While the discussion briefly mentions the potential impacts of GI policies, it could provide more concrete suggestions for policymakers, especially regarding implementing penalty-based policies. For instance, specifying how such policies could be enforced across jurisdictions or considering potential limitations would strengthen the section.

**Response:**

Thank you for your insightful suggestion regarding the discussion of the potential impacts of GI policies. In response to your comment, we have expanded the discussion to include concrete suggestions for policymakers in Sections 4.2 and 4.3 (see Lines 617-619 and 680-682).

• Lines 490-500: This discussion could explore the model's adaptability to other similar regions or hydroclimatic conditions and cross-case comparisons. The authors could broaden the study's relevance by highlighting how it might apply to other areas. Expand the discussion on the policy implications of GIs, considering practical challenges and enforcement strategies. Including recommendations for policymakers, such as adaptive management guidelines or climate-resilient infrastructure planning, would enhance the study's applicability.

**Response:**

Thank you for your constructive suggestion regarding the exploration of our model's adaptability to other regions and hydroclimatic conditions, as well as the expansion of policy implications. In response, we have added a relevant discussion in Subsection 4.4 (see Lines 693-695). This discussion addresses: a) the feasibility and operability of applying the proposed model framework to regions with different hydroclimatic conditions, and b) the

potential for extending our model framework to simulate GI-driven socio-hydrologic dynamics in other watersheds under different water policies, such as water trading schemes (Eheart and Lyon, 1983)

**References**

- Eheart, J.W. and Lyon, R.M., 1983. Alternative structures for water rights markets. Water Resources Research, 19(4), pp.887-894.
- Lines 520-530: The conclusion summarizes the key findings well but could further emphasize the study's contributions and the potential for broader application. Highlight how the study advances the field of socio-hydrologic modeling, specifically regarding multi-agent frameworks for GI integration. A concluding sentence on how this framework could guide future studies in water management would leave a stronger impression.

**Response:**

Thank you for your valuable suggestion regarding the conclusion section of our manuscript. To address your comment, we have taken the following actions:

- 1. Enhance the Conclusion: We improved the conclusion section to further emphasize our study's contributions (See Lines 711 724).
- 2. **Broader Application**: We discussed the potential for broader application of our model framework to other watersheds, focusing on urban and watershed water management scenarios (See Lines 719 724).

**Technical corrections**

• Minor grammatical issues and ambiguous phrases appear throughout the text. For example, line 175, "up-and downstream imbalances," could be clarified as "upstream-downstream imbalances." A thorough proofreading would enhance readability.

**Response:**

Thank you for your careful review and for pointing out the minor grammatical issues and ambiguous phrases. We have changed all instances of "up-and downstream imbalances" to "upstream-downstream imbalances" (see Lines 563 and 601). Additionally, we have thoroughly proofread the entire manuscript to identify and correct any similar minor errors and ambiguous phrases.

• Figure 1 provides a schematic of the model, but its components and interconnections need to be labeled clearly. A legend or detailed figure description indicating each component's function within the model would enhance interpretability. Improvement Suggestion: Include a step-by-step description or flowchart illustrating the interactions between socio-economic factors, hydrologic processes, and policy influences. This would help in clarifying the multi-agent interactions and coupling between scales.

**Response:**

Thank you for your valuable suggestion regarding Figure 1. In response, we have added detailed descriptions for each component and included explanations of acronyms in the caption of Figure 1 to enhance readability and understanding (See Page 4, Figure 1).

• Figures 2 and 5: These figures would benefit from concise captions that specify what variables or trends they are intended to show. For instance, state whether they display policy impacts, flow distributions, or demand-supply imbalances explicitly.

**Response:**

Thank you for your insightful suggestion regarding the captions of Figures 2 and 5. In response, we have expanded the captions of the corresponding Figures 5 to 7 (see revised manuscript, page 20 for Figure 5, and page 24 for Figures 6 and 7) to clearly specify the variables or trends they illustrate.

Figure 3: While Figure 3 illustrates IGWM patterns for UWMs, more context on the visualized policy implications and the decision-making dynamics among UWMs would make the figures more impactful. Additionally, increasing the color contrast between scenarios in this figure would improve readability.
 Improvement Suggestion: Provide a rationale for each experimental scenario, focusing on its real-world applications. Consider adding a flowchart or table summarizing the experimental setups and their objectives to help readers follow the study's design.

**Response:**

Thank you for your valuable suggestion regarding Figure 3. To address your comment, 1) we have expanded the caption of the corresponding Figure

---

## Referee Report (RR2)

**Title: A Multiagent Socio-hydrologic Framework for Integrated Green Infrastructures and Water Resource Management at Various Spatial Scales**

The manuscript has substantially improved based on reviewer feedback. The key scientific and conceptual concerns have been addressed, and the additional sensitivity analysis and policy recommendations strengthen its relevance. However, minor typographical and grammatical errors persist. A final proofreading pass is recommended before submission.

**Remaining minor comments:**

Line 268: "textcolorred"?

2.1 Overview of the multiagent system architecture. I like the framing of a multiagent framework, but the language used in the text still confuses me. For example, I would suggest saying two agents instead of two ABM models In line 187. Similarly, in line 193 "we build an ABM and two multiagent systems" – it sounds really confusing. It works better for me when referring to the watershed scale model as the multiagent system and city-scale models as agents, which is developed using agent-based modeling technique. I will leave the decision to the authors.

Figure 1 (Conceptual Framework)

- The figure remains complex and difficult to interpret. May consider simplify A. City -scale IGWM (the processes are not readable anyway) and remove some of the dashed lines.

Figure 2 (B): the equation is not readable.

Figure 3 (A): the font size is too small.

---

## Author Response (AR2)

Dear Editor and Referees,

First of all, we deeply appreciate for all your valuable and insightful suggestions to the paper titled "*A Multiagent Socio-hydrologic Framework for Integrated Green Infrastructures and Water Resource Management at Various Spatial Scales*". We have checked the paper carefully and made minor revisions as referees suggested and also spent some time in checking the English writing and grammar. In addition, we have further carefully edited the manuscript to eliminate the shortcomings. All the revisions will be shown in the list as follows.

If you have any question about this paper, please do not hesitate to let us know. Your acknowledgement will be highly appreciated.

Sincerely yours,
Mengxiang Zhang & Ting Fong May Chui

Department of Civil Engineering, The University of Hong Kong, Hong Kong SAR, China

Corresponding Author: Dr. Ting Fong May Chui
Tel: +852-22194687;
Fax: +852-25595337.
E-mail: maychui@hku.hk

**Response to Referee #1**

Thank you so much for your objective and constructive comments to this study, which are of great significance to improve the quality of the paper. We have tried to revise the paper thoroughly following your instruction, and according to your other helpful suggestions, we have made revisions as following:

Overall Impression: I recommend accepting the manuscript but ask that the authors *consider* updating the overall focus on green infrastructures (GIs) to be more specific to rainwater harvesting (RWH) systems and to specify that this is a fictitious case study designed to demonstrate how such a framework could be carried out in the event that RWH becomes widely adopted and thus impacts theoverall water supply balance of cities. However, I would still accept the paper without this change, if the authors disagree and prefer to maintain focus on the terminology of GIs, because I can tell that they tried to add robust justification for this choice throughout the paper after a similar suggestion in the previous round of reviews. Their overall case study methodology uses a diverse set of GI types, so it might not be feasible to make this change. The actual methodology is very thorough. However, discussion of the technical results is limited and could benefit from deeper insights.

**Response:**

**1. About "scopes of GIs"**

Thank you for your suggestion to emphasize 'rainwater harvesting systems.' However, as you noted, this paper focuses on the broader impacts of GI on urban and watershed-scale water management. As defined in the paper, GIs encompass decentralized, nature-based solutions for capturing and recharging rainwater and stormwater. Specifically, we examine three types of GIs: (1) rainwater harvesting systems, which collect rainwater from rooftops before it reaches the ground; (2) stormwater harvesting systems, which capture runoff from land areas, including roofs and ground surfaces (Steffen et al., 2013); and (3) infiltration-based GIs, which enhance groundwater recharge by increasing the infiltration rates of pervious surfaces.

In theory, these three GI types exert distinct hydrologic effects on the urban water cycle and watershed-scale hydrologic regime. On one hand, this diversity complicates urban and watershed hydrological dynamics, posing challenges for water resource management. On the other hand, it provides urban water managers with a versatile toolkit to boost local water supplies and reduce water use costs. Our paper aims to explore these complex socio-hydrologic interactions driven by GIs in urban and watershed water management. Moreover, our results (see Figure 5 Page 20) highlight the distinct roles of these GI types across four urban water use patterns under varying hydrologic and climate conditions, underscoring their differential impacts on water management.

Thus, we believe it is essential to investigate the effects of diverse GI types on urban and watershed water resource management. We sincerely thank you again for your understanding and support of our paper's core focus. We also find your suggestion highly valuable - rainwater harvesting systems indeed play a prominent role in integrated GI and water resource management (IGWM) across spatial scales—and we plan to prioritize this aspect in future studies.

**2. About "Discussion"**

We appreciate your first-round comments and suggestions regarding the 'Discussion' section of our paper. In response, we have thoroughly revised Sections 4.2, 4.3, and 4.4 (Results and

Discussion) to deliver a more detailed, nuanced, and comprehensive analysis of our findings. These revisions directly address your concerns by deepening the discussion with richer insights into the socio-hydrological dynamics of IGWM across various spatial scales. Below, we outline the key modifications and explain how they enhance the paper, aligning with your feedback. These improvements cover critical aspects such as water costs and inter-community conflicts, communication between governance levels, the role of modeling assumptions, implications for water policy, and considerations of social equity.

*1) Water costs and inter-community conflicts*

In the original Section 4.2, we briefly noted an upstream-downstream imbalance in water resource access and the dual impact of GI on costs and equity. The revised version significantly expands this analysis with quantitative evidence and conceptual framing. For example:

- We now highlight that upstream areas (e.g., Urban Area 1) rely on surface water for 85% of their needs, while downstream areas (e.g., Urban Area 9) depend on it for only 73% in scenarios without GI. This disparity forces downstream communities to turn to costlier groundwater sources, exacerbating economic burdens (See Lines 575 - 578).

- GI adoption, while reducing city-scale water use costs (e.g., through stormwater harvesting), decreases downstream inflows by 12 - 18%, intensifying inter-urban conflicts. We liken this to a "tragedy of the commons," where localized benefits for upstream areas undermine watershed-scale equity (See Lines 578 - 584).

These revisions provide a clearer picture of how water costs and access disparities drive conflicts, emphasizing the need for a balanced approach to GI deployment across adjacent communities.

*2) Importance of communication between governance levels*

The original text hinted at the need for watershed managers (WM agents) to address upstream-downstream imbalances but lacked specificity. The modified Section 4.2 now underscores the critical role of communication and coordination between WM agents and urban water managers (UWM agents):

- We identify fragmented governance as a key driver of inefficiencies, where UWM agents optimize local costs (e.g., via GI) without considering downstream consequences, such as reduced streamflow (See Lines 618 - 625).

- To address this, we propose interactive data platforms that provide real-time hydrologic and economic data. These tools would enable adaptive GI investments and withdrawal limits, aligning city-scale actions with watershed-wide goals (See Lines 625 - 629).

This addition highlights the necessity of bridging communication gaps to mitigate conflicts and promote equitable resource management.

*3) The role of modeling assumptions*

Our original discussion did not fully explore the implications of modeling assumptions. The revised sections (4.2 and 4.3) now delve into how these assumptions shape our understanding of socio-hydrological dynamics:

- Markov Property: In Section 4.2, we clarify that the Markov property - where UWM agents base decisions on immediate inputs - leads to short-sighted strategies that

neglect downstream effects, such as reduced surface water availability. This exacerbates inequities, especially under variable hydrologic conditions driven by climate change (See Lines 619 - 625).

- Stackelberg Framework: In Section 4.3, we use this framework to model power asymmetries between WM and UWM agents. While WM agents enforce equity through penalties, UWM agents may counteract with actions such as excessive groundwater extraction, undermining aquifer sustainability and inflating costs. This dynamic reveals the challenges of policy enforcement in a hierarchical system (See Lines 669 - 675).

These insights deepen the technical discussion by linking modeling choices to real-world governance and equity outcomes.

*4) Implications for water policy as GIs become more popular*

The original manuscript lacked a robust policy discussion, which we have rectified in the revised Sections 4.3 and 4.4:

- We emphasize GI's dual nature: while it lowers local costs, uncoordinated adoption risks entrenching watershed-scale inequities. For example, downstream communities face reduced inflows and higher costs despite upstream GI benefits (See Lines 581 - 584).

- To address this, we recommend negotiated water-sharing agreements and adaptive management principles. In Section 4.3, we suggest using iterative penalty rate adjustments in streamflow penalty strategies, ensuring they balance cost reduction and equity in water resources distributions (See Lines 699 - 703). Section 4.4 concludes with a call for frameworks that harmonize localized GI benefits with broader sustainability goals (See Lines 714 - 716).

These revisions provide actionable policy insights, addressing the referee's call for deeper implications as GI adoption grows.

*5. Insights on social equity in water decisions*

Social equity was underexplored in the original text. The revised sections now offer a detailed equity analysis:

- In Section 4.3, we use the Gini coefficient to show how downstream regions disproportionately bear water use costs, even under streamflow penalty strategies. Upstream areas may face higher penalties, but downstream communities remain structurally disadvantaged due to reduced streamflow (See Lines 693 - 696).

- We propose redirecting cost savings from efficient water management to support vulnerable downstream stakeholders, alongside policies that prioritize their needs (Section 4.4). This approach aims to reduce disparities and promote sustainable equity across urban boundaries (See Lines 714 - 716).

This expanded focus ties technical results to social justice, enriching the discussion with a human-centered perspective.

In short, in Section 4.2, we enhanced with quantitative data (e.g., surface water reliance, inflow reductions) and conceptual framing (e.g., "tragedy of the commons") to explore water costs, conflicts, and governance communication.

In section 4.3, we introduced the Stackelberg framework and penalty strategy analysis, linking modeling assumptions to policy outcomes and equity challenges.

In Section 4.4, we added concluding remarks advocating for adaptive management and interactive data platforms to prioritize downstream equity.

**References**

Steffen, J., Jensen, M., Pomeroy, C.A. and Burian, S.J., 2013. Water supply and stormwater management benefits of residential rainwater harvesting in US cities. JAWRA Journal of the American Water Resources Association, 49(4), pp.810-824.

**Response to Referee #2**

Thank you very much again for your arduous and excellent comments and suggestions on our paper. Your comments at 1st round for the paper would take you a lot of energy, which is undoubtedly of great significance to improve the quality of the paper, especially the paper structure. We sincerely appreciate your understanding and acceptance of our work.